# Targeting *Pseudomonas aeruginosa* biofilm with an evolutionary trained bacteriophage cocktail exploiting phage resistance trade-offs

Fabian Kunisch [1,2,3,4], Claudia Campobasso[5,6], Jeroen Wagemans [5], Selma Yildirim[7], Benjamin K. Chan[3,4], Christoph Schaudinn[8], Rob Lavigne [5], Paul E. Turner [3,4,9], Michael J. Raschke[1,10], Andrej Trampuz [2,7] ✉ & Mercedes Gonzalez Moreno [2,7]

Spread of multidrug-resistant *Pseudomonas aeruginosa* strains threatens to render currently available antibiotics obsolete, with limited prospects for the development of new antibiotics. Lytic bacteriophages, the viruses of bacteria, represent a path to combat this threat. In vitro-directed evolution is traditionally applied to expand the bacteriophage host range or increase bacterial suppression in planktonic cultures. However, while up to 80% of human microbial infections are biofilm-associated, research towards targeted improvement of bacteriophages' ability to combat biofilms remains scarce. This study aims at an in vitro biofilm evolution assay to improve multiple bacteriophage parameters in parallel and the optimisation of bacteriophage cocktail design by exploiting a bacterial bacteriophage resistance trade-off. The evolved bacteriophages show an expanded host spectrum, improved antimicrobial efficacy and enhanced antibiofilm performance, as assessed by isothermal microcalorimetry and quantitative polymerase chain reaction, respectively. Our two-phage cocktail reveals further improved antimicrobial efficacy without incurring dual-bacteriophage-resistance in treated bacteria. We anticipate this assay will allow a better understanding of phenotypic-genomic relationships in bacteriophages and enable the training of bacteriophages against other desired pathogens. This, in turn, will strengthen bacteriophage therapy as a treatment adjunct to improve clinical outcomes of multidrug-resistant bacterial infections.

Worldwide emergence and spread of multidrug-resistant (MDR) *Pseudomonas aeruginosa* strains represents a critical public health threat, since currently available antibiotics are losing their efficacy. Furthermore, the development of new antibiotics is in decline, reinforcing the need for new approaches to treat infections with antibiotic-resistant strains[1–4]. As a ubiquitous Gram-negative pathogen, *P. aeruginosa* commonly causes opportunistic nosocomial infections, for instance, healthcare-associated pneumonia, bacteraemia and infections of the urinary tract[5]. Harnessing the selective permeability of its outer membrane, the ability to expel antibiotics from its cell

through efflux systems, mutational changes and horizontal gene transfer, *P. aeruginosa* is intrinsically resistant to several antibiotics[6]. Finally, transiently adapting gene and/or protein expression levels can cause changes in its motility, induce a persister state, or biofilm growth, generally conferring additional protection[6,7].

Biofilms are highly organised bacterial communities that adhere to one another and/or to surfaces and are embedded within a matrix of self-produced extracellular polymeric substances (EPS). Biofilms cause between 65% to 80% of human microbial infections[8–11]. Accounting for around 85–90% of the total organic fraction[12] in biofilms, the EPS of *P. aeruginosa* comprises exopolysaccharides (Pel, Psl, alginate), nucleic acids (external DNA) and proteins (CdrA/LapA)[13,14]. This bacterial lifestyle protects the enclosed cells from desiccation, impedes mechanical removal, and undermines the efficacy of antibiotics and disinfectants[15,16]. In addition, biofilms provide a safe haven from the innate and adaptive immune system, making them difficult to erradicate[17].

Bacteriophages (phages) have received renewed attention over the last decade as alternative antimicrobial agents to treat *P. aeruginosa* biofilms[18,19]. Lytic phages, naturally occurring viruses infecting bacteria, ultimately lyse their host cells. Three principal factors make phages promising antimicrobial agents for bacterial biofilm infections. First, once a phage infects and replicates within a bacterial biofilm cell, it causes a localised increase of infectious progeny. They subsequently spread further into the biofilm, thereby infecting, and lysing other bacterial cells. This in turn leads to the destabilization of the biofilm population and impairs any regeneration attempts. Second, various virion-associated depolymerising enzymes[20–24] and lysins[25] help phages reach and disintegrate their bacterial hosts and associated biofilm matrix[26–29]. Last, some phages have the ability to infect biofilm persister cells (dormant, non-dividing cells that exhibit increased tolerance to antimicrobials), which are killed once they revert to normal growth, as the phages initiate the lytic replication cycle[30], thus decreasing the risk of infection relapse.

Faced with the challenge of evolved bacterial phage resistance in a therapeutic setting, two main approaches have been a focus. Besides their expanded host range, phage cocktails – mixtures of several phages – exert distinct selection pressures on the target strain and thus reduce the probability of simultaneously evolved resistance towards multiple phages[31–33]. Phage training aims to select phages that show an improved circumvention of bacterial defence mechanisms due to de novo mutations or recombination events[34–36]. Furthermore, there have been directed evolution approaches building on the inherent properties of phages to improve phage killing efficacy in terms of, for instance, expanded host range[37], phage thermal stability[38,39], or greater bacterial suppression[34,40].

In our study, we integrated the combined improvement of the phages' host spectrum, antimicrobial and antibiofilm capabilities by implementing an in vitro directed evolution approach against biofilm bacteria. To further improve the efficacy of evolved phages against MDR *P. aeruginosa* strains, we designed a resistance-adapted phage cocktail. This study provides a phage training platform and helps to better understand bacterial phage resistance and enables us to identify different genomic parameters conferring enhanced phage efficacy.

## Results

### Training phages on biofilms by serial passage in a directed evolution assay

To overcome the challenges that *P. aeruginosa* biofilms pose to infection management, we developed an evolutionary serial passage assay (Fig. 1), that utilises a directed evolution approach, to simultaneously improve several phage infectivity parameters. This assay was implemented once as a proof of concept using four lytic *P. aeruginosa* phages (JS, MK, FIM, FJK), belonging to three distinct genera, which were trained on eight *P. aeruginosa* strains (PAO1, Paer03, Paer09, Paer33, Paer57, Paer60, Paer84, and Paer85), which also display

diversity in genomic phylogenetics, antibiotic resistance, and biofilm formation traits (Fig. 2a; Supplementary Fig. S1; Source Data D1, D2). During each round, pre-established (24 h) biofilms of ancestral bacteria were incubated with a mixture of the phages under isothermal microcalorimetric heat production control. After each passage, all samples showing a heat reduction greater than 75% (compared to growth control) and the undiluted phage samples (always included) were pooled together into the new phage mixture. In total, 30 rounds of evolution were performed.

Throughout the evolution assay, focusing on rounds 1, 5, 10, 15, 20, 25, and 30, we observed an overall increased calorimetric heat reduction after 8 h in consecutive rounds of evolution for each individual strain (Supplementary Fig. S2), except for Paer33 which was not susceptible to the evolving phages (Source Data D3). This improved activity was further confirmed at lower phage concentrations, achieving heat reductions equivalent to those from higher concentrations in previous rounds. Ultimately, we observed how approximately 10 PFU/ml (plaque forming units/ml) of the phage mixture resulted in heat reductions of 84.2% and 91.5% for the strains PAO1 (Supplementary Fig. S2a; round 30) and Paer57 (Supplementary Fig. S2d; round 15), respectively.

When focusing on the entire monitoring period (24 h) instead, we observed that five strains (PAO1, Paer03, Paer33, Paer57 and Paer84) co-incubated with the phage mixture reached heat levels equivalent to those from their respective growth controls (Supplementary Fig. S2). However, it should be noted that a delay in reaching the heat plateau was observed. Presumably, this delay corresponds to the time of bacterial suppression before phage resistance emerged. Among the analysed time points, the duration of this initial heat suppression is strain-specific and could reach up to 18.3 h (Paer57, round 5, $3.1 \times 10^6$ PFU/ml) (Source Data D3), excluding strains Paer09 and Paer85. In the case of Paer09, from round 3 onward, we observed a complete heat suppression (≥75%) for the entire monitoring period in the undiluted phage sample. Lower phage concentrations could still prevent the sample from reaching the growth control plateau level (with exceptions in samples containing a higher phage dilution). Similarly, the Paer85 heat production was continuously suppressed and did not reach growth control plateau levels.

### Combined improvement of three phage infectivity parameters

Upon completion of the phage evolution assay, to compare the infectivity parameters of the untrained, unevolved genomic phage ancestors with the evolved phages, we isolated individual phages from the phage mixtures of evolution round fifteen and thirty. For direct comparison of the phages' host range, antimicrobial biofilm activity, and antibiofilm efficacy we employed soft agar overlay spot assays, co-incubated pre-established biofilms with phages under isothermal microcalorimetry monitoring and determined the bacterial biofilm count reduction by real-time quantitative polymerase chain reaction (qPCR).

Based on plaque morphology and host strain, we isolated 31 individual evolved phages from the phage mixtures (17 from round 15 and 14 from round 30) (Supplementary Fig. S3). Of those, our analyses focused on a representative subset of ten: MK.R3-15/30, MK.R57-15/30, MK.R84-15/30, FIM.R60-15/30 and FJK.R9-15/30 named after the phage's genetic ancestor (e.g., MK), bacterial strain of isolation (e.g., Paer57) and the round of evolution it was isolated (round 15 or 30). The representatives are unique phages descended from the ancestral phages, showing distinct efficiency and host range. Similarly, from the 80 bacterial strains in our collection, three strains with distinct phage susceptibility and resistance profiles were selected, two of which were included in the evolution assay (Paer09 and Paer57) and one that was not included (Paer36), as target strains for further analysis of the antibiofilm and antimicrobial activity.

In terms of host range, each evolved phage could overcome the resistance of specific bacterial strains observed with the ancestral

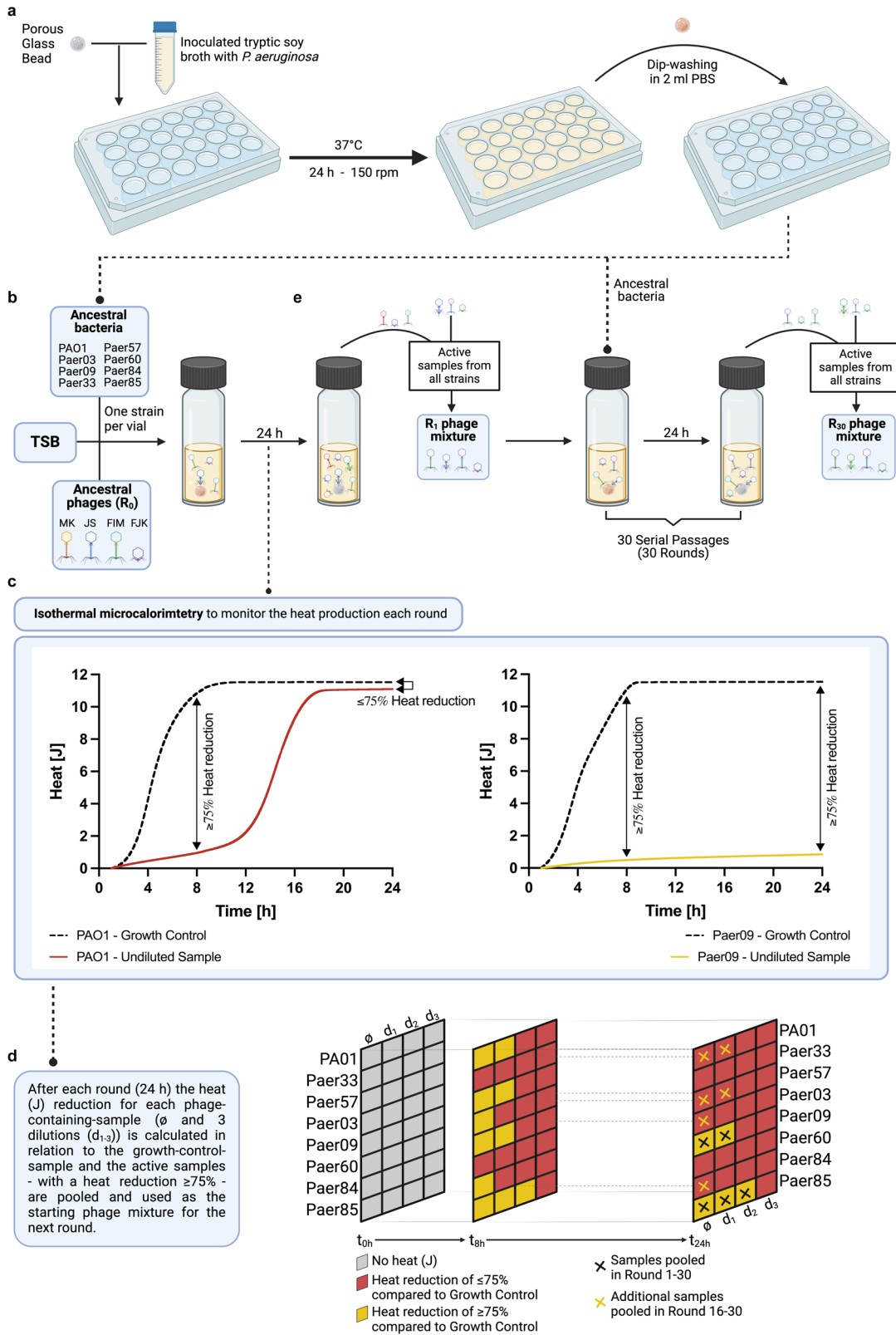

phage, while at the same time losing infectivity towards other strains (Fig. 2b). In this manner, all but two (evolved phages of FIM), showed an extended host range compared to their ancestors (Fig. 2c). In total, 67 infectivity gains were found on 25 strains, while 28 losses occurred on 14 strains. Between these two groups, only three strains (Paer58, Paer85, and Paer90) exhibited both gains and losses simultaneously. Compared to its genomic ancestors (MK and JK), phage MK.R57-30 showed the largest host range expansion by 76.5% and 42.9%, respectively. It could infect 11 additional strains, eight of which were initially not susceptible to any of the ancestral phages and three were susceptible to other phages but resistant to MK. This expansion enabled MK.R57-30 to infect 37.5% of all *P. aeruginosa* strains, a 20.0% increase compared to the ancestral phage FIM, which displayed the broadest host range among the ancestral phages. For the evolved

**Fig. 1 | Experimental design of the antibiofilm evolution assay.** Experimental setup of the evolution assay to extend the host range and improve the antimicrobial and antibiofilm activity of bacteriophages (phages). **a** Biofilm formation on porous glass beads by incubation with *P. aeruginosa* in tryptic soy broth (TSB) under agitation and 37 °C. Dip-washing of beads with 24 h pre-established biofilm in sterile phosphate-buffered saline (PBS) before transferal into the calorimetric ampules. **b** Representation of one calorimetric ampule containing a 24 h pre-stablished biofilm bead of one *P. aeruginosa* strain in TSB and the phage mixture at round 0 (R0) containing the four ancestral phages. A total of 43 ampules (one growth control and four phage mixture dilutions per strain and three sterility controls) were used during each round of evolution. **c** Heat production (J) was recorded for each individual ampule by isothermal calorimetry during 24 h. **d** At the end of each round, the percentage heat reduction of phage-containing ampules relative to growth control ampules was determined at a 24 h (rounds 1–15) or 8 h (rounds 16–30) time point. Samples showing a heat reduction equal or above 75% (referred to as active samples) were selected and pooled together with the undiluted phage samples (always included) into the new phage mixture. **e** Across all eight bacterial strains, the undiluted and the active samples after each round were pooled together, creating the phage mixture that served as starting point for the next round, and hence was added to pre-stablished biofilm beads anew. In total, 30 rounds of evolution were performed. Figure 1 was created with BioRender.com and released under a Creative Commons Attribution-NonCommercial-NoDerivs 4.0 International license (https://creativecommons.org/licences/by-nc-nd/4.0/deed.en).

phages, we calculated a 38.7% greater increase in infectivity among the eight bacterial strains included in the evolution assay than in the strains not included (Fig. 2d). Focusing instead on the entire collection of *P. aeruginosa* strains ($n = 80$), 30 strains (37.5%) were not susceptible to any phage (unevolved and evolved), and overall, the evolved phages (rounds 15 and 30) were able to increase the number of susceptible strains to 47, an increase of 14.6%.

Regarding the antimicrobial activity against pre-established biofilms, the evolved phages showed a higher suppressive effect on bacterial cells than their respective unevolved ancestral phages. In addition, the evolved phages could suppress bacterial heat production for a longer period before bacterial outgrowth occurred, presumably corresponding to the emergence of phage resistance (Source Data D4). When incubated with Paer09, the evolved phage FJK.R9-30 revealed a minimum heat flow of 11.6 μW, corresponding to the highest suppression effect on bacterial heat production prior to their outgrowth. This represents a 68.4% ($p = 0.0004$) greater heat flow suppression compared to the ancestral phage FJK (36.6 μW) (Fig. 3b). At the same time, phage FJK.R9-30 resulted in a 64.6% longer lag time than the ancestral phage (13.8 h vs. 22.7 h, $p = 0.0085$). Ancestral phage MK showed a minimum heat flow of 77.2 μW on Paer57, whereas the evolved phage MK.R57-30 reached a 96.5% ($p = 0.0053$) greater reduction at 2.7 μW (Fig. 3f). With a 375.7% ($p = <0.0001$) longer lag time, MK.R57-30 suppressed the bacterial growth of Paer57 for 15.7 h (ancestral phage MK, 3.3 h) (Fig. 3h). Against Paer36, a strain not included in the evolution assay, the evolved phage MK.R3-30 could both suppress bacterial heat production for longer (15.2 vs. 6.1 h, $p = 0.0105$) and reduce heat flow to lower levels (20.3 vs. 80.4 μW, $p = 0.0369$) than the corresponding unevolved ancestral phage (MK).

Across all tested strain-phage combinations, we could show that the phages evolved for thirty rounds demonstrated a greater antibiofilm activity with lower biofilm cell counts compared to their unevolved ancestors and, with one exception, than their counterparts evolved for fifteen rounds (Source Data D5). After 6 h co-incubation of phage FJK.R9-30 and Paer09, the cell count was 74.8% ($p = 0.0010$) lower compared to results from ancestral phage FJK (Fig. 3a). Compared to the ancestral phage MK, the evolved phage MK.R57-30 showed a 99.7% ($p = 0.2473$) and 86.7% ($p = 0.0431$) greater cell count reduction of Paer57 biofilm cells after 6 and 24 h, respectively (Fig. 3e). After 3 h of incubation (Paer57) phage MK.R57-30 resulted in a 58.7% ($p = 0.0032$) lower cell count than phage MK.R57-15. Against Paer36 (not included in the evolution), the phage evolved for thirty rounds (MK.R3-30) outperformed the ancestral phage (MK) by 85.4% ($p = 0.0529$) after 3 h of incubation. Similarly, after 6 h of incubation (Paer36) phage MK.R3-30 showed an 85.0% ($p = 0.0067$) greater biofilm cell count reduction than the phage MK1.R3-15 (Fig. 3i).

## Directed evolution primarily drives mutational events in structure-associated genes

To better understand the evolutionary mutational changes that underpin the improvement of our evolved phages' infectivity, we sequenced all unevolved and evolved phages. By comparing the genomes of the phages improved over fifteen and thirty rounds with their genetic ancestors, we anticipated to find mutations located among structural protein-coding genes associated with host recognition and the degradation of extracellular polymeric substances.

By directed evolution, phages FJK.R9-15 and FJK.R9-30 lost a part of the early gene region (FJK.R9-15, 2431 bp, 11 genes; FJK.R9-30, 1348 bp, 5 genes) and present single-nucleotide polymorphisms (SNP) in their genome leading to missense mutations in two genes encoding an amidoligase (gp22) and an L-glutamine-D-fructose-6-phosphate-aminotransferase (gp23), as well as mutations in four genes encoding structural proteins (Fig. 4a). Although not in the same position, all SNPs for both evolved phages are in the same genes. For the structural proteins, gp52 contains a peptidoglycan transglycosylase (HHPred; $7.5 \times 10^{-41}$ e-value) domain, that can degrade the bacterial cell wall peptidoglycan layer during the phage infection step[41]. Likewise, gp62 has similarity with tail tubular protein A (TTPA) of *Klebsiella* phage KP32 (HHPred; $8.6 \times 10^{-5}$ e-value) which has EPS depolymerase activity[42]. The tertiary structure of gp62 predicted with ColabFold (Fig. 4d) illustrates the mutation of amino acid 98 from a cysteine to a phenylalanine. Compared to TTPA, gp62 contains an additional α-helix close to the β-sheets, in which the mutation occurred, that could have an impact on the enzymatic activity, thereby potentially explaining the increased antibiofilm activity of the evolved phage (Supplementary Fig. S4).

Both phages FIM.R60-15 and FIM.R60-30 contained SNPs in tail fibre coding genes (gp42 and gp43). The phage evolved for thirty rounds displayed an additional SNP in gene gp8, encoding a hypothetical protein, and gp44, containing a tail fibre assembly domain (HMMER; $3.3 \times 10^{-54}$ e-value) (Fig. 4b). Of those tail fibre proteins, gp43, has an undefined catalytic activity (Mll0443 protein; HMMER; $5.3 \times 10^{-19}$ e-value) and a peptidase S74 domain (HMMER, $7.7 \times 10^{-9}$ e-value). This domain is commonly found in phage endosialidases (polysaccharide depolymerases), where it acts as an intramolecular chaperone[43,44]. It is therefore likely that gp43 functions as an endosialidase and can specifically degrade bacterial polysialic acid on the phage's path to the bacterial cell membrane[45], thereby increasing their effectiveness, as more susceptible strains can be infected.

Evolved phages derived from the phages MK and JS (MK.R3-15, MK.R3-30, MK.R84-15, MK.R84-30, MK.R57-15, MK.R57-30) appear to be a recombination of these two ancestral phages. Therefore, to identify SNPs involved in the observed improved phenotypic changes, we focused on the structural gene region (Fig. 4c). Only unique SNPs compared to both ancestral phages were considered. This analysis revealed an accumulation of SNPs across five genes, encoding the head-tail joining protein, the tail protein with Baseplate J domain (HMMR; $5.6 \times 10^{-49}$ e-value)[46], the endosialidase-like tail fibre protein (HMMR; $3.8 \times 10^{-9}$ e-value)[44], the tail fibre assembly protein (HHblits; $5.7 \times 10^{-9}$ e-value) and one structural protein. This structural protein, gp11, shows similarity to a lipase (HMMR; $1 \times 10^{-120}$ e-value) and, consequently, could help the phage hydrolyse encountering lipids (e.g., short-chain fatty acids, long-chain acylglycerols)[47].

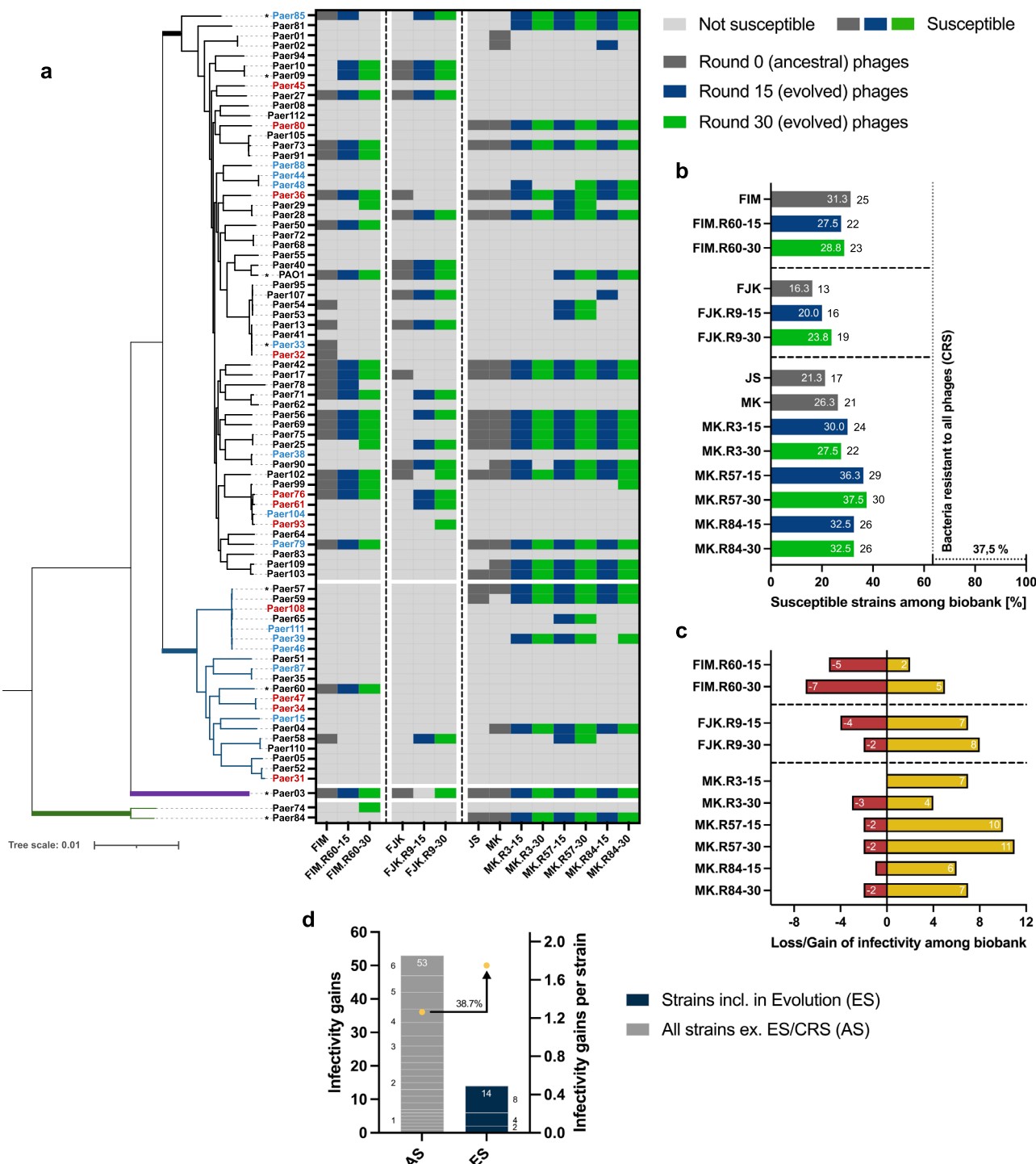

**Fig. 2 | Host range analysis of unevolved and evolved phages. a** Phylogenetic tree based on the core genes (3,667 of 19,556 genes) of the *P. aeruginosa* strain collection (80 strains) encompassing four genomic clusters (Cluster 1, black branch, 71.3%; Cluster 2, blue branch, 25.0%; Cluster 3, violet branch, 1.3%; Cluster 4, green branch, 2.5%). Classification of each strain according to its resistance profile (4MRGN, red lettering; 3MRGN, blue lettering; MRGN, multi-resistance Gram-negative). Strains marked with an asterisk were included in the experimental evolution. Activity determination of the unevolved (dark grey) and evolved (R15, blue; R30, green) phages against the bacterial collection by spot assay overlay (heat map). **b** Percentual (white lettering) and absolute (black lettering) infectivity of each phage among the bacterial collection. Vertical dotted line represents the percentage of bacterial strains susceptible to at least one phage (62.5%). 37.5% (30 strains) of bacterial strains were not susceptible to any phage (CRS, completely resistant strains). **c** Representation of the infectivity gains and losses of the evolved phages. An infectivity gain (yellow bar) is defined as the capability of an evolved phage to infect a bacterial strain that was initially not susceptible to the phage's genomically closest unevolved progenitor phage. A loss of infectivity (red bar) occurred when a bacterial strain is susceptible to the unevolved ancestral phage but resistant to the corresponding evolved phage. **d** Representation of the cumulated infectivity gains of the evolved phages among the evolution strains (ES) or all strains (AS, excluding the ES and CRS). Among the ES (*n* = 8), three strains (stacked blue boxes, numbers at the side of the boxes indicate the infectivity gains on the according strain) resulted in 14 infectivity gains (Paer85, 8 gains; PAO1, 4 gains; Paer09, 2 gains). The 53 infectivity gains among AS (*n* = 42) correspond to gains on 22 strains (stacked grey boxes, numbers at the side of the boxes indicate the infectivity gains on the according strain). For direct comparability of the two strain populations (ES and AS), the cumulated values were divided by the number of strains in each group, resulting in averaged gains per strain (yellow circles).

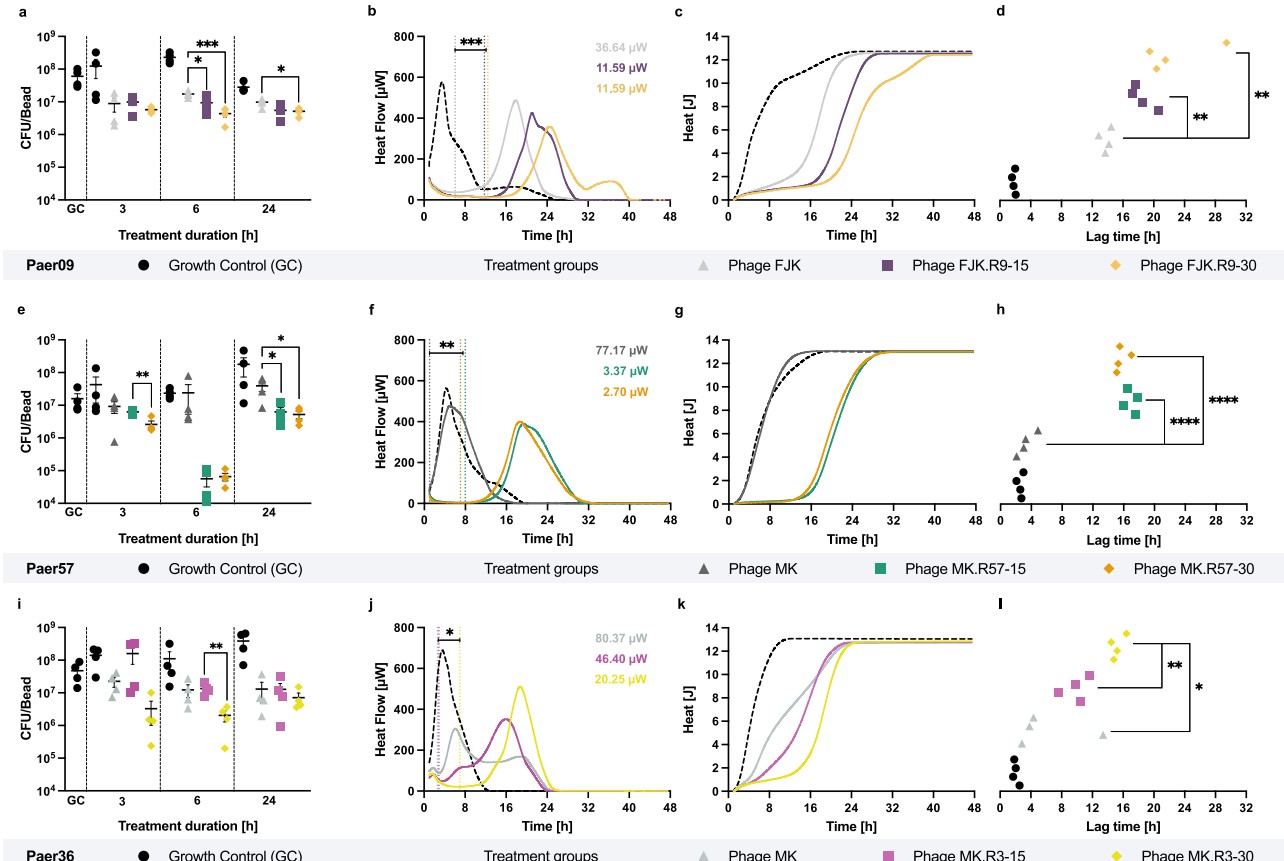

**Fig. 3 | Antibiofilm and antimicrobial activity of unevolved and evolved phages. a** qPCR-determined bacterial load (CFU/bead) of 24 h pre-established Paer09 (included in the evolution) biofilm untreated (GC) and after co-incubation (3, 6 and 24 h) with phage FJK (unevolved), FJK.R9-15 (15 rounds of evolution) and FJK.R9-30 (30 rounds of evolution). The *p*-values are 0.0365 (6 h, FJK vs. FJK.R9-15), 0.0010 (6 h, FJK vs. FJK.R9-30) and 0.0242 (24 h, FJK vs. FJK.R9-30). **b** Heat flow (μW) curves measured for 48 h of Paer09 biofilm co-incubated with phage FJK, FJK.R9-15, FJK.R9-30, or untreated (GC, dashed line). Vertical dotted lines indicate the minimum heat flow (μW values in the right upper corner) detected in each treated sample, related to the highest suppressive effect of the phage on bacterial cells. The *p*-value is 0.0004 (FJK vs. FJK.R9-15/30). **c**, **d** Respective heat (J) curves of phage-treated Paer09 biofilm and the calculated suppression time for each tested condition. The *p*-values are 0.0014 (FJK vs. FJK.R9-15) and 0.0085 (FJK vs. FJK.R9-

30). Identical setup for Paer57 (included in the evolution) treated with phage MK, MK.R57-15, MK.R57-30 or untreated (GC). The *p*-values are 0.0032 (**e**, 3 h, MK.R57-15 vs. MK.R57-30), 0.0493 (**e**, 24 h, MK vs. MK.R57-15), 0.0431 (**e**, 24 h, MK vs. MK.R57-30), 0.0056 (**f**, MK vs. MK.R57-15), 0.0053 (**f**, MK vs. MK.R57-30) and <0.0001 (**h**, MK vs. MK.R57-15/30). Identical setup for Paer36 (not included in the evolution) treated with phage MK, MK.R3-15, MK.R3-30 or untreated (GC). The *p*-values are 0.0067 (**i**, 6 h, MK.R3-15 vs. MK.R3-30), 0.0182 (**j**, MK vs. MK.R3-30), 0.0369 (**j**, MK.R3-15 vs. MK.R3-30), 0.0105 (**l**, MK vs. MK.R3-30) and 0.0012 (**l**, MK.R3-15 vs. MK.R3-30). All experiments were conducted in fourfold and the curves (**b**, **c**, **f**, **g**, **j**, **k**) display the mean. The error bars represent the standard error of the mean. Statistical analysis was conducted by unpaired two-tailed Student's *t* test and *p*-values indicated with asterisks (*, *p* = <0.05; **, *p* = <0.01; ***, *p* = <0.001; ****, *p* = <0.0001). Source data are provided as a source data file.

## Impaired bacterial escape from evolved phage predation

Since our above results showed the genotypic improvement of several phage infectivity parameters, we anticipated bacteria to have more trouble escaping phage predation. To investigate this question, we co-incubated planktonic bacterium Paer09 with either ancestral phage FJK or one of the evolved phages (FJK.R9-15/30) at an MOI of 0.001 for three days. We then sequenced the surviving bacteria, isolated after co-incubation, to identify their mutational changes. Further, we employed soft agar overlay spot assays to examine their susceptibility to the co-incubation phages, as well as phages MK and MK.R3-15 to assess resistance trade-off events.

Within the experiment, bacteria developed varying degrees of resistance to the treatment phage, with a high rate of cross-resistance among FJK phages (Fig. 5a). Altogether, 43 of the 48 isolates (89.6%) were found to be resistant to phage FJK (*n* = 16, isolates from FJK incubation), while for evolved phages FJK.R9-15 (*n* = 17) and FJK.R9-30 (*n* = 15), 25 (52.1%) and 21 (43.8%) resistant isolates were found, respectively. Those 21 FJK.R9-30 resistant isolates showed cross-resistance to FJK.R9-15 and FJK, as demonstrated by Spearman's rank correlation coefficient (r_s) at 0.99 (*p* = <0.0001) and 0.79

(*p* = <0.0001), respectively. Bacterial exposure to phage FJK resulted in an immediate optical density increase, whereas phages FJK.R9-15 and FJK.R9-30 suppressed the optical density for 17.4 h and 21.9 h, respectively (Fig. 5d–g). Furthermore, we observed that out of the eight bacterial replicates co-incubated with each phage, both evolved phages had one replicate with a low increase in optical density, reaching 0.3 (FJK.R9-15) and 0.1 (FJK.R9-30) after three days. These values are close to the negative controls (OD_{600} of 0.09) and possibly indicate bacterial eradication.

Among the 48 bacterial isolates, we identified mutations in only six genes (with only one altered gene per mutant), likely conferring resistance to phage predation. Overall, frameshift and nonsense mutations conferred greater resistance against the tested phages than missense variants of the same proteins (Fig. 5a). Included among those encoded proteins were all four enzymes (RmlA, RmlB, RmlC and RmlD) involved in the L-rhamnose biosynthesis pathway producing dTDP-L-rhamnose (Fig. 5b, c). The majority (52.1%, *n* = 25) of strains had a mutated version of the *rmlA* gene (glucose-1-phosphate thymidylyl-transferase). dTDP-L-rhamnose links to the lipopolysaccharides' (LPS) core oligosaccharide to act as the acceptor molecule for the covalent

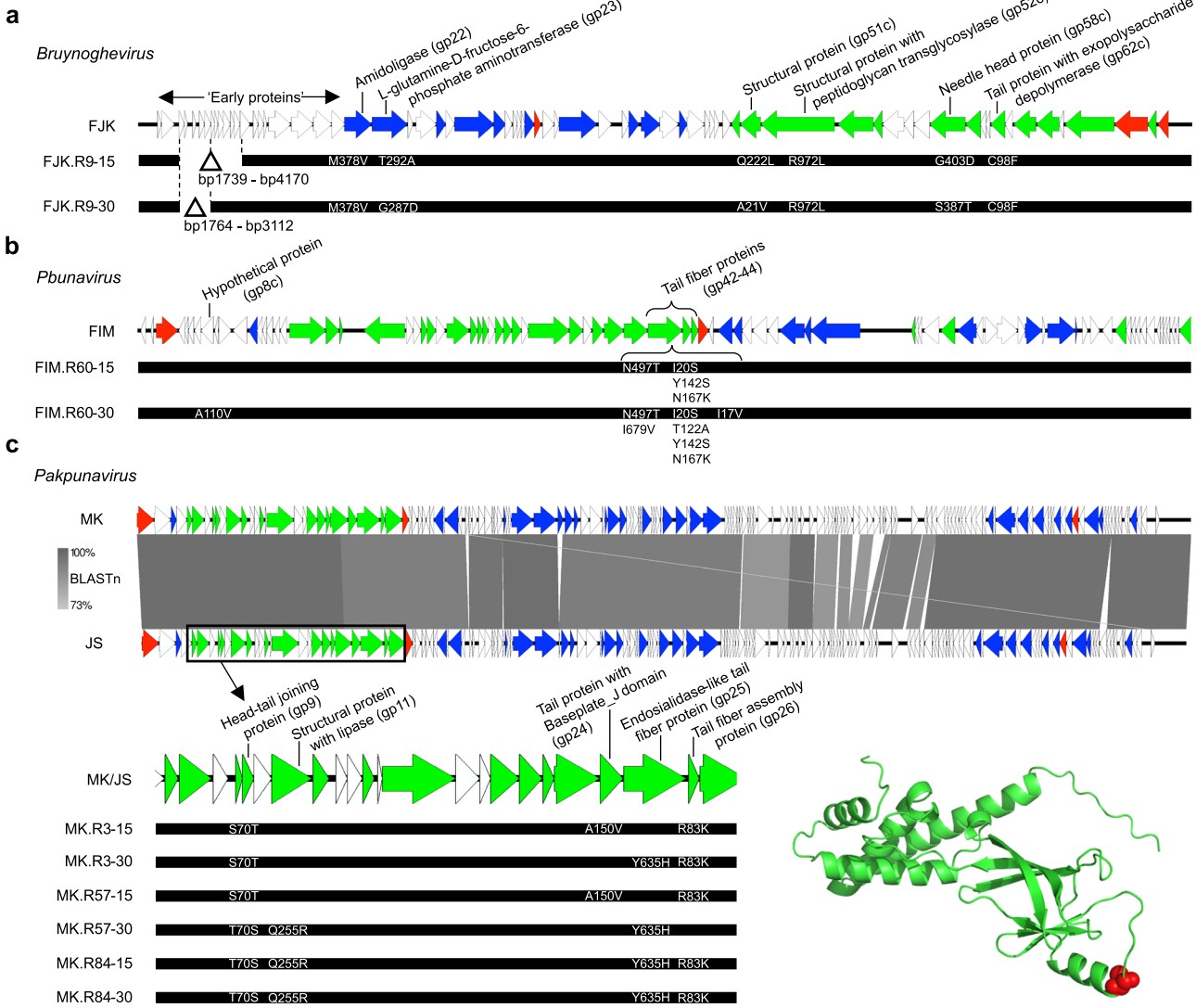

**Fig. 4 | Schematic genome representation of the evolved phages. a–c** For each bacteriophage (phage) genus, a genome map is provided for the ancestral phage (FJK, FIM, or MK/JS) with the relevant functional annotations highlighted on top. Each coloured arrow represents a coding sequence with white encoding hypothetical proteins, blue (DNA) metabolism-associated proteins, green virion proteins, and red genome packaging or cell lysis proteins. The genomic changes in the evolved phages after round 15 and round 30 are displayed below each ancestral phage genome, with deletions being shown as a delta symbol and the individual SNPs annotated. For the *Pakpunavirus* members, a BLAST comparison between both ancestral phages is also shown. **d** Tertiary structure prediction of EPS depolymerase FJK_gp62. In red, the cysteine residue on position 98 is highlighted, which is mutated to the aromatic amino acid phenylalanine in the evolved phage.

attachment of the A- or B-band O-antigen in *P. aeruginosa*[48,49]. The enzyme involved in this linkage is the alpha-1,3-rhamnosyltransferase (WapR) which was also mutated in several isolates ($n = 8$). Loss of the O-antigen has been associated with a reduced virulence, ineffective swimming and swarming motility and less protection from phagocytosis[50–52]. In one isolate, PslA, part of the Psl biosynthesis pathway, was linked to phage resistance. Psl is essential for biofilm attachment, formation, and differentiation in non-mucoid *P. aeruginosa*[53].

Six representative bacterial isolates, each mutated in only one of the six identified genes (*rmlA*, nonsense; *rmlB*, missense; *rmlC*, frameshift; *rmlD*, nonsense; *wapR*, frameshift; *pslA*, nonsense), were selected for further characterisation in terms of their growth, virulence in *Galleria mellonella* (*G. mellonella*), biofilm formation and motility (Supplementary Fig. S5; Source Data D6). Compared to the naive Paer09 bacterium (lag time of 3.5 h), the OD measured growth curves of the *rmlC* (7.6 h) and *rmlD* (9.4 h) mutants showed a 116% and 166% prolonged lag time, respectively, while the other mutants showed no remarkable difference. All mutant strains exhibited lower virulence in vivo, resulting in a higher larval survival rate after 100 h, when all naive Paer09-infected larvae were dead. While crystal violet staining demonstrated reduced biofilm biomasses for the *rmlA*, *rmlB* and *rmlC* mutants, only the *rmlC* mutant had a lower biofilm cell count on porous glass beads. The *rmlA* ($p = <0.01$), *rmlC* ($p = <0.001$) and *rmlD* ($p = <0.0001$) mutants showed a reduction in swarming motility. Contrarily, the *wapR* mutation (frameshift) resulted in a 10.5% ($p = <0.05$) greater swarming motility and both the *rmlB* and *wapR* mutants had an increased swimming motility of 32.8% ($p = <0.01$) and 68.0% ($p = <0.0001$), respectively (Supplementary Fig. S5f, g). The other mutants displayed a reduced swimming motility.

We further discovered a resistance trade-off between FJK-phages and MK-phages, illustrated by a negative correlation of susceptibility to phage FJK versus phage MK ($r_s = −0.59$, $p = <0.0001$) and MK.R3-15 ($r_s = −0.58$, $p = <0.0001$). Accordingly, 43 isolates (89.6%), of which 42 were resistant to phage FJK, had an increased susceptibility towards phage MK (EOP = 200 – 3000) and phage MK.R3-15 (EOP = 43 – 10714). Plasmid complementation of the mutated genes with the wild-type genes in the six representative bacterial isolates resulted in a reversal

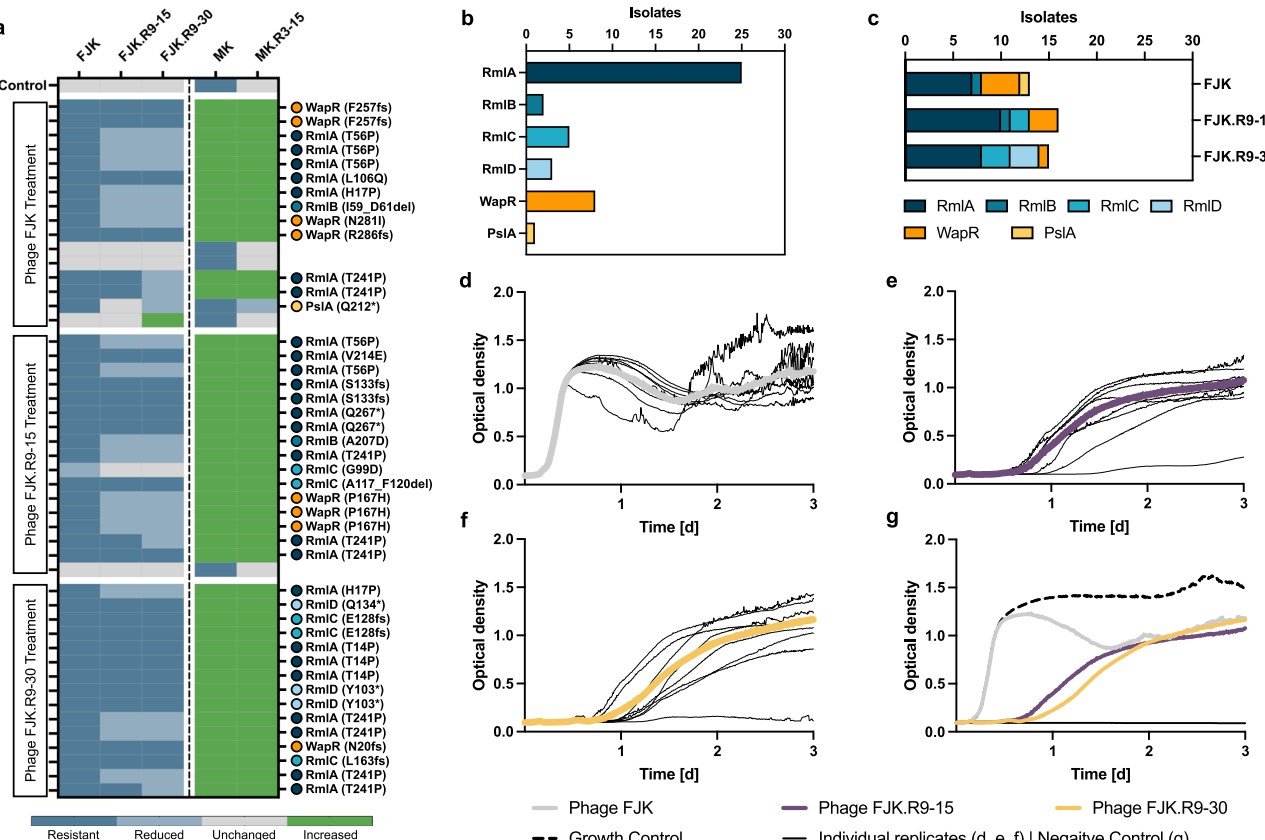

**Fig. 5 | Phenotypic and genomic characterisation of phage-treated Paer09 isolates. a** Efficiency of plating (EOP) of five bacteriophages (phages) (FJK, FJK.R9-15, FJK.R9-30, MK, and MK.R3-15) for 48 Paer09 strains treated with individual phages (FJK, FJK.R9-15, FJK.R9-30). The EOP is defined as the concentration ratio of a phage on the phage-treated isolate (numerator) and the naive Paer09 strain (denominator; Control). An EOP above 10 was considered as increased efficiency (green), while an EOP of 0.1 to 10 was ranked as unchanged efficiency (grey). A reduced efficiency was defined as an EOP between 0.001 and 0.1 (light blue). When no individual plaques were visible or the EOP was equal to or under 0.001 the isolate was determined resistant to the phage (blue). Mutation-associated protein of each strain with amino acid change in parentheses. A lack of labelling refers to isolates in which no apparent mutation could be identified. **b** Illustration of the cumulative number of isolates by mutant protein. **c** Illustration of the number of isolates by mutant protein according to the treatment group (phage FJK; phage FJK.R9-15; phage FJK.R9-30). Optical density (OD600) measurements of a three-day co-incubation of planktonic naive Paer09 strain with phage FJK (**d**), phage FJK.R9-15 (**e**) and phage FJK.R9-30 (**f**) at an MOI of 0.001. The thin black lines represent each individual replicate ($n = 8$), and the thick coloured line represents the average. For phages FJK.R9-15/30 the completely suppressed replicate was excluded from the average calculation. **g** Illustration of the averaged co-incubation curves for phages FJK (grey line; panel d), FJK.R9-15 (violet line; **e**) and FJK.R9-30 (yellow line; **f**) over three days in comparison to a growth control (dotted black line) and negative control (black line). Source data are provided as a source data file.

of this trade-off, which was most prominent in the *rmlA*, *rmlC*, and *wapR* isolates (Supplementary Fig. S5c). While the ancestral phage MK was not able to infect the naive Paer09 strain, its evolved descendant (MK.R3-15) could infect Paer09 with an EOP of 0.00019, compared to its own host strain (Paer03). Among the 48 bacterial isolates, five isolates (10.4%; no apparent mutations) showed the same phage susceptibility profile as the control, except one who had an increased efficiency of plating for phage FJK.R9-30 (Fig. 5a). These five isolates and the control showed resistance to phage MK, while no isolate was resistant to the evolved phage MK.R3-15.

### Combination of phages to exploit a bacterial phage resistance trade-off

Building on this resistance trade-off, we combined FJK.R9-30 and MK.R3-15 into a cocktail. Using optical density monitoring, isothermal microcalorimetry and qPCR, we then compared the cocktails' planktonic and biofilm antimicrobial activity, as well as its antibiofilm efficacy, with the individual phages (Source Data D7). Given the simultaneous antagonistic selective pressures of the cocktail, we anticipated that it would generate improved results in all three dimensions.

Compared with the individual phage (FJK.R9-30, MOI of 0.001) (Fig. 5g), the cocktail could increase the lag time of planktonic Paer09

growth by 565.0% (MOI of 0.001) and 126.5% (MOI of 0.0001) (Fig. 6i). While FJK.R9-30 resulted in one replicate (after three days) with an optical density of 0.1, comparable to the negative controls (OD$_{600}$ of 0.09), the cocktail caused two (MOI of 0.0001, OD$_{600}$ 0.09 and 0.1, after three days; Fig. 6h) and four (MOI of 0.001, OD$_{600}$ 0.08–0.09, after seven days; Fig. 6g) replicates, to presumably go extinct.

Regarding the antimicrobial biofilm activity, the co-incubation of Paer09 biofilm with the individual phage MK.R3-15, revealed a heat flow and heat curve like the growth control (Fig. 6d,e). In contrast to that, FJK.R9-30 could supress the bacterial heat flow (11.6 µW) and increase the lag time (22.7 h). When we tested the phage cocktail (FJK.R9-30 + MK.R3-15), the minimum heat flow was further reduced to 7.2 µW ($p = 0.0404$). Concordantly, the phage cocktail, with a lag time of 49.8 h, further increased the duration of heat suppression by 119.1% ($p = 0.0051$) (Fig. 6f). The maximum slope, indicative of the growth rate, was reduced to 0.9 J/h compared with 1.6 J/h ($p = 0.0056$) and 2.1 J/h ($p = 0.0076$) for phage FJK.R9-30 and the growth control, respectively. At all three tested time points (3, 6, and 24 h), the individual phage FJK.R9-30 presented a higher biofilm cell count reduction than the phage cocktail (Fig. 6c).

Isolates retrieved after the co-incubation with the phage cocktail ($n = 18$) revealed mutations in three proteins (RmlC, WapR and glycoside

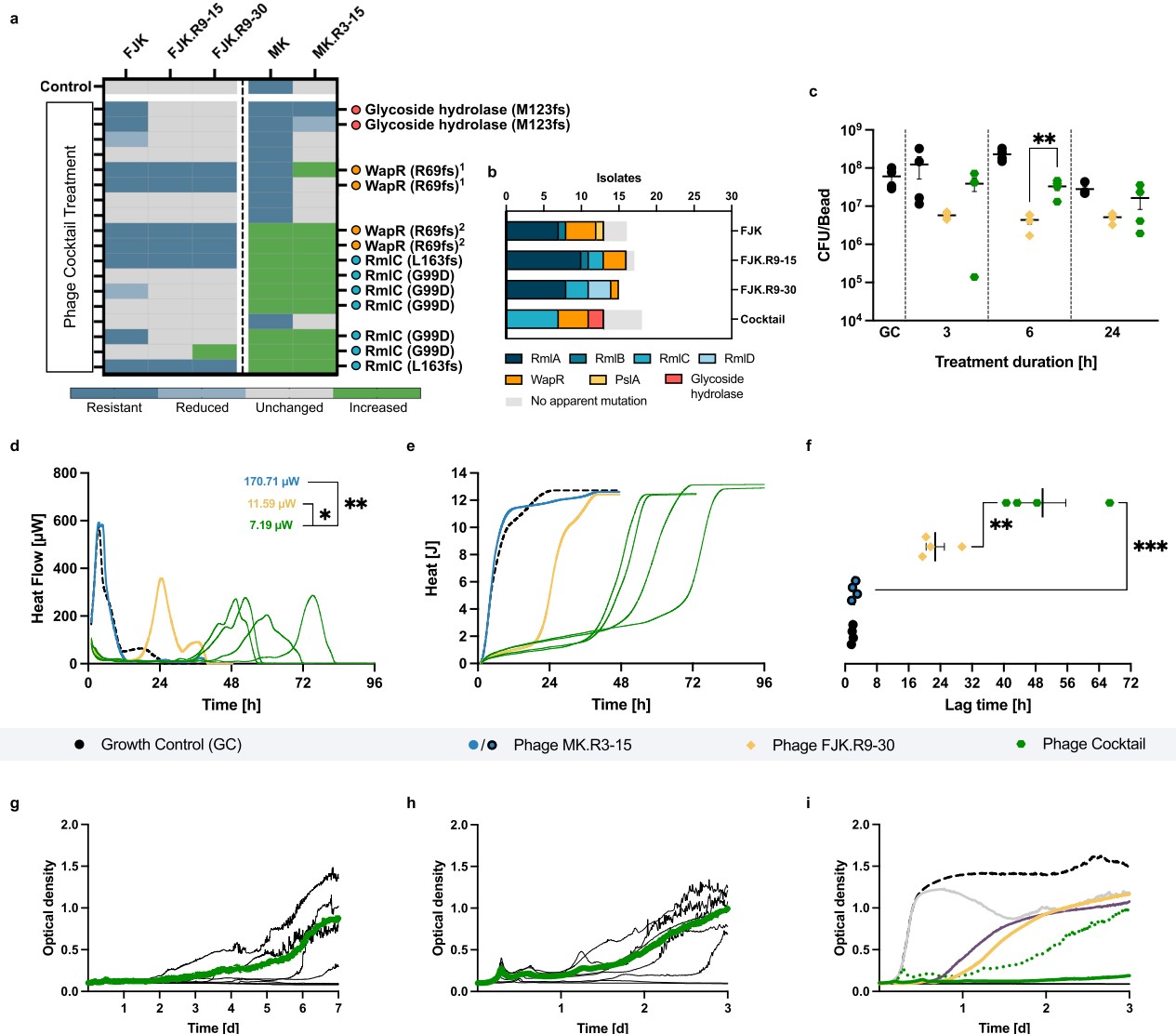

**Fig. 6 | Antibiofilm and antimicrobial activity of the phage cocktail. a** Efficiency of plating (EOP) of five phages (FJK, FJK.R9-15, FJK.R9-30, MK, and MK.R3-15) on 18 Paer09 strains treated with the phage cocktail (FJK.R9-30 + MK.R3-15). EOP is defined within the methods. Mutation-associated protein of each strain with amino acid change in parentheses ([1]c.203_204dupTC p.R69fs and [2]c.205delA p.R69fs). Lack of labelling refers to isolates with no apparent mutation. **b** Illustration of the number of isolates by mutant protein and treatment group (FJK; FJK.R9-15; FJK.R9-30; Cocktail). **c** qPCR-determined bacterial load (CFU/bead) of 24 h pre-established Paer09 biofilm untreated (GC) and after co-incubation (3, 6 and 24 h) with FJK.R9-30 or the cocktail. The *p*-value is 0.0077 (6 h, Cocktail vs. FJK.R9-30). **d, e** Heat flow (μW) and heat (J) curves measured for 48, 72 and 96 h of Paer09 biofilm co-incubated with MK.R3-15, FJK.R9-30, cocktail (individual replicates) or untreated (GC, dashed black line). Minimum heat flow values in the right upper corner (**d**). The *p*-values are 0.0404 (FJK.R9-30 vs. Cocktail) and 0.0024 (MK.R3-15 vs. Cocktail). **f** Calculated lag time (h) from the heat curves for each tested condition. The *p*-values are 0.0051 (FJK.R9-30 vs. Cocktail) and 0.0002 (MK.R3-15 vs. Cocktail). $OD_{600}$ measurement of a seven-day (**g**) and three-day (**h**) co-incubation of planktonic naive Paer09 strain with the cocktail (MOI: 0.001, **g**; MOI: 0.0001, **h**). Thin lines represent the individual replicates (*n* = 8), and the thick coloured line the average, excluding completely suppressed replicates (*n* = 4, **g**; *n* = 2, **h**). **i** Illustration of averaged co-incubation curves (including grey, violet, and yellow curves from Fig. 5d) over three days compared to a growth control (dotted black line) and negative control (black line). The cocktail (MOI: 0.0001) is shown as a dotted green curve. All experiments were performed in fourfold and unless specified otherwise, the calorimetric curves (**d, e**) display the mean. The error bars represent the standard error of the mean. Statistical analysis was conducted by unpaired two-tailed Student's *t* test and *p*-values indicated with asterisks (**p* = <0.05; ***p* = <0.01; ****p* = <0.001). Source data are provided as a source data file.

hydrolase) (with only one altered gene per mutant) (Fig. 6a). The isolates incubated with the cocktail did not show any mutations in the *rmlA* gene, the most frequently mutated protein among all isolates treated with the individual phages (Fig. 6b). A representative subset of isolates, each mutated in only one of the three identified genes (*rmlC*, frameshift; *wapR*, frameshift; *glycoside hydrolase*, frameshift), were selected for further characterisation, and the *glycoside hydrolase* mutant showed no altered growth curve or biofilm cell count change, but reduced biofilm biomass and reduced virulence in *G. mellonella* (Supplementary Fig. S5). It is noteworthy that none of the eighteen isolates developed a simultaneous resistance to both phages of the phage cocktail.

In conclusion, while the phage cocktail does not have a major impact on the short-term (within the initial 24 h) antibiofilm efficacy compared to the individual phage FJK.R9-30, it exhibits a higher suppressive activity at prolonged incubation times, as bacteria cannot develop full resistance to the phage cocktail.

## Discussion

To improve infection management of *P. aeruginosa* biofilms, we developed a directed evolution assay for the combined improvement of the host spectrum, antimicrobial and antibiofilm efficacy, shown for four lytic *P. aeruginosa* phages. These evolved phages reduced the

bacterial capacity to escape predation and presumably caused the eradication of planktonic bacterial cultures. The two-phage cocktail based on a bacterial resistance trade-off further exerted a prolonged suppressive activity, likely owing to the absence of bacterial mutants simultaneously resistant to both phages.

Our findings demonstrate that we could successfully direct the improvement of several infectivity parameters for phages from different genera, leading to an improved antimicrobial performance also against a bacterium not included in the evolutionary assay. Nevertheless, the selective pressure to adapt to pre-established bacterial biofilms within our evolution assay, might have been weakened by the release of planktonic bacteria into the surrounding medium, only avoidable within dynamic biofilm models. As a proof of concept, the results of our study are based on a single directed evolutionary assay, but as previous studies have found, the evolution of phages, given the small genomes and thereby limited evolutionary pathways, demonstrates a great reproducibility down to the codon and nucleotide level[40,54–56]. By contrast, Esvelt et al. and Wichman et al. highlight how parallel mutations can vary in their order of appearance, resulting in different adaptive trajectories[57,58]. In the end, for Esvelt et al. those trajectories converged, stressing the aspect of time (e.g., number of viral generations, rounds) in evolution experiments. In addition, Wichman et al. point out that early mutational changes conferring greater boosts in fitness may not always show in all replicates, which could explain the fact that each mutant had only a single mutation, which might set them on different adaptation pathways[58]. Thus, determining the phage proportions in each round and isolating phages for characterisation, not just in rounds 15 and 30, would have provided more insights into the phages' adaptive pathway within the evolution assay. Investigating synergistic and antagonistic interferences within the evolutionary phage mixture or the two-phage cocktail, would have also provided a deeper understanding of phage-phage interactions and the evolutionary outcomes[59,60].

However, as the road from proof of concept to bedside is long, several hurdles remain. These include further validations of the evolution platform with different phage-bacteria systems, to reiterate the combined improvement of the infectivity parameters. Not limited to Gram-negative bacteria, the evolution assay could also find applicability to improve phage efficacy against Gram-positive bacteria and pathogens relevant in agriculture, aquaculture, and food safety to reduce the number of antibiotics currently employed[61,62]. Underpinning the phage improvement, the efficacy of phage-derived enzymes (depolymerases and lysins) could be enhanced, and the identification of such mutational sites could provide new targets for genetic engineering and enzyme-based therapies[21,22]. At the same time, the platform itself could be further streamlined by optimising the number and length of each evolution cycle, as it is a labour-intensive approach impeding generalised applicability. Potential for optimisation includes the combination of more than two highly similar phages to increase the number of combinatory possibilities or have the phages undergo untargeted mutagenesis prior to selecting for antibiofilm efficacy. Improving the phage infectivity parameters in sequence rather than in parallel could result in varying adaptive trajectories and aid in the identifying of mutational sites of evolutionary adaptation.

Isothermal microcalorimetry allowed us to continuously monitor the phage-bacterial biofilm interaction in real-time with high sensitivity and accuracy[63,64] while qPCR helped us to precisely quantify the antibiofilm degradation capabilities of our phages, providing a starting point for further usage of this technique. As qPCR does not distinguish between live and dead bacteria, the results provided represent a more conservative antibiofilm efficiency, considering the possibility that DNA from dead bacteria not dip-washed away is also quantified. Given the problems concerning the treatment of biofilm infections, trained phages, especially phage cocktails, provide an alternative or adjunct to antibiotic chemotherapy[65,66]. The formulation of such cocktails should be based on combining phages that do not interfere with each other and target distinct bacterial receptors[32,67]. In view of this, our resistance-adapted two-phage cocktail prevented the emergence of bacterial resistance and increased bacterial growth suppression beyond 24 h over the use of the individual phage FJK.R9-30. Along those lines, Yang et al. composed a five-phage *P. aeruginosa* cocktail (10⁹ PFU/ml), comprising two phages that exploit a phage resistance trade-off between the O-antigen and core lipopolysaccharide. Tested against an exponential phase planktonic *P. aeruginosa* PAO1, it took around five days for resistant mutants to arise[68]. In extension to those experiments, our two-phage cocktail at a concentration of approx. $10^3$ PFU/ml (MOI of 0.001) showed a continued suppression of planktonic bacteria (clinical isolate Paer09) up to seven days, while no isolated bacterial mutant had developed a dual-phage-resistance (Fig. 6a, g, i). These results highlight possibilities of reapplying the cocktail in a clinical setting until remission of the infection and further emphasises the importance of a rational phage cocktail design maximising their clinical applicability. Considering clinical applications, biofilm-adapted phages could be included in biobanks and combinations with antibiotics could be assessed, as could the translatability from in vitro to in vivo models[69,70].

Taken together, our evolution platform could provide insights into evolutionary bacteria-phage interactions, defence strategies and interdependencies as well as strengthen phage therapy as a treatment option to improve the outcome of multidrug-resistant bacterial infections with trained resistance-adapted phage cocktails. For personalised medical approaches, phages could be specifically trained against patient strains before administration. All the while, the evolved phages would still be considered as natural, non-genetically engineered entities, limiting the risks if released into the environment and allowing for easier approval[71].

## Methods

The research conducted in this study complies with all relevant ethical regulations. The bacteria for this study were from laboratory strain collections in Belgium, Switzerland, Italy, and Germany. Bacteriophages were isolated from hospital sewage samples in Germany.

### Collection of Pseudomonas aeruginosa strains

A collection of 80 *P. aeruginosa* clinical isolates was used in this study (Source Data D1). *P. aeruginosa* PAO1 was included as a laboratory reference strain. Bacterial isolates were obtained from hospital and laboratory strain collections in Belgium ($n = 41$), Switzerland ($n = 17$), Italy ($n = 6$) and Germany ($n = 17$). Bacterial stocks were prepared in 20% glycerol and stored at −80 °C for further use. An external diagnostic laboratory (Labour Berlin – Charité Vivantes GmbH, Berlin, Germany) conducted bacterial identification using Vitek2-ID (bioMérieux, Marcy-l'Étoile, France) and MALDI-TOF (bioMérieux, Marcy-l'Étoile, France), as well as antibiogram analysis using Vitek2-AST (bioMérieux, Marcy-l'Étoile, France) and MICRONAUT-S (MERLIN Diagnostika, Hersel, Germany), including 3- and 4MRGN (multi-resistance Gram-negative)[72–74] classification. Bacteria were propagated at 37 °C using tryptic soy broth (TSB; 3% w/v; USBiological, Salem, USA), tryptic soy agar (TSA; 3% w/v TSB + 1.5% w/v agar) or tryptic soy soft agar (soft agar; 3% w/v TSB + 0.6% w/v agar).

### Bacteriophage isolation

Phages were isolated from hospital sewage samples collected in Germany, following a standard enrichment procedure as previously described[75], using *P. aeruginosa* strains isolated from the same hospital. The 0.22 μm-filtered enrichment solutions were subsequently spot-tested on soft agar overlays to identify phages. Plaques appearing on the plates were purified by four consecutive single-plaque-passages to ensure phage purity. Next, each isolated phage was produced from a single plaque on their isolation strain using either a liquid[76] or solid[77]

propagation method. Ultimately, phage lysates were concentrated and purified by PEG 8000 precipitation[78] before storage in SM-buffer at 4 °C for further use.

Evolved phages after rounds 15 and 30 of the evolution assay were isolated by spotting 5 µl tenfold serial dilutions in SM-buffer of the corresponding phage mixture on soft agar overlays for each individual bacterial strain in the evolution assay (PAO1, Paer03, Paer09, Paer33, Paer57, Paer60, Paer84 and Paer85). Based on qualitative plaque assessment and host strain we identified 31 phages that were purified and produced on their isolation strains as described above (Supplementary Fig. S3). Of those, 10 (MK.R3-15, MK.R3-30, MK.R57-15, MK.R57-30, MK.R84-15, MK.R84-30, FIM.R60-15, FIM.R60-30, FJK.R9-15 and FJK.R9-30), representing phages descended from the different ancestral phages determined by BLASTn v2.13.0[79], were concentrated and purified by PEG 8000 precipitation[78] before storage in SM-buffer at 4 °C for further use. The name is composed of the ancestral phage name (MK, FIM, and FJK), the isolation strain (e.g., R84 for Paer84), and the number of passages denoted as 15 (isolation after round 15) or 30 (isolation after round 30).

### Bacteriophage transmission electron microscopy

Phage morphology was visualised by transmission electron microscopy (TEM) using negative staining. An aliquot of 15 µl of the phage particle preparation was dropped onto Parafilm prior to the transfer onto a Ni-mesh grid (G2430N; Plano GmbH, Wetzlar, Germany) which has been carbon-coated and glow discharged (Leica MED 020, Leica Microsystems, Wetzlar, Germany). Samples were allowed to adsorb for 10-15 min at room temperature. Grids were washed three times with Aquadest and subsequently treated with 1% aqueous uranyl acetate (SERVA Electrophoresis GmbH, Heidelberg, Germany) for 20 sec for negative staining followed by the removal of excess staining with filter paper before being air-dried. Grids were then imaged by TEM using a Zeiss EM 906 microscope (Carl Zeiss Microscopy Deutschland GmbH, Oberkochen, Germany) at a voltage of 80 kV. For each phage, using ImageJ v1.54g[80] four particles were used to calculate the tail length, tail width, and average capsid size in three axes (Source Data D2).

### Host range analysis

The host range for the ancestral and ten representative evolved phages was determined by soft agar overlay spot assays against the entire collection of *P. aeruginosa* strains.

A bacterial overnight culture was mixed (2.5% v/v) with soft agar, poured over TSA and allowed to dry for 10 min. Phage solutions prepared as tenfold SM-buffer dilutions ($10^{-1}$–$10^{-8}$) in 96-well microplates (PN 353072; Corning Inc., Corning, USA) were spotted (5 µl) on the overlays and incubated overnight at 37 °C. Bacterial strains were considered susceptible to a phage when single phage plaques were visible on any of the dilutions. The experiment was either conducted as two biological replicates with two technical replicates each (ancestral phages) or as three biological replicates (evolved phages).

### Bacterial biofilm formation and imaging

Bacterial biofilms were formed on autoclaved 4 mm sintered porous glass beads (ROBU® Glasfilter-Geräte GmbH, Hattert, Germany) by incubation in a sterile 24-well plate (Corning Inc., Corning, USA). Each bead was individually incubated in a well containing 1 ml TSB inoculated with 1:100 dilution from a one-time use glycerol stock of *P. aeruginosa* and kept at 37 °C and 150 rpm orbital shaking for 24 h under humidity conditions.

For scanning electron microscopy (SEM), glass beads were first dip-washed in phosphate buffered saline (PBS) (Merck KGaA, Darmstadt, Germany) and fixated in a solution of 1% paraformaldehyde and 2.5% glutaraldehyde in 50 mM HEPES for 48 h at room temperature. All samples were subsequently washed in 50 mM HEPES, dehydrated in consecutive steps of 30, 50, 70, 90, 95, 100, and again 100% ethanol,

chemically dried overnight in hexamethyldisilazane (Sigma-Aldrich, Darmstadt, Germany), mounted on aluminum stubs, sputter coated with a 16 nm layer of gold-palladium, and examined in the SEM (ZEISS 1530 Gemini, Carl Zeiss Microscopy Deutschland GmbH, Oberkochen, Germany) operating at 3 kV using the in-lens electron detector. SEM imaging was conducted for one biological replicate of each strain.

### In vitro bacteriophage evolution assay

The in vitro phage evolution assay to improve multiple phage parameters in parallel, consisted of a serial passaging approach with thirty consecutive rounds, inspired by the directed evolution approach of the Appelmans protocol[81]. Adaptations to accommodate bacterial biofilms included the use of microcalorimetric real-time monitoring, revised active sample criteria and performance-dependant phage-mixture-dilutions. For each 24 h round, the undiluted and three tenfold serial dilutions of a mixture of phages were independently co-incubated with 24-h-biofilms of each *P. aeruginosa* strain (PAO1, Paer03, Paer09, Paer33, Paer57, Paer60, Paer84 and Paer85) formed on glass beads. The initial phage mixture comprised equal amounts ($10^6$ PFU/ml) of the ancestral phages (JS, MK, FIM, and FJK) and after each round of evolution, a new mixture was made, combining all active samples at that round. A schematic illustration of the in vitro evolution assay is depicted in Fig. 1.

The criteria for inclusion of the ancestral phages in the evolution assay were (i) a strictly lytic infection cycle and (ii) phage taxonomy. Four phages from the genera *Pakpunavirus* (MK and JS), *Pbunavirus* (FIM), and *Bruynoghevirus* (FJK) were selected. Inclusion criteria for the bacterial strains were (a) susceptibility to at least one of the ancestral phages, (b) genomic diversity, (c) the antibiotic resistance profile, (d) diversity in biofilm formation (SEM images), and (e) non-auto-plaque former. In total, eight *P. aeruginosa* strains were included in the assay and irrespective of the inclusion criteria, PAO1 was included as a laboratory standard strain.

A 48-channel isothermal microcalorimeter (TAM III; TA Instruments, New Castle, USA) was used to monitor, in real-time and with high sensitivity (0.2 µW), the heat flow produced by each sample in each round. The heat flow, proportional to the observed exothermic biological processes, allows for an assessment of microbial metabolism, such as bacterial growth is indicated by an increased heat flow, while the suppression and eradication of these bacteria results in delayed or absent heat production[82–84]. Contained in airtight 4 ml disposable glass vials (Waters GmbH, Eschborn, Germany), each sample comprised 450 µl TSB, one in PBS dip-washed 24-h-biofilm glass bead and 50 µl of the corresponding phage mixture dilution in SM-buffer. Growth controls without phages were included in each round. Sterility controls, also included in each round, contained TSB, either with (1) the undiluted phage mixture, (2) the undiluted phage mixture and a sterile glass bead, or (3) a sterile glass bead.

Active samples were defined based on a reduction in heat (J) of ≥75% compared to the corresponding growth control sample. This threshold strikes a balance between detecting phage activity through bacterial heat production reduction and the exclusion of samples that would reduce phage diversity. During the first fifteen rounds of the evolution assay, the comparative heat reduction analysis was conducted considering the cumulative heat after 24 h, while from round 16 onwards, the cumulative heat from the initial 8 h of the assay was considered (Fig. 1c). Active samples and samples containing the undiluted phage mixture across all bacterial strains were pooled into a single mixture after each round. This pooled mixture was centrifuged at 5,752 x g for 20 min and the supernatant filtered (0.22 µm) before introduction into the next round.

Throughout the evolution assay, the following criteria were applied for each bacterial strain to define which phage-mixture-dilution should be included in the subsequent round:

1. the undiluted phage mixture was kept constant for each round.

2. if at least two of the diluted samples were active – as defined above – all three dilutions were additionally diluted tenfold for the next round.

3. if the three diluted samples were all not active, the dilutions prepared for the next round were diluted one-tenth less.

4. if criteria 2 or 3 did not apply, then the tenfold dilutions were not varied for the next round.

At round 0 and after rounds 4, 9, 14, 19, 24, and 29, the pooled phage mixture was serially diluted tenfold and spotted on soft agar overlays of the eight ancestral bacterial strains of the evolution assay. Phage plaques were enumerated after overnight incubation at 37 °C and concentrations in PFU/ml of the phage mixture were determined for each strain. The test was performed in three biological replicates. The calculated phage concentrations were then correlated with the corresponding dilution factors at the onset of rounds 1, 5, 10, 15, 20, 25, and 30, allowing for a direct comparison of the heat production between samples with the same initial phage concentration in the different evolution rounds (Supplementary Fig. S2).

## Whole genome sequencing and analysis

Total bacterial genomic DNA was extracted using the DNeasy Ultra-Clean Microbial Kit (Qiagen, Hilden, Germany) following the manufacturer's instructions and sequenced as previously described[85]. An Illumina DNA library was prepared using the Nextera Flex Kit (Illumina, San Diego, CA, USA) and sequenced on an Illumina MiniSeq instrument with the MiniSeq High Output Reagent Kit (300 cycles). Additionally, the Rapid Barcoding Kit (Oxford Nanopore Technology, Oxford, UK) was used to prepare the same DNA samples for long-read sequencing on a MinION device using an R9.4.1 flowcell (Oxford Nanopore Technology, Oxford, UK). Guppy v3.1.5 (Oxford Nanopore Technology, Oxford, UK) was used as basecaller. Next, the Unicycler hybrid assembly pipeline v0.4.8.0[86] was performed to assemble the bacterial genomes. The quality of each assembly was visualised with Bandage v0.8.1[87]. Subsequently, the genomes were functionally annotated using Prokka v1.14.6[88]. After determining the core genome (3,667 of 19,556 genes; 99% ≤ strains ≥ 100%; min. pct. identity for BLASTp: 95) using Roary v3.13.0[89], RAxML v8.2.4[90] was used to infer a maximum likelihood phylogenetic tree of the complete *P. aeruginosa* strain collection, which was then visualised with iTOL v6.5[91]. To analyse the phage treated Paer09 strains' genomic data, single nucleotide polymorphisms (SNP), small deletions and insertions (indels) between the ancestral Paer09 genome and each isolated colony were identified using Snippy v4.6.0[92]. The assembled genomes of all biobank *P. aeruginosa* genomes are available under NCBI BioProject PRJNA906522. For the phage treated Paer09 *P. aeruginosa* genomes, deposited under the accession codes listed in Source Data D8, the Illumina sequencing datasets are available in the Sequence Read Archive (SRA) database via the same BioProject [https://www.ncbi.nlm.nih.gov/bioproject/PRJNA906522].

Phage genomes were extracted[93] and the concentration and purity were determined by NanoDrop ND-1000 UV-Vis Spectrophotometer (PEQLAB, Erlangen, Germany). Phage DNA was subsequently sequenced with Illumina as described in Makalatia et al.[94]. Phage genomes were assembled using the SPAdes-based PATRIC genome assembly v3.6.12[95], except for phage FJK, FJK.R9-15, FJK.R9-30, FIM, FIM.R60-15 and FIM.R60-30, which were assembled using Shovill v1.1.0[96]. The most similar reference phages were then retrieved using BLASTn v2.13.0[79]. The genera of the ancestral phages were determined by an intergenomic distance between them and their most similar reference phages above 70%, as determined by VIRIDIC[97]. Genome alignment to these identified phages was performed using MEGA11[98]. Resulting aligned phage genomes were functionally annotated through the RASTtk pipeline and manually curated using the

BLASTp program v2.13.0[99], HHpred[100], HHblits[101], and HMMER v.3.3[102] integrated in MPI Bioinformatics Toolkit[103]. GenBank files of the ancestral phages were finalised using Artemis v18.1.0[104] and deposited under the accession codes listed in Source Data D2. Illumina reads for both the ancestral and evolved phages were submitted in the SRA database and are available under NCBI BioProject PRJNA906522. To visualise and illustrate phage genomes, linear comparison figures were generated using Easyfig v2.2.2[105]. To analyse the phages, single nucleotide polymorphisms (SNP), small deletions and insertions (indels) between the ancestral unevolved phages and each evolved phage were identified using Snippy v4.6.0[92] (Source Data D9). The tertiary structure of FJK_gp62 was predicted using ColabFold[106,107]. Similar tertiary structures were identified with DALI[108]. The related TTPA structure was downloaded from the Protein Data Bank (PDB code 5MU4). Protein structures were visualised and analysed using PyMOL 2.5[109,110].

## Antibiofilm effect determined by quantitative real-time qPCR

A real-time quantitative polymerase chain reaction (qPCR) was used to quantify the number of viable cells following exposure of 24-h-biofilms of three representative *P. aeruginosa* strains: Paer09, Paer57 (both included in the evolution assay) and Paer36 (not included in the evolution assay) to ancestral and evolved phages isolated at round 15 and 30 by adapting a previously described method[111]. In addition, the antibiofilm activity of combining phages FJK.R9-30 and MK.R3-15 was further investigated on Paer09 biofilms. Briefly, 24-h-biofilms were formed on sterile porous glass beads as described above, dip-washed in 2 ml PBS to remove any planktonic cells and transferred into 48-well plates (LABSOLUTE; Th. Geyer GmbH & Co. KG., Renningen, Germany) containing 450 μl of sterile TSB only (for the growth controls) or with additional 50 μl of phages (for treated samples). Plates were subsequently incubated at 37 °C and 150 rpm orbital shaking for 3 h, 6 h or 24 h. After incubation, beads were dip-washed in 2 ml PBS, transferred to an Eppendorf tube containing 200 μl of PBS, and sonicated (BactoSonic14; BANDELIN, Berlin, Germany) at 200 $W_{eff}$ and 40 kHz for 10 min. Next, the sonicated suspension was used for the DNA extraction of the dislodged biofilm bacterial cells, using the DNeasy Ultra-Clean Microbial Kit (QIAGEN, Hilden, Germany). Extracted bacterial DNA was stored at 4 °C for further use.

The NZYTech *Pseudomonas aeruginosa* Real-time PCR Kit targeting the toxin A synthesis regulating gene (RegA) was used according to the manufacturer's instructions (MD02381; NZYTech, Lisboa, Portugal). The extracted DNA was amplified and quantified in the Mastercycler RealPlex[2] (Eppendorf, Hamburg, Germany). In each experiment, as part of the PCR kit, a positive control, negative control, and internal extraction control were included. The experiments were performed as two biological replicates with two technical replicates each. Phage titre (≈1.68 × 10[7] PFU/ml) used for this experiment were determined in biological triplicates on the corresponding bacterial strain, to be tested according to an adapted double agar overlay plaque assay[112].

## Antimicrobial activity testing by isothermal microcalorimetry

Isothermal microcalorimetry was used to compare the antimicrobial activity of ancestral and evolved phages isolated at rounds 15 and 30 against the biofilm of the strains Paer09, Paer36 and Paer57. In addition, the antimicrobial activity of combining phages FJK.R9-30 and MK.R3-15 was further investigated against Paer09 biofilms. The experiments were performed in two biological replicates with two technical replicates each. Phage titre (≈1.68 × 10[7] PFU/ml) used for this experiment were determined by an adapted double agar overlay plaque assay[112] in biological triplicates on the corresponding bacterial strain to be tested, except for phage MK.R3-15, determined on Paer03 (≈1.68 × 10[7] PFU/ml) instead of Paer09 in the phage cocktail experiments.

24-h-biofilms were formed on porous glass beads as described above, dip-washed in 2 ml PBS to remove any planktonic cells and transferred into 4 ml glass vials containing 450 μl of sterile TSB only (for the growth controls) or with an additional 50 μl of phages (for treated samples). Vials were sealed airtight and immediately placed in the calorimeter, where the heat production was monitored during 48 h for single-phage treatment or during 72 h and 96 h for combined-phage treatment.

### Induction, verification, and characterisation of bacteriophage resistant Paer09

The co-incubation of the Paer09 strain with either phage FJK, phage FJK.R9-15, phage FJK.R9-30, or the combined phages FJK.R9-30 and MK.R3-15 (cocktail) was performed to induce phage resistance and selected upon in continuation of previous experiments and the observed phage-resistance trade-off. This experiment was carried out in two biological replicates with four technical replicates (individual phages) or in eight biological replicates (cocktail). Phage titre used in this assay was determined in three biological replicates on the corresponding host strain (Paer09 for phage FJK, FJK.R9-15 and FJK.R9-30 and Paer03 for phage MK.R3-15).

Paer09 was grown in TSB at 37 °C for 24 h and adjusted to approx. $10^7$ CFU/ml (determined in biological triplicates) by dilution in fresh sterile TSB. Then, 160 μl of sterile TSB and 20 μl of bacteria were transferred into a transparent, flat-bottom, 96-well microplate (Corning Inc., Corning, USA). As growth controls, 20 μl of SM buffer was added. For the treated samples 20 μl of phages were applied at a multiplicity of infection (MOI) of 0.001 (approx. $10^3$ PFU/ml) and, in case of the phage cocktail, an additional sample with an MOI of 0.0001 (approx. $10^2$ PFU/ml) was added. The microplate was incubated at 37 °C and 150 rpm orbital shaking for 72 h under $OD_{600}$ monitoring (BioTek Epoch 2NSC, Winooski, USA) at 10 min intervals. The cocktail samples at an MOI of 0.001 were incubated for 168 h. The solution from each well was then centrifuged at 9,391 x g for 1 min and washed three times in PBS, before being plated on TSA. After overnight incubation at 37 °C, based on the replicate ($n = 32$) and distinct morphological appearance, 66 colonies (one replicate, no colonies; one replicate, one colony; twenty-five replicates, two colonies; five replicates, three colonies; from here on referred as picked-colonies) were picked and re-plated two more times, before being stored as 20% glycerol stocks at −80 °C.

Phage susceptibility was evaluated for each picked-colony (individual biological replicate) and the ancestral Paer09 strain (biological duplicate) by spotting tenfold serial dilutions of the phages (FJK, FJK.R9-15, FJK.R9-30, MK, and MK.R3-15) on soft agar overlays. Phage plaque enumeration was performed after overnight incubation at 37 °C and concentrations were determined as PFU/ml. The relative efficiency of plating (EOP) was defined as the ratio of the phage titer on the picked-colony (numerator) and the phage titer on the naive Paer09 strain (denominator) (Source Data D8)[113]. An EOP above 10 was considered as increased efficiency, while an EOP of 0.1 to 10 was ranked as unchanged efficiency. A reduced efficiency was defined as an EOP between 0.001 and 0.1. When no individual plaques were visible or the EOP was equal to or under 0.001 the isolate was determined resistant to the phage[114]. As phage MK was not active on the naive Paer09 strain, a theoretical value of a single plaque in the undiluted spot ($2 \times 10^3$ PFU/ml) was used in the denominator.

From the 66 picked-colonies a subset of seven representative bacterial mutants, each mutated in only one of the seven identified genes (*rmlA*, nonsense; *rmlB*, missense; *rmlC*, frameshift; *rmlD*, nonsense; *wapR*, frameshift; *pslA*, nonsense; *glycoside hydrolase*, frameshift), were selected for further characterisation. Growth curves were prepared in three biological replicates, with three technical replicates each, using 96-well microplates (PN 655198; Greiner Bio-One,

Kremsmünster, Austria) with a starting bacterial concentration of $1 \times 10^6$ CFU/ml in TSB at 37 °C. The $OD_{600}$ measurement was taken at 1 h intervals for 24 h after 20 sec orbital shaking at 200 rpm. To test the virulence in *G. mellonella*, bacteria were grown overnight at 37 °C, spun down at 4000 x g for 10 min, resuspended in 1 ml of PBS, spun down again (4000 x g, 10 min), and resuspended in 4 ml of PBS. Larvae were injected with 10 μl of bacteria ($10^3$ CFU/ml) ($n = 10$ per strain) or PBS ($n = 10$), as a control, into their hindmost left proleg. Following the injection, larvae were incubated at 37 °C in individual wells of a 12-well plate (PN 665180; Greiner Bio-One, Kremsmünster, Austria) for 100 h. Their activity, melanisation, and survival were monitored every 5 h following the health index scoring system[115]. The biofilm cell count determination was conducted in three biological replicates with two technical replicates each. For each bacterium, 10 ml of TSB ($10^7$ CFU/ml) were added to a 50 ml tube containing 10 porous glass beads. After static incubation for 24 h at 37 °C, the tube was gently washed three times with 10 ml of PBS. Each bead was added to a 1 ml tube containing PBS before being vortexed (30 sec), sonicated (60 sec, 120 W, 47 kHz; Branson 2210E-MT Ultrasonic Cleaner; Branson Ultrasonics Corp., Brookfield, USA) and vortexed (30 sec) again, as previously described[116]. Tenfold serial dilutions of bacteria were spotted on TSA plates and enumerated after overnight incubation at 37 °C. Before crystal violet staining was performed, 100 μl of each bacterium ($10^6$ CFU/ml) was statically incubated for 24 h at 37 °C in separate wells of a 96-well microplate (Greiner Bio-One, Kremsmünster, Austria). After removal of the liquid, each well was washed with 125 μl of PBS. Then, 125 μl of crystal violet solution (0.1% w/v in Milli-Q water) were added to each well and allowed to stain the biofilms for 15 min at room temperature. After removal of the crystal violet solution, each well was washed 2 times with PBS and allowed to air-dry. Next, 200 μl of 95% ethanol was added to each stained well. Dye was allowed to solubilise for 15 min at room temperature, before being mixed by pipetting up and down. 125 μl of the crystal violet/ethanol solution from each well were transferred to a new clear flat-bottom 96-well microplate (Greiner Bio-One, Kremsmünster, Austria). The optical density (OD) was measured at 570 nm. The experiment was performed as three biological replicates with three technical replicates each. The swarming and swimming motility were determined for each mutant by measuring the bacterial radial growth diameter after overnight incubation at 37 °C. The experiments were performed in four biological replicates on TSA plates (swarming, 3% w/v TSB + 0.5% w/v agar; swimming, 3% w/v TSB + 0.3% w/v agar). For the complementation assay, the wild-type genes (*rmlA*; *rmlB*; *rmlC*; *rmlD*; *wapR*; *pslA*; *glycoside hydrolase*) were cloned into the *Pseudomonas* inducible expression vector pHERD20T (PN V005568; NovoPro Bioscience Inc., Shanghai, China) and transformed to the respective mutants (complemented mutants). Controls were transformed with empty plasmids (non-complemented mutants) and induction for both mutants was carried out with a final concentration of 0.2% arabinose, while a final concentration of 200 μg/ml carbenicillin was added to select for the plasmid-carrying strains, confirmed by colony PCR and Sanger sequencing. Phage susceptibility was evaluated for each complemented and non-complemented mutant in biological triplicates by spotting tenfold serial dilutions of the phages (FJK, FJK.R9-15, FJK.R9-30, MK, and MK.R3-15) in SM-buffer on soft agar overlays. Phage plaque enumeration was performed after overnight incubation at 37 °C and concentrations were determined as PFU/ml. The relative efficacy of plating (EOP) was defined as the concentration ratio of a phage on the complemented mutant (numerator) and the non-complemented mutant (denominator) (Source Data D10). EOP values were grouped as described above, except if no individual plaques were visible or the EOP was equal to or under 0.001 it was considered greatly reduced. As phages FJK, FJK-R9-15 and FJK-R9-30 were not always active on the non-complemented mutant, a theoretical value of a single plaque in the undiluted spot ($2 \times 10^2$ PFU/ml) was used in the denominator.

**Visualisation, statistics and reproducibility**

Figures 1 and S3 were created with BioRender (BioRender.com). The graphical illustration of the heat, heat flow and optical density graphs was prepared using GraphPad Prism 9 (GraphPad Software, San Diego, USA). For the heat and heat flow curves of the antimicrobial activity testing the experimental replicates were interpolated to a 36 sec equidistant timeline and graphed as mean. The heat curves of the in vitro bacteriophage evolution assay were used directly for further calculations. By calculating the first derivative of each replicate heat curve (between each point) and the averaged optical density curves (between every second point), the maximum slope and its corresponding tangent were identified. The x-axis value of the intersection point between the baseline and the tangent represents the duration (h) of the lag time[117].

qPCR, heat, and heat flow results were statistically analysed using an unpaired two-tailed Student's *t* test analysis integrated in GraphPad Prism 9. The Spearman's rank correlation coefficient was calculated with all 66 bacterial isolates using GraphPad Prism 9 and reported as $r_s$ with corresponding *p*-value. No statistical method was used to predetermine sample size. Instead, sample sizes were selected based on inclusion criteria, previous similar studies in the field, the specific objectives of each experiment and practical considerations. No data were excluded from the analyses.

**Reporting summary**

Further information on research design is available in the Nature Portfolio Reporting Summary linked to this article.

## Data availability

The genomic data generated and analysed during the current study is available under the accession codes listed in Source Data D1, D2, D8 in the NCBI BioProject PRJNA906522. Source data are provided with this paper.

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

## Acknowledgements

We thank the Berliner Wasserbetriebe for their support in the collection of hospital sewage. We thank Dr. Fintan Moriarty and Dr. Virginia Post from the AO Research Institute Davos, Prof. Dr. Willem-Jan Metsemakers and Dr. Jolien Onsea from University Hospital Leuven, and Asst. Prof. Dr. Mariagrazia Di Luca from the University of Pisa for providing *Pseudomonas aeruginosa* clinical isolates for this study. From the Laboratory of Gene Technology at KU Leuven we thank Alison Kerremans for her technical support, Leena Putzeys and Cedric Lood for the sequencing of the Belgien bacterial collection, and Dominique Holtappels for his guidance on the annotation of phage genomes. We also want to thank Petra Schrade of the Core Facility for Electron Microscopy at the Charité – Universitätsmedizin Berlin, for the help in the collection of scanning- and transmission-electron microscopy images. We thank the Labour Berlin – Charité Vivantes GmbH for the support in the identification and antibiogram determination of our collection of *Pseudomonas aeruginosa* bacterial strains. We thank Mrs. MPH Pimrapat Gebert from the Institut für Biometrie und Klinische Epidemiologie at the Charité – Universitätsmedizin Berlin and Mrs. Sara Gottlieb-Cohen and Mr. Parker Holzer from the StatLab at Yale University for their guidance on the statistical analysis. From Yale University we also thank Elizabeth Sylander for her organisation of the international bacterial and phage transport. This work was funded as part of the JPIAMR: Cross-border research project ANTIBIO-LAB (BMBF/DLR Grant number: 01KI1823). RL & JW are supported by KU Leuven, Internal Funds KU Leuven, Interdisciplinary Networks (ID-N) grant (IDN/20/024). For the realisation of his doctorate studies FK received scholarships from the German Academic Scholarship Foundation (Studienstiftung des deutschen Volkes), the German Society of Internal Medicine (DGIM), the German Society for Orthopaedics and Orthopaedic Surgery (DGOOC) and the Sonnenfeld Foundation. His research at Yale University was supported by the Heinrich Hertz-Stiftung of the state North Rhine-Westphalia (NRW). We would like to thank all our sponsors for their support.

## Author contributions

Based on the Contributor Roles Taxonomy by CRediT that are highlighted in parentheses. Within each role, the contributors are ordered alphabetically. (Conceptualisation) A.T., M.G.M., and R.L., conceptualised the overarching research goals and aims of the project. (Data curation) F.K. and M.G.M. managed the data curation. (Formal analysis) F.K. conducted the formal analysis of the data and was supported by J.W. for the analysis of the genomic data. (Funding acquisition) A.T., M.G.M., and R.L. acquired the overall funding for the project, while F.K. and M.J.R. acquired the financial support for the research stay at Yale University, New Haven, USA. (Investigation) A.T., M.G.M., and R.L. collected the Pseudomonas aeruginosa strains. F.K. isolated the ancestral and evolved bacteriophages. C.S. was responsible for the bacteriophage transmission electron microscopy. F.K. and S.Y. jointly performed the host range analysis. C.S. performed the bacterial biofilm imaging. F.K. carried out the in vitro bacteriophage evolution assay. J.W. conducted the whole genome sequencing for the bacterial strains and bacteriophages. S.Y. determined the antibiofilm effect by quantitative real-time qPCR. F.K., M.G.M., and S.Y. performed the antimicrobial activity testing by isothermal microcalorimetry. F.K. conducted the induction, verification, and characterisation of bacteriophage-resistant Paer09. C.C. performed the mutant characterisation and the complementation experiment. (Methodology) F.K. and M.G.M. developed and designed the in vitro bacteriophage evolution assay and designed the host range analysis. F.K. and S.Y. developed and designed the determination of the antibiofilm effect by quantitative real-time qPCR. F.K. and M.G.M. designed the antimicrobial activity testing by isothermal microcalorimetry. B.K.C., and F.K. designed the induction, verification, and characterisation of bacteriophage resistant Paer09. C.C. and F.K. devised the mutant characterisation and complementation assay. (Project administration) F.K. and M.G.M. managed and coordinated the research activity of the project. (Resources) A.T. provided the research resources at Charité – Universitätsmedizin Berlin, Berlin, Germany, P.E.T. at Yale University, New Haven, USA and R.L. at KU Leuven, Leuven, Belgium. (Supervision) B.K.C., M.J.R., P.E.T., and R.L. provided oversight and leadership for the research activity planning and execution, as well as mentorship. (Validation and Visualisation) The validation and visualisation of the data were carried out by F.K. (Writing – original draft) F.K. wrote the original draft. (Writing – review & editing) F.K., J.W., M.G.M., P.E.T., and R.L. provided critical review, commentary, and revision advice.

## Funding

## Competing interests

P.E.T. declares a conflict of interest as cofounder of Felix Biotechnology, Inc., a company that seeks to develop phages for human therapy. The other authors declare no competing interests.

## Additional information

[1]Faculty of Medicine, Universität Münster, Münster, Germany. [2]Center for Musculoskeletal Surgery, Charité – Universitätsmedizin Berlin, Corporate Member of Freie Universität Berlin and Humboldt-Universität zu Berlin, Berlin, Germany. [3]Department of Ecology and Evolutionary Biology, Yale University, New Haven, CT, USA. [4]Center for Phage Biology and Therapy, Yale University, New Haven, CT, USA. [5]Department of Biosystems, KU Leuven, Leuven, Belgium.

⁶Department of Biology, Università di Pisa, Pisa, Italy. ⁷Berlin Institute of Health at Charité – Universitätsmedizin Berlin, BIH Center for Regenerative Therapies (BCRT), Berlin, Germany. ⁸Advanced Light and Electron Microscopy (Zentrum für Biologische Gefahren und Spezielle Pathogene 4), Robert Koch Institute, Berlin, Germany. ⁹Program in Microbiology, Yale School of Medicine, New Haven, CT, USA. ¹⁰Department of Trauma, Hand and Reconstructive Surgery, Universitätsklinikum Münster, Münster, Germany. ✉e-mail: andrej.trampuz@qut.edu.au

