## [Peer Review File · Nature Communications]

REVIEWER COMMENTS

Reviewer #1 (Remarks to the Author):

This manuscript describes a new platform to improve phage-bacteria interactions using an adapted phage training protocols on biofilms. The authors put in perspective their results by suggesting way how to combine phage in a cocktail to suppress the development of resistance against phages. The study is very well designed, the platform innovative (Calorimetric assay), the techniques (Sequencing, RT-PCR assay for viable counts) used up-to-date. The manuscript is well written.

Here are some comments and suggestion that I hope will improve the manuscript.

1. I was wondering why Phage JS did not show any adaptive potential. Are there reasons for that? The authors did not mentioned why they dropped it for further experiments. The same way, why is phage MK more prone to adaptation, as the authors choose 3 mutants for further experiments (only one each for FIM and FJK)
2. Line 152: I suggest to put a Flow chart in supplemental to have a graphic view of the selection process.
3. Are there differences in the capacity to drive phage evolution across the bacterial clusters (selection of the host for replication for the evolution assays?)
4. Are the bacterial mutants resistant to phage affected in their virulence? Evidence in the literature show that some bacterial mutants resistant to phage loose some virulence (e.g. those lacking of LPS)
5. Biofilms are characterized by bacterial populations with low metabolic rate. I would suggest the authors to investigate whether the evolved phages are active against stationary phase bacteria in planktonic cultures.

Minor comments

1. Abstract is missing the goal of the study
2. Line 17-18: please rephrase, it is not clear. Does the authors mean by “under” and “determined” “as assessed by?”
3. Line 206 – Do the author mean “FJK.R9-30 achieve a heat flow production of 11.6mcW at the maximum of suppression? (i.e. before resistance occurs?). I would rephrase it
4. Line 285, which two?
5. Characters in figures are difficult to read (too small)

Reviewer #2 (Remarks to the Author):

In this manuscript, the authors describe an in vitro evolution assay to improve the efficacy of four phages

against a panel of 8 *Pseudomonas aeruginosa* strains. The four phages were newly isolated from sewage samples and were briefly characterized by genome sequencing and electron microscopy to classify them. A subset of evolved phages obtained through a novel phage evolution platform was then selected and tested against a panel of 80 *P. aeruginosa* strains and some of the evolved phages were able to infect additional strains but also no longer to infect a few others. The selected evolved phages were characterized by genome analysis as well as antimicrobial/antibiofilm properties and were shown to be more efficient (reduced cell counts and longer lag time for resistant strains to emerge), possibly due to specific viral mutations. Phage-resistant bacterial strains were also characterized, and several mutations were predicted, which may be involved in the phage resistance phenotype. Finally, two evolved phages were used in a cocktail and were shown to be highly efficient compared to single evolved phages in eradicating *Pseudomonas*.

First and foremost, I certainly appreciate the remarkable amount of work presented in this manuscript. Quite impressive. However, it also unfortunately raised a significant number of questions for a manuscript submitted to Nature Communications.

Major comments

1. The authors are presenting a new labor-intensive in vitro platform to evolve phages (Figure 1). Yet, this platform appears to have been tested only once with four phages (simultaneously) to generate evolved phages. Perhaps I missed the other trials, if so, data lack standard errors. This needs to be clarified. If a new platform is proposed, one would have expected a few more trials. Also, if the platform was inspired from the Applemans protocol, maybe the differences could be highlighted in the Materials and Methods section.
2. The authors claim that the in vitro platform evolved phages to expand their host range. While true, it was also modest. In Figure 2, 41 strains out of 80 (51%) were sensitive to at least one of the ancestral phages. After 30 rounds of replication, the evolved phages could infect 8 additional strains (+10%) but three strains (-4%) could no longer be infected. Therefore, at the end, 45 strains (out of 80) were sensitive to one of the phages. While this increased host range is noticeable, this seems a lot of work to modestly increase the host range. Perhaps this could be discussed.
3. The authors also argued that the evolved phages have an improved antibiofilm and antimicrobial activities (Figure 3). Yet, the cell counts were reduced by only two logs (from 10^8 to 10^6 CFU/beads) at best. Again, while clearly better, it is still modest in terms of phage efficacy. The emergence of phage resistant strains was indeed delayed with the evolved phages but still, those mutants could still be obtained readily. Again, this seems like a modest gain.
4. Understanding how phages and bacteria evolved is essential to improve phage therapy. The authors did a remarkable job in isolating and characterizing evolved phages and bacteria. Yet, the bacterial part is incomplete (Figure 5). The authors could not identify mutations in a few of the resistant strains, which raised questions on how the bacterial genomes were analyzed. For one, it is unclear if complete genomes were obtained. For most of the resistant strains, mutations were observed, but they may be due to sequencing errors as they were surprisingly not confirmed by PCR+Sanger sequencing. More

importantly their involvement in phage resistance was not confirmed by reintroducing the wild-type gene in the resistant clones to restore the phage sensitivity profile. As of now, the roles of the observed bacterial mutations in the phage resistance are still unclear. Similarly, a number of mutations were also observed in the evolved phages but it is difficult to determine why these phage mutants were able to thrive on specific hosts (Figure 4). Therefore, we do not know more about the evolutionary genetics of interactions between evolving phages and bacteria as mentioned by the authors in the introduction.

5. Finally, the authors selected two evolved phages and thoroughly tested their efficacy in a cocktail. Yet, they did not compare the cocktail containing the two evolved with a cocktail contain one of the wild-type phage (FJK) which can replicate on the naïve Paer09 strain (Figures 2 and 6).

The authors have a fantastic dataset, but it may not have been presented in an optimal way. Most importantly, it may not have been fully analyzed.

Minor Comments

1. Line 10, I suggest replacing “natural predators of bacteria” by “the viruses of bacteria”
2. Line 21, As it is submitted to Nature Communications, I would have expected that other bacterial pathogens would have been tried.
3. Lines 79-80, “... to identify different bacterial or viral genomic parameters”.
4. Line 90, the name of the four phages should be indicated here.
5. Line 9, the name of the eight strains should be indicated here.
6. Line 94, were the phages first amplified on the host used to isolate them (data S2)? Should be indicated. Would their amplification on another host would have changed the evolutionary pattern?
7. Line 94, at the beginning of the experiment, was the EOP of the four phages close to 1 on the various strains? Perhaps a table of the EOPs could be provided in addition to Figure E1A.
8. Line 100, out of curiosity, any reason why the evolution experiment was stopped after 30 rounds?
9. Line 102, the data for Paer33 should be presented in the Extended Data Figure E2.
10. Line 110, Presumable it was also the case with Paer33 but it was discarded. Should be indicated.
11. Line 115, time and phage counts are shown where in the manuscript? No standard errors are provided.
12. Line 116, round 3? You mentioned on line 100 that the evolution assays were round 1, 5, 10, 15, 20, 25, and 30?
13. Line 131, five ampules per strain, could you clarify what these five ampules contained? 6 phage titers appear to have been tested in Figure E2? Were these assays performed at least in duplicate? No standard errors are provided in Figure E2?
14. Line 152, these 31 plaques were recovered and amplified on which *Pseudomonas* host(s)?
15. Line 154, why pick only 31 plaques? More importantly, why focus on only 10 of them? What happened to the 21 others? Seems like a missed opportunity to analyze more, by picking many plaques on different hosts for example.
16. Line 157: “Representatives are unique phages descended from all four ancestral phages? “It seems that no phages were recovered that are derived from JS.
17. Line 158: Seem surprising that those data are not shown at least in supplementary data.
18. Line 158-161: I do not understand this sentence, seems out of place. Do you mean that the strain

Paer36 was selected here because it was resistant to the four ancestral phages? Were the ten selected plaques able to infect these 3 strains?

19. Line 171, should the percentage of increase be compared to the ancestral phages too?

20. Line 178, should indicate that the tree was made using the core genes (how many genes?). The tree, while useful still makes this figure difficult to analyze. It would have been much easier for the readers to group the sensitive strains according to their phage sensitivity profiles. As presented, it puts an emphasis on the tree, while the manuscript is more about phage sensitivity. The four clusters should be kept though.

21. Line 182, MRGN should be defined here.

22. Line 192, where are these evolution strains in the Figure?

23. Figure 2ab, you should indicate the number of strains that are infected by each phage.

24. Figure 2d, can't understand this Figure-.

25. Line 203, you are referring to which Figure here? 2? If so, yes but still limited.

26. Line 205, supplementary table S1, how many times were these assays performed? No standards errors were provided.

27. Line 205, it should be noted that only three strains were tested here (Table S1).

28. Line 209, indicate the Pseudomonas strain for phage MK here.

29. Line 221, should it be 74.8% instead of 74.9%? (see Supplementary Table S2)

30. Lines 217-229, while the various percentages appear interesting, the improvement is still less than 2-log cell count reduction (see Supplementary Table S2 for the CFU counts). This is rather modest. On the positive side, the lag time before the emergence of resistant cells is notable (see Supplementary Table S1 for lag time). It would be of interest to know if these resistant cells can grow at the same level as their ancestral strains.

31. Line 259, indicate the number of bp deleted in the early gene region as well as the number of genes? Any predicted functions in the deleted genes/proteins?

32. Line 261 & 262, what are the gp number of these two enzymes?

33. Line 263, "remarkable" as compared to what?

34. Line 272, you indicated above that four structural genes were mutated but you discussed only two here (gp52 and gp62). What about the other two (gp51, gp58)?

35. Line 282, what would those mutations do? Change specificity? Improve efficiency?

36. Line 285, how was it determined that the evolved phages derived from the ancestral phages MK and JS appear to be a recombination of the two?

37. Line 307, should that predicted conformational change (line 271-272) be shown here?

38. Line 314-315, why have you selected those phage-host pairs for this assay?

39. Line 319, what were those trade-off events observed in preliminary tests (data not shown)? Was it the varying degree of phage resistance?

40. Line 335, I found this very surprising that only one mutation was observed per bacterial strain, especially considering the typical size of a *P. aeruginosa* genome (>6 Mb). Did you completely sequence the 48 genomes? What was the coverage? Did you get only one contig per strain? What about plasmids?

41. Line 337, as indicated above (major point) to really confirmed that the mutations were indeed involved in the phage resistance phenotype, additional genetic experiments should have been performed (ie introduce the wild-type gene into the resistant clone and see if the phage sensitivity profile is restored). Also, were these mutations confirmed by PCR and Sanger sequencing directly on the respective mutants?

42. Lines 362 & 379, how do you explain the absence of no apparent mutation and the resistance phenotype? It raises some question regarding the sequencing and/or the analyses.
43. Lines 392-397 & 465, I think that a better comparison would be with a cocktail of the two ancestral phages (or at least FJK which is capable of infecting with the naive Paer09 strain). Was this tested? That a cocktail of phages performed better than a single phage is not that surprising. But are the evolved phages performing better than the ancestral phage?
44. Line 411, how many isolates were retrieved? 18 (line 415)?
45. Lines 500-526, in my opinion, this section was a bit wordy and perhaps could be reduced in size and should focus on the next steps to improve the proposed platform rather than the clinical aspects. Would it worked with other phages, as one of the phages (JS) doesn't seem to have evolved that much? It is also unclear if it would be applicable to other bacteria (Gram-positive for example).
46. Line 549, is it the first time that these four phages are described?
47. Line 562, still unclear how these 31 plaques were isolated, and the host used. You mentioned plaque morphology as a selection criterium, did the selected phages had a different plaque size compared to the ancestral phages?
48. Line 571, I didn't see any TEM figure in the manuscript. Morphotypes are mentioned in Data S2. Capsid and tail sizes should at least be mentioned in Data S2.
49. Line 591, phages were diluted in which buffer?
50. Line 601, how many beads per well?
51. Line 607, the SEM data were not discussed in the manuscript.
52. Line 626, which host was used to amplify the ancestral phages?
53. Line 632, how was the phage genera determined?
54. Line 679, how many strains were sequenced? Did you assemble each of them into one contig? I am asking because you could not identify mutation in some of the phage-resistant derivative. Is there any plasmids in these strains?
55. Line 705, how were the ends of the phage genomes determined?
56. Line 711, GenBank
57. Supplementary Data S2, not sure I understand what the percentage of coverage is and what the percentage of sequence means here. Should they be 100%?
58. The reference section will need some work. Hard to believe that the authors really looked at that section...
59. Refs 3, 6, 7, 14, 18, 19, 20, 23, 25, 29, 30, 33, 39, 40, 43, 44, 45, 46, 51, 52, 54, 55, 59, 62, 75, 86, 89, 94, 102: Remove capital letters in the titles.
60. Refs 5, 6, 7, 11, 13, 14, 15, 17, 18, 19, 21, 22, 24, 25, 26, 35, 36, 37, 40, 42, 48, 49, 50, 51, 52, 53, 55, 60, 62, 63, 70, 105: Bacterial name should be in italic.
61. Refs 5, 14, 15, 17, 19, 22, 26, 31, 35, 41, 44, 45, 48, 53, 54, 58, 59, 62, 65, 69, 70, 74, 83, 87, 91, 103, 104: Journal name is abbreviated while not in the others.
62. Refs 32, 76: Title is missing.
63. Ref 78: This has been published, see PMID: 28594827
64. Ref 83: W259
65. Refs 23, 29, 33, 39, 52, 86: page numbers or enumbers are missing.
66. Ref 97: This has been published, see PMID: 35637307
67. Ref 100: Title and Year are missing.

Reviewer #3 (Remarks to the Author):

This study set up an antibiofilm evolution assay to train the phages and obtained evolved phages showing an expanded host spectrum, and improved antimicrobial efficacy. This is an interesting and comprehensive study.

However, it is not a conceptual advance in this field as the evolution method is not novel, the overall results are not surprising to me as these mutations (FIG4) are quite common in phage-host coevolution experiments but the molecular mechanisms are not investigated, so it is descriptive. Most importantly, the final phage cocktail seems not to have a major impact on the antibiofilm efficacy.

Concerns:

1)The rationale for using an isothermal microcalorimeter to monitor the evolution process is not clearly described, and this is not a commonly used approach in monitoring phage evolution. What's the correlation of heat flow with the anti-biofilm efficacy? There are various methods to monitor biofilm, but why only use an isothermal microcalorimeter or QPCR to monitor it? since these methods are not clearly described and the controls are not well established, I think the results might not be solid.

2)The setup of this evolution experiment is not clearly described and is very similar to The Appelmans Protocol. But I am not convinced that using beads could select for phage with enhanced anti-biofilm ability. Is it better than the Appelmans Protocol??

3)The mutation of the phage structural proteins is quite common in other phage-host co-evolution experiments, including the tail fiber gene, and polysaccharide depolymerase gene. However, the function and the mechanisms of these mutations are not validated, so it is descriptive.

4)Most importantly, from FIG6C, seems like this evolved phage could not significantly inhibit biofilm, and it seems not likely to enhance the antibiofilm activity in phage therapy. Because in your evolution experiment, phages might infect the released PA strains and not only survive by infecting the bacteria in the biofilm. So, the results seem not surprising to the phage filed.

5)The antimicrobial and antibiofilm efficacy of the phage and evolved phages should be tested using an animal model to complement the in vitro experiments.

Reviewer #4 (Remarks to the Author):

This manuscripts explores the possibility of directing phage evolution towards more effective infective processes against planktonic and biofilm forming strains of *Pseudomonas aeruginosa*. It uses experimental evolution of a mixture of characterized phages against 8 different strains of *P. aeruginosa*. The authors find that there is an overall gain in infectivity, that initial mutations among evolved phages

are mainly associated to structural genes, and that bacterial phage resistance evolution can also concomitantly lead to trade-offs that maximize infectivity in the long-term and minimize resistance evolution. The findings are interesting and relevant, and well analyzed and presented. I only have a major criticism associated to the choice of phage mixture for experimental evolution. I explain more in detail below:

- The authors used a mix of four phages for the directed evolution and not have control treatments with the individual phages against the strains used. This is in principle fine, and it may also have been a choice of feasibility which is very important and valid. My main concern is that there is no tracking implemented during the rounds of evolution of the proportion of the phages at the end of each round, and hence each round/strain/replicate might have had different evolutionary outcomes and selection processes due to these differences. Not having information about the proportions of the phages throughout makes it unclear which evolutionary path is being selected and how this influences the final outcome. This might be the reason why there is not much information about the evolved lines of phage JS or FIM.
- It is also unclear if the authors have information about potential interference between the phages against the different strains or if there are synergistic effects between the phages.
- Overall, not knowing how the proportion of phages changes over time and how the phages interfere or interact with each other make it hard to assess the full potential of the method/system and the phages individually. Perhaps evolved lines of phage JS could have been more effective than any of the others derived from the mixtures. Without those additional treatments/controls it is impossible to determine. I understand that it would have been a massive amount of work. Perhaps the authors can acknowledge those limitations more thoroughly in the discussion and/or in the beginning of the methods.

Minor comments:

- I was not familiar with heat suppression as a method to determine growth. I would appreciate a description of the principle or the justification for its use in the methods, or through the results. I am aware the authors mention a paper where the whole setup is described, but that is not enough.
- Related to above, it is unclear why a 75% heat reduction is the threshold selected by the authors. More clarity about their methods, their measures and what those represent would be helpful.
- Is plaque morphology sufficient to differentiate between and among the ancestral and evolved phages? It would be helpful to have either a supplementary figure or a main figure showing the differences in plaques between the phages. Also, is there any knowledge of the likelihood of plaque morphology to change during experimental evolution?
- In Figure 2 the colors of the clusters in the tree are not really easy to visualize, perhaps thicker lines would be better
- The data shown in figure 2 is very interesting to me. I would appreciate some more information or discussion. Were the losses and gains in infectivity genetically related, i.e., do the gains in infectivity occur in clusters genetically different from those it had or lost infectivity before? Also, are the changes in infectivity or number of strains each phage is able to infect, significant?
- Line 324-325, this sentence needs revision.
- It is unclear why the authors do optical density in figure 5 and 6 instead of the previous calorimetric method. Perhaps a line or two clarifying why this distinction matters at this point would be helpful.

Point-by-point response

to the reviewers' comments on the manuscript entitled "Targeting MDR *Pseudomonas aeruginosa* biofilm with an evolutionary trained bacteriophage cocktail exploiting phage resistance trade-offs" (NCOMMS-23-23549A).

We would like to thank the reviewers and editor for their valuable scientific comments and constructive experimental suggestions to further strengthen our study. In the revised manuscript, taking all reviewers' comments into consideration, we have made significant clarifying revisions and added all necessary experiments as suggested.

For each line reference the corresponding manuscript section is abbreviated as (a.) (abstract), (i.) (introduction), (r.) (results), (m.) (methods), (d.) (discussion) and (r.) (references).

Reviewer #1

This manuscript describes a new platform to improve phage-bacteria interactions using an adapted phage training protocols on biofilms. The authors put in perspective their results by suggesting way how to combine phage in a cocktail to suppress the development of resistance against phages. The study is very well designed, the platform innovative (Calorimetric assay), the techniques (Sequencing, RT-PCR assay for viable counts) used up-to-date. The manuscript is well written.

Here are some comments and suggestion that I hope will improve the manuscript.

[Major comments]

1. I was wondering why Phage JS did not show any adaptive potential. Are there reasons for that? The authors did not mentioned why they dropped it for further experiments. The same way, why is phage MK more prone to adaptation, as the authors choose 3 mutants for further experiments (only one each for FIM and FJK)

Response to comment: Our apologies that this was not made clear. In reference to Supplementary Data S2 included in the file “NCOMMS-23-23549A_DBPR_Supplementary_Information” the evolved MK bacteriophages (phages) show a high degree of sequence identity (BLASTn) to both ancestral phages JS and MK, which show 96.24% sequence identity between both. With one exception, the similarity of the evolved phages is highest to the ancestral phage MK. Although phage MK.R3-15 has a higher sequence identity with phage JS (98.02%) than with MK (97.53%), it was also named after MK for an easier readability of the manuscript. From the phage genome analysis, it appears that the evolved phages contain genes from both MK and JS and are therefore a recombination of these two. To this end, we decided to use three of their evolved phages.

We have further revised the manuscript accordingly.

Line 297ff. (r): Evolved phages derived from the phages MK and JS (MK.R3-15, MK.R3-30, MK.R84-15, MK.R84-30, MK.R57-15, MK.R57-30) appear to be a recombination of these two ancestral phages.

2. Line 152: I suggest to put a Flow chart in supplemental to have a graphic view of the selection process.

Response to comment: We agree, this would be useful. Figure E3 included in the file “NCOMMS-23-23549A_DBPR_Extended_Data” represents the isolation process of the evolved phages from the phage mixtures of round 15 and 30 of the phage evolution.

We have revised the phrase accordingly.

Line 155f. (r.): Based on plaque morphology and host strain, we isolated 31 individual evolved phages from the phage mixtures (17 from round 15 and 14 from round 30) (Extended Data Figure E3).

Extended Data Figure E3:

Fig. E3: Overview of the isolation of evolved bacteriophages after rounds 15 and 30.

a, Tenfold serial dilutions of the bacteriophage (phage) mixtures after round 15 and 30 were individually spotted (5 μ l) on soft agar overlays for each individual bacterial strain in the evolution assay (PAO1, Paer03, Paer09, Paer33, Paer57, Paer60, Paer84 and Paer85). **b**, Based on the host strain and qualitative plaque assessment (plaques' size, contour, and turbidity/clarity) 17 evolved phages were picked from round 15 spot assays and 14 from round 30. The phages were purified by four consecutive single-plaque-passages. **c**, Each isolated phage was produced from a single plaque using either a liquid or solid propagation method. **d**, Phage genomes were extracted, sequenced with Illumina, and assembled before the corresponding ancestral phage was identified using BLASTn.

3. Are there differences in the capacity to drive phage evolution across the bacterial clusters (selection of the host for replication for the evolution assays)?

Response to comment: Thank you for this question. Taking the increase in phage infectivity as a surrogate for phage evolution, we were able to show that the phages display a disproportionate increase in infectivity gains among the eight strains included in the evolution (14 gains; 1.75 per strain), compared to the 42 strains not included in the evolution assay (53 gains; 1.26 per strain). The highest increase in phage infectivity (8 gains) occurred in Paer85, which was susceptible to eight evolved phages, while resistant to their corresponding ancestral phages. From the eight evolution strains, originating from all four genomic clusters (4 strains from 1; 2 strains from 2; 1 strain from 3; 1 strain from 4) for the purpose of diversity, only three strains (Paer09, Paer85 and PA01) were responsible for all infectivity gains. Interestingly, all three strains belong to the genomic cluster 1 (Figure 2a), which was the largest cluster among all strains (n=57, 71.3%).

To summarize, within our phage evolution assay the phages have primarily adapted to the eight evolution strains, disproportionately increasing their infectivity on these. Although the three strains belong to the genomic cluster 1, we do not want to hypothesize that this cluster has a higher capacity to drive phage evolution. This was neither the goal of our study nor do we have a sufficient sample size for a definitive answer.

To maximize diversity, inclusion criteria for the bacterial strains in the evolution assay were (a) susceptibility to at least one of the ancestral phages, (b) genomic diversity, (c) the antibiotic resistance profile, (d) diversity in biofilm formation (SEM images), and (e) non-auto-plaque former (line 685ff.).

4. Are the bacterial mutants resistant to phage affected in their virulence? Evidence in the literature show that some bacterial mutants resistant to phage lose some virulence (e.g. those lacking of LPS)

Response to comment: This is an interesting point which should indeed be addressed. To test this hypothesis, we selected a representative subset of seven phage resistant bacterial mutants: one for each mutation observed (WapR, PslA, RmlA, RmlB, RmlC, RmlD and glycoside hydrolase). These strains were compared to the untreated Paer09 in terms of their growth, motility (swimming and swarming), biofilm formation using porous glass beads (CFU count) and matrix production using crystal violet assays. Moreover, their virulence was assessed in a *Galleria mellonella* model. Please refer to line 874ff. of the methods section for additional details.

To summarize, all mutant strains displayed a reduced virulence resulting in a greater survival of larvae after 100 h at which point all naive Paer09-infected larvae were dead (Figure E4b). Included in the file “NCOMMS-23-23549A_DBPR_Extended_Data” Figure E4 illustrates the results of the strain analysis of phage resistant bacterial mutants.

Line 370ff. (r.): All mutant strains exhibited lower virulence *in vivo*, resulting in a higher larval survival rate after 100 h, when all naive Paer09-infected larvae were dead.

Extended Data Figure E4:

Fig. E4: Characterisation of phage treated Paer09 mutants.

a, Optical density (OD₆₀₀) measurements of a 24 h incubation of the planktonic naive Paer09 strain (GC, growth control) alongside seven phage treated Paer09 mutants with one altered gene (WapR, frameshift; PslA, nonsense; RmlA, nonsense; RmlB, missense; RmlC, frameshift; RmlD, nonsense; glycoside hydrolase, frameshift) per mutant. Each line represents the average of three biological replicates with three underlying technical replicates corrected by the blank control (TSB). **b**, Percentual survival of *Galleria mellonella* larvae after injection with 10 μ l of bacteria (10³ CFU/ml) (n=10 per strain; Paer09 and mutants stated above) or PBS (control; black line with no deaths; n=10) monitored for 100 h. **c**, Efficacy of plating (EOP) of five phages (FJK, FJK.R9-15, FJK.R9-30, MK, and MK.R3-15) for seven representative Paer09 mutants (mutants stated above) complemented with their respective wild-type gene. The EOP is defined as the concentration ratio of a phage on the complemented mutant (numerator; plasmid with wild-type gene) and the non-complemented mutant (denominator; empty plasmid). An EOP above 10 was considered as “increased” efficiency (green), while an EOP of 0.1 to 10 was ranked as “unchanged” efficiency (grey). A “reduced” efficiency was defined as an EOP between 0.001 and 0.1 (light blue). When no individual plaques were visible or the EOP was equal to or under 0.001 the efficiency was considered “greatly reduced” (dark blue). **d**, Optical density (OD₅₇₀) measurements of crystal violet/ethanol solution obtained from stained bacterial microtiter plate biofilms (Paer09 and mutants stated above). **e**, Bacterial load (CFU/bead) of 24 h pre-established bacterial biofilms (Paer09 and mutants stated above), determined by plating. **f**, Swarming motility of each strain (Paer09 and mutants stated above) on TSA plates (0.5% w/v agar). **g**, Swimming motility of each strain (Paer09 and mutants stated above) on TSA plates (0.3% w/v agar).

Unless specified otherwise, all experiments were performed in three biological replicates with either three (a and e), two (d) or no (c) technical replicates, except the motility which was determined in

four biological replicates (f and g). The error bars represent the standard error of the mean. The statistical analysis was conducted by unpaired two-tailed Student's *t* test and *p*-values were indicated by an asterisk (*, $p < 0.05$; **, $p < 0.01$; ***, $p < 0.001$; ****, $p < 0.0001$).

5. Biofilms are characterized by bacterial populations with low metabolic rate. I would suggest the authors to investigate whether the evolved phages are active against stationary phase bacteria in planktonic cultures.

Response to comment: We concur and have introduced additional experiments to address this. To test this hypothesis, we infected planktonic stationary cultures of *Pseudomonas aeruginosa* Paer09, Paer36 and Paer57 with their corresponding phages under OD₆₀₀ monitoring. Briefly, bacteria were grown at 37 °C and 100 rpm orbital shaking for 32 h before being infected with phages at an approximate MOI of 0.1 and 0.01 and co-incubated under OD₆₀₀ monitoring at 37 °C and 100 rpm orbital shaking for 8 h.

To summarize, neither the ancestral nor the evolved phages were able to actively replicate on bacteria within the stationary phase.

Response letter Figure 1:

Response letter Fig. 1: Activity of unevolved and evolved phages against bacteria in stationary phase.

a, Optical density (OD₆₀₀) measurements of a 32 h incubation of planktonic Paer09 (to reach stationary phase prior phage addition) followed by 8 h co-incubation with phages FJK, FJK.R9-15 and FJK.R9-30 at an MOI of 0.1 (full line) and 0.01 (dotted line) or without phage (Growth Control, dashed black line). The lines represent the average from three replicates. **b**, Optical density (OD₆₀₀) measurements of a 32 h incubation of planktonic Paer57 (to reach stationary phase prior phage addition) followed by 8 h co-incubation with phages MK, MK.R57-15 and MK.R57-30 at an MOI of 0.1 (full line) and 0.01 (dotted line) or without phage (Growth Control, dashed black line). The lines represent the average from three replicates. **c**, Optical density (OD₆₀₀) measurements of a 32 h incubation of planktonic Paer36 (to reach stationary phase prior phage addition) followed by 8 h co-incubation with phages MK, MK.R3-15 and MK.R3-30 at an MOI of 0.1 (full line) and 0.01 (dotted line) or without phage (Growth Control, dashed black line). The lines represent the average from three replicates.

Minor comments

1. Abstract is missing the goal of the study

Response to comment: Thank you. We have revised the abstract accordingly.

Line 14ff. (a.): This study aims at an *in vitro* biofilm evolution assay to improve multiple phage parameters in parallel and the optimization of phage cocktail design by exploiting a bacterial phage resistance trade-off.

2. Line 17-18: please rephrase, it is not clear. Does the authors mean by “under” and “determined” “as assessed by?”

Response to comment: Apologies, we have revised the phrase to clarify this misunderstanding.

Line 16ff. (a.): The obtained evolved phages show an expanded host spectrum, improved antimicrobial efficacy and enhanced antibiofilm performance, as assessed by isothermal microcalorimetry and RT-qPCR, respectively.

3. Line 206 – Do the author mean “FJK.R9-30 achieve a heat flow production of 11.6mcW at the maximum of suppression? (i.e. before resistance occurs?). I would rephrase it

Response to comment: Thank you. We have revised the phrase accordingly.

Line 215ff. (r.): When incubated with Paer09, the evolved phage FJK.R9-30 revealed a minimum heat flow of 11.6 μ W, corresponding to the highest suppression effect on bacterial heat production prior to their outgrowth. This represents a 68.4% ($p=0.0004$) greater heat flow suppression compared to the ancestral phage FJK (36.6 μ W) (Figure 3b).

4. *Line 285, which two?*

Response to comment: Apologies, we have clarified the phrase as indicated below.

Line 297ff. (r.): Evolved phages derived from the phages MK and JS (MK.R3-15, MK.R3-30, MK.R84-15, MK.R84-30, MK.R57-15, MK.R57-30) appear to be a recombination of these two ancestral phages.

5. *Characters in figures are difficult to read (too small)*

Response to comment: We concur and have revised figures 1, 2, 3, 4, 5 and 6 accordingly. Please refer to the manuscript for the revised figures.

Reviewer #2

In this manuscript, the authors describe an in vitro evolution assay to improve the efficacy of four phages against a panel of 8 Pseudomonas aeruginosa strains. The four phages were newly isolated from sewage samples and were briefly characterized by genome sequencing and electron microscopy to classify them. A subset of evolved phages obtained through a novel phage evolution platform was then selected and tested against a panel of 80 P. aeruginosa strains and some of the evolved phages were able to infect additional strains but also no longer to infect a few others. The selected evolved phages were characterized by genome analysis as well as antimicrobial/antibiofilm properties and were shown to be more efficient (reduced cell counts and longer lag time for resistant strains to emerge), possibly due to specific viral mutations. Phage-resistant bacterial strains were also characterized, and several mutations were predicted, which may be involved in the phage resistance phenotype. Finally, two evolved phages were used in a cocktail and were shown to be highly efficient compared to single evolved phages in eradicating Pseudomonas.

First and foremost, I certainly appreciate the remarkable amount of work presented in this manuscript. Quite impressive. However, it also unfortunately raised a significant number of questions for a manuscript submitted to Nature Communications.

Response to comment: We are very thankful for the appreciation of our work and the comments as outlined below. We hope to have clarified the key points which have been raised.

Major comments

1. The authors are presenting a new labor-intensive in vitro platform to evolve phages (Figure 1). Yet, this platform appears to have been tested only once with four phages (simultaneously) to generate evolved phages. Perhaps I missed the other trials, if so, data lack standard errors. This needs to be clarified. If a new platform is proposed, one would have expected a few more trials. Also, if the platform was inspired from the Applemans protocol, maybe the differences could be highlighted in the Materials and Methods section.

Response to comment: Our study represents a proof of concept for this novel *in vitro* phage evolution platform allowing to improve multiple phage parameters in parallel. As a starting point for this trial with *P. aeruginosa*, our aim was to highlight the overall capabilities of this protocol and it will therefore need further validation and improvement in the future. Nevertheless, previous studies have found, the evolution of phages, given the small genomes and thereby limited evolutionary pathways, demonstrates a great reproducibility down to the codon and nucleotide level¹⁻⁴. Repeating the entire protocol, will require at least a year or more and extensive resources. We propose to indicate this as a limitation of the study and emphasize the labour-intensive nature of this approach.

We have further revised the manuscript accordingly.

Line 88ff. (r.): This assay was implemented once as a proof of concept using four lytic *P. aeruginosa* phages (JS, MK, FIM, FJK), belonging to three distinct genera, which were trained on eight *P. aeruginosa* strains (PAO1, Paer03, Paer09, Paer33, Paer57, Paer60, Paer84 and Paer85), which also display diversity in genomic phylogenetics, antibiotic resistance, and biofilm formation traits (Figure 2a; Extended Data Figure E1; Supplementary Data S1).

Line 513ff. (d.): As a proof of concept, the results of our study are based on a single directed evolutionary assay, but as previous studies have found, the evolution of phages, given the small genomes and thereby limited evolutionary pathways, demonstrates a great reproducibility down to the codon and nucleotide level^{40,54-56}.

Line 534ff. (d.): At the same time, the platform itself could be further streamlined by optimizing the number and length of each evolution cycle, as it is a labour-intensive approach impeding generalized applicability.

Line 671ff. (m.): The *in vitro* phage evolution assay to improve multiple phage parameters in parallel, consisted of a serial passaging approach with thirty consecutive rounds, inspired by the directed evolution approach of the Appelmans protocol⁸¹. Adaptations to accommodate bacterial biofilms included the use of microcalorimetric real-time monitoring, revised active sample criteria and performance-dependant phage-mixture-dilutions.

2. *The authors claim that the in vitro platform evolved phages to expand their host range. While true, it was also modest. In Figure 2, 41 strains out of 80 (51%) were sensitive to at least one of the ancestral phages. After 30 rounds of replication, the evolved phages could infect 8 additional strains (+10%) but three strains (-4%) could no longer be infected. Therefore, at the end, 45 strains (out of 80) were sensitive to one of the phages. While this increased host range is noticeable, this seems a lot of work to modestly increase the host range. Perhaps this could be discussed.*

Response to comment: We respectfully disagree with this statement, which, from our perspective, makes an incorrect comparison. Perhaps this misunderstanding is due to an unclear statement in our manuscript. The aim of our study was to improve multiple phage parameters in parallel, which includes the expanded host range. Within our evolution platform we trained four phages on a subset of eight *P. aeruginosa* strains selected from our diverse, extensive collection (n=80). Keeping in mind that most strains had never seen our phages, we focused on the individual phages host range improvement rather than the global host range increase.

We observed that phage MK.R57-30, which showed the largest host range expansion, infected 10 additional strains, for a total of 30 susceptible strains, an increase of 76.5% and 42.9% compared to the ancestral phages JS (n=17) and MK (n=21), respectively.

To avoid misunderstandings, we have revised the manuscript.

Line 180ff. (r.): Focusing instead on the entire collection of *P. aeruginosa* strains (n=80), 30 strains (37.5%) were not susceptible to any phage, and overall, the evolved phages were able to increase the number of susceptible strains to 45, an increase of 9.8%.

3. *The authors also argued that the evolved phages have an improved antibiofilm and antimicrobial activities (Figure 3). Yet, the cell counts were reduced by only two logs (from 10e8 to 10e6 CFU/beads) at best. Again, while clearly better, it is still modest in terms of phage efficacy. The emergence of phage resistant strains was indeed delayed with the evolved phages but still, those mutants could still be obtained readily. Again, this seems like a modest gain.*

Response to comment: Thank you for this comment. While the antimicrobial biofilm and antibiofilm activity might be considered as moderately efficient against an established biofilm, we consider it important, nonetheless. Especially as this needs to be combined with the greatly improved antimicrobial planktonic capability. Together, we believe it exactly highlights the intended outcome of the evolution assay.

Indeed, besides an increased host range, the evolved phages had an improved antimicrobial biofilm activity and antibiofilm efficacy. It is important to realize those phage improvements were achieved against a robust *in vitro* biofilm formed on a porous glass bead, providing a large three-dimensional surface for biofilm to form (Extended Data Figure E1)^{5,6} and which is notoriously difficult to disrupt. This makes the observed reductions notable. Moreover, the use of real-time quantitative polymerase chain reaction (RT-qPCR) represents a more conservative quantification of antibiofilm efficacy, as live and dead bacteria not dip-washed away are not differentiated by qPCR. This is underscored by the fact, that the antimicrobial planktonic activity of the evolved phages was greatly improved and in some cases resulted in possible bacterial eradication at an MOI of 0.001 (Figure 5e, f, g).

We have adapted the manuscript to highlight this in more detail.

Line 326ff. (r.): To investigate this question, we co-incubated planktonic bacterium Paer09 with either ancestral phage FJK or one of the evolved phages (FJK.R9-15/30) at an MOI of 0.001 for three days.

Line 546ff. (d.): As RT-qPCR does not distinguish between live and dead bacteria, the results provided represent a more conservative antibiofilm efficiency, considering the possibility that DNA from dead bacteria not dip-washed away is also quantified.

4. *Understanding how phages and bacteria evolved is essential to improve phage therapy. The authors did a remarkable job in isolating and characterizing evolved phages and bacteria. Yet, the bacterial part is incomplete (Figure 5). The authors could not identify mutations in a few of the resistant strains, which raised questions on how the bacterial genomes were analyzed. For one, it is unclear if complete genomes were obtained. For most of the resistant strains, mutations were observed, but they may be due to sequencing errors as they were surprisingly not confirmed by PCR+Sanger sequencing. More importantly their involvement in phage*

resistance was not confirmed by reintroducing the wild-type gene in the resistant clones to restore the phage sensitivity profile. As of now, the roles of the observed bacterial mutations in the phage resistance are still unclear. Similarly, a number of mutations were also observed in the evolved phages but it is difficult to determine why these phage mutants were able to thrive on specific hosts (Figure 4). Therefore, we do not know more about the evolutionary genetics of interactions between evolving phages and bacteria as mentioned by the authors in the introduction.

Response to comment: Upon request of the reviewer, we have introduced the controls listed above.

After three days of co-incubation of the planktonic bacterium Paer09 with either ancestral phage FJK or one of the evolved phages (FJK.R9-15/30), 48 bacteria were isolated. From these, total bacterial genomic DNA was extracted and sequenced with Illumina. Please refer to line 845ff. and line 737ff. of the methods section for additional details on the isolation and whole genome sequencing.

Among the majority of strains (43/48) we identified mutations. For a subset of seven representative phage resistant bacterial mutants, one for each mutation observed (WapR, PslA, RmlA, RmlB, RmlC, RmlD and glycoside hydrolase) the genomic regions carrying the mutations were amplified using polymerase chain reaction and sequenced by Sanger sequencing. To summarize, all mutations could be validated.

To verify the mutations' involvement in the phage resistance we performed a complementation assay with the cloned wild-type genes transformed to a representative subset of seven phage resistant bacterial mutants, one for each mutation observed (WapR, PslA, RmlA, RmlB, RmlC, RmlD and glycoside hydrolase). To summarize, the plasmid complementation with the wild-type gene led to a reversal of the resistance trade-off, resulting in greater susceptibility (increased EOP) to FJK, FJK.R9-15 and FJK.R9-30 and loss of susceptibility (decreased EOP) to MK and MK.R3-15. This was most prominent in the RmlA, RmlC and WapR isolates. Please refer to line 910ff. of the methods section for additional details. The remaining isolates (5/48) showed no mutation and had an identical phage susceptibility profile to the nonexposed Paer09 control (Figure 5a). These isolates managed to avoid phage contact, by for example remaining in an ecological, spatial refuge such as bacterial growth on the microplate well wall and did not require adaptation to phage predation, resulting in no mutations and no change of their susceptibility profile⁷.

We have introduced these new experimental data in the manuscript.

Line 77ff. (i.): This study provides a phage training platform and helps to better understand bacterial phage resistance and enables us to identify different genomic parameters conferring enhanced phage efficacy.

Line 410ff. (r.): Among the 48 bacterial isolates, five isolates (10.4%; no apparent mutations) showed the same phage susceptibility profile as the control, except one who had an increased efficiency of plating for phage FJK.R9-30 (Figure 5a). These five

isolates and the control showed resistance to phage MK, while no isolate was resistant to the evolved phage MK.R3-15.

Line 405ff. (r.): Plasmid complementation of the mutated genes with the wild-type genes in the six representative bacterial isolates resulted in a reversal of this trade-off, which was most prominent in the RmlA, RmlC and WapR isolates (Extended Data Figure E4c).

5. Finally, the authors selected two evolved phages and thoroughly tested their efficacy in a cocktail. Yet, they did not compare the cocktail containing the two evolved with a cocktail contain one of the wild-type phage (FJK) which can replicate on the naïve Paer09 strain (Figures 2 and 6).

Response to comment: The experimental setup of the study was designed so that each successive experiment builds on the previous results. For the antimicrobial activity, 24-h-biofilms or planktonic Paer09 were co-incubated with individual phages under isothermal microcalorimetry or optical density monitoring, respectively. The initial direct comparison of the ancestral phage (FJK) with the evolved phage (FJK.R9-30) showed a significantly higher heat flow suppression and lag time (Figure 3b, c, d and Figure 5d). Building on this, we tested the evolved phages individually and as a 1:1 cocktail against Paer09, using the same setup. The cocktail demonstrates a significantly greater heat flow suppression and lag time (Figure 6d, e, f, i).

To summarize, the evolved phages outperformed the unevolved, ancestral phages and were in turn outperformed by the phage cocktail combining the two evolved phages. Although we do not have experimental confirmation, we strongly feel that exchanging the evolved phage FJK.R9-30 for its unevolved ancestral phage FJK would not improve the efficacy of the cocktail, the intended outcome of the experiment.

The authors have a fantastic dataset, but it may not have been presented in an optimal way. Most importantly, it may not have been fully analyzed.

Minor Comments

1. Line 10, I suggest replacing “natural predators of bacteria” by “the viruses of bacteria”

Response to comment: We concur. We have revised the phrase accordingly.

Line 10 (a.): Lytic bacteriophages (phages), the viruses of bacteria, represent a path to combat this threat.

2. Line 21, As it is submitted to Nature Communications, I would have expected that other bacterial pathogens would have been tried.

Response to comment: To underscore the broader applicability of our new evolution platform, we here provide new data to extend the proof of the approach to include an

example from a Gram-positive bacterial species, more specifically the extended host spectrum, antibacterial and antibiofilm efficacy improvement of *S. aureus* phages. However, it would have our strong preference not to include this in the manuscript, given our already extensive and thorough dataset, and as to not disrupt the storyline and logical flow of our manuscript. As this suggestion was indicated as a minor comment, and also not raised by the other reviewers, we hope the editor and reviewer can agree with our assessment.

Extended dataset on *S. aureus* phage evolution

With this experiment we aimed to improve multiple antimicrobial capabilities of *S. aureus* phages in parallel using an *in vitro* serial-passage biofilm evolution platform employed for 30 rounds under real-time isothermal microcalorimetry monitoring. In line with our study on *P. aeruginosa* we could extend the phages' host range and improve their antimicrobial and antibiofilm activity. This study also includes the investigation of *S. aureus* phage cocktails.

Host range expansion: The host range of each phage was determined by soft agar overlay spot assay. The evolved phages (n=15) demonstrated an expanded host range among the biobank of *S. aureus* strains (n=72). Compared to the broadest unevolved phage (n=32; 44.4%), the broadest evolved phage could infect 28 additional strains, an increase of 87.5%, resulting in 60 susceptible strains (83.3%). Overall, 69 strains (95.8%) were susceptible to at least one evolved phage while only 51 (70.8%) were susceptible to the ancestral phages (n=5).

Antimicrobial biofilm activity: Biofilms of *S. aureus* were co-incubated with unevolved and evolved phages (individual or as a cocktail) and monitored by isothermal microcalorimetry for 72 h. The investigation of two strains included in the evolution assay demonstrates that evolved phages display the highest antimicrobial biofilm activity, while not all evolved phages surpassed the ancestral unevolved phages' efficacy (Response letter Figure 1). Depending on strain and time point a cocktail of two phages further improved the antimicrobial biofilm activity.

Response letter Figure 2:

Response letter Fig. 2: Calorimetric heat curves of biofilms exposed to unevolved and evolved phages.

Each curve shows the heat (J) produced over time by viable bacteria in the biofilm over 72 h monitoring. **A)** GE-MRSA15 biofilm exposed to ancestral (CUB-M and ISP) and evolved (CUB_GE-MRSA15_R7/R14/R23) phages. **B)** GE-MRSA15 biofilm exposed to evolved phages CUB_GE-MRSA15_R14 and CUB_MRSA-COL_R23 alone or combined 1:1 as a phage cocktail. **C)** MRSA-COL biofilm exposed to ancestral (ISP) and evolved (CUB_MRSA-COL_R9/R20/R23) phages. **D)** MRSA-COL biofilm exposed to evolved phages CUB_GE-MRSA15_R14 and CUB_MRSA-COL_R23 alone or combined 1:1 as a phage cocktail.

GC corresponds to the growth control sample not exposed to phages. Data are expressed as mean \pm standard error from three biological replicates. Solid lines represent the mean and corresponding shaded regions the standard error. Plots were prepared using RStudio.

Antibiofilm activity: The ability of the unevolved and evolved phages (individual or as a cocktail) to reduce biofilm cell counts was assessed by RT-qPCR following 4 h, 8 h, 24 h or 48 h of biofilm exposure to phages. The results on two strains included in the evolution indicate that the evolved phages outperform the unevolved phages on a strain-dependant manner and are especially potent in reducing bacterial biofilm cell counts for prolonged periods (48 h) without incurring regrowth (Response letter Figure 2). The phage cocktail resulted in a slightly greater cell count reduction than the individual phages. Focusing instead on a strain, not included in the evolution assay, the evolved phages demonstrate a greater antibiofilm efficacy than the ancestral phage but are not able to sustain the prolonged cell count reduction, as demonstrated in the strains included in the evolution.

Response letter Figure 3:

Response letter Fig. 3: Antibiofilm activity of unevolved and evolved phages.

Bacterial cell count, expressed as colony-forming-units per bead (CFU/bead). **(A)** GE-MRSA15, **(B)** MRSA-COL and **(C)** MRSA-USA300 biofilms after exposure to phages (10⁷ PFU/mL) for 0 h, 4 h, 8 h, 24 h and 48 h determined by RT-qPCR.

GC indicates the growth control sample not exposed to phages. Results were statistically analysed using a student's *t* test with Welch correction.

3. Lines 79-80, “... to identify different bacterial or viral genomic parameters”.

Response to comment: We have revised the phrase.

Line 77ff. (i.): This study provides a phage training platform and helps to better understand bacterial phage resistance and enables us to identify different genomic parameters conferring enhanced phage efficacy.

4. Line 90, the name of the four phages should be indicated here.

Response to comment: We have adapted the phrase as requested.

Line 88ff. (r.): This assay was implemented once as a proof of concept using four lytic *P. aeruginosa* phages (JS, MK, FIM, FJK), belonging to three distinct genera, which were trained on eight *P. aeruginosa* strains (PAO1, Paer03, Paer09, Paer33, Paer57, Paer60, Paer84 and Paer85), which also display diversity in genomic phylogenetics, antibiotic resistance, and biofilm formation traits (Figure 2a; Extended Data Figure E1; Supplementary Data S1).

5. Line 91, the name of the eight strains should be indicated here.

Response to comment: Thank you. We have revised the phrase accordingly.

Line 88ff. (r.): This assay was implemented once as a proof of concept using four lytic *P. aeruginosa* phages (JS, MK, FIM, FJK), belonging to three distinct genera, which were trained on eight *P. aeruginosa* strains (PAO1, Paer03, Paer09, Paer33, Paer57, Paer60, Paer84 and Paer85), which also display diversity in genomic phylogenetics, antibiotic resistance, and biofilm formation traits (Figure 2a; Extended Data Figure E1; Supplementary Data S1).

6. Line 94, were the phages first amplified on the host used to isolate them (data S2)? Should be indicated. Would their amplification on another host would have changed the evolutionary pattern?

Response to comment: The ancestral phages were propagated on their isolation strains using either a liquid or solid method and concentrated and purified by PEG 8000 precipitation before introduction into the evolution assay. We have revised the manuscript to include this information in the methods section, as line 93ff. is referring to each round, not just the first one.

Although each amplification represents a selection process, we do not want to postulate that pre-selected phages form another host strain would have led to a different evolutionary outcome, as this was neither the aim of our study nor do we have sufficient information for a definitive answer.

These elements have been introduced in the updated manuscript.

Line 600ff. (m.): Next, each isolated phage was produced from a single plaque on their isolation strain using either a liquid⁷¹ or solid⁷² propagation method.

7. Line 94, at the beginning of the experiment, was the EOP of the four phages close to 1 on the various strains? Perhaps a table of the EOPs could be provided in addition to Figure E1A.

Response to comment: The efficacy of plating (EOP) of the four ancestral phages on the eight strains included in the evolution varied between 0.02 and 1.9. Included in the file “NCOMMS-23-23549A_DBPR_Extended_Data” Figure E1a now illustrates the EOP values.

NCOMMS-23-23549A_DBPR_Extended_Data

Line 16ff.: **a**, Transmission electron microscopy images of the four ancestral bacteriophages (*Pakpunavirus*: MK/JS, orange square; *Bruynoghevirus*: FJK, yellow square and *Pbunavirus*: FIM, red square) and their host range (connecting lines; numbers indicate EOP) among the eight *P. aeruginosa* strains included in the *in vitro* evolution and belonging to four genomic clusters (Cluster 1: PAO1/Paer09/Paer33/Paer85, black circle; Cluster 2: Paer57/Paer60, blue circle; Cluster 3: Paer03, violet circle; Cluster 4: Paer84, green circle). The efficacy of plating (EOP) is shown as mean with standard error of the mean of four replicates, except for phage JS and MK (Paer57), for which there are duplicates.

Extended Data Figure E1:

Fig. E1: Overview of the bacteriophages and bacterial strains included in the evolution assay.

8. Line 100, out of curiosity, any reason why the evolution experiment was stopped after 30 rounds?

Response to comment: The 30 rounds were determined in advance and selected in accordance with the Appelmans protocol⁸.

9. Line 102, the data for Paer33 should be presented in the Extended Data Figure E2.

Response to comment: Apologies for not clearly stating that the evolved phages lost the capability to infect Paer33, thus making the illustration impossible, as no phage mixture concentration could be determined. Instead, Table S1 included in the file “NCOMMS-23-23549A_DBPR_Supplementary_Information” provides the underlying data for Paer33.

We have further revised the phrase accordingly.

Line 100ff. (r.): Throughout the evolution assay, focusing on rounds 1, 5, 10, 15, 20, 25, and 30, we observed an overall increased calorimetric heat reduction after 8 h in consecutive rounds of evolution for each individual strain (Extended Data Figure E2), except for Paer33 which was not susceptible to the evolving phages (Supplementary Table S1).

10. Line 110, Presumable it was also the case with Paer33 but it was discarded. Should be indicated.

Response to comment: We concur. Please find the requested additions as indicated below.

Line 110ff. (r.): When focusing on the entire monitoring period (24 h) instead, we observed that five strains (PAO1, Paer03, Paer33, Paer57 and Paer84) co-incubated with the phage mixture reached heat levels equivalent to those from their respective growth controls (Extended Data Figure E2).

11. Line 115, time and phage counts are shown where in the manuscript? No standard errors are provided.

Response to comment: Table S1 included in the file “NCOMMS-23-23549A_DBPR_Supplementary_Information” provides the analysed data for all evolution strains. Phage concentrations (PFU/ml) were evaluated in three biological replicates. The heat suppression times are given without standard error since the evolution assay was carried out once. For the corresponding methods we refer to line 724ff. (phage concentration) and line 933ff. (lag times) of the methods section.

Line 115ff. (r.): Among the analysed time points, the duration of this initial heat suppression is strain-specific and could reach up to 18.3 h (Paer57, round 5, 3.1×10^6 PFU/ml) (Supplementary Table S1), excluding strains Paer09 and Paer85.

Line 933ff. (m.): For the heat and heat flow curves of the antimicrobial activity testing the experimental replicates were interpolated to a 36 sec equidistant timeline and graphed

as mean. The heat curves of the *in vitro* bacteriophage evolution assay were used directly for further calculations.

12. Line 116, round 3? You mentioned on line 100 that the evolution assays were round 1, 5, 10, 15, 20, 25, and 30?

Response to comment: Apologies that this remark was not clear. The evolution assay was performed for 30 consecutive rounds, with the heat production being monitored in each round. For a detailed comparative analysis of the heat production and phage concentration, we focused on rounds 1, 5, 10, 15, 20, 25 and 30. Please refer to line 724ff. of the methods section for additional details. Focusing solely on the heat production, from round 3 onward, we observed a complete heat suppression ($\geq 75\%$) for the entire monitoring period in the undiluted phage sample of Paer09. We have rephrased accordingly.

Line 100ff. (r.): Throughout the evolution assay, focusing on rounds 1, 5, 10, 15, 20, 25, and 30, we observed an overall increased calorimetric heat reduction after 8 h in consecutive rounds of evolution for each individual strain (Extended Data Figure E2), except for Paer33 which was not susceptible to the evolving phages (Supplementary Table S1). [...] In the case of Paer09, from round 3 onward, we observed a complete heat suppression ($\geq 75\%$) for the entire monitoring period in the undiluted phage sample.

13. Line 131, five ampules per strain, could you clarify what these five ampules contained? 6 phage titers appear to have been tested in Figure E2? Were these assays performed at least in duplicate? No standard errors are provided in Figure E2?

Response to comment: Each of the five ampules contained a 24 h pre-established biofilm bead of one *P. aeruginosa* strain in tryptic soy broth and in four of them, the undiluted and appropriate serial dilutions of the phage mixture. Please refer to line 696ff. of the methods section for additional details.

The concentration of the phage mixtures at round 0 and after rounds 4, 9, 14, 19, 24 and 29 were determined *ex post* using soft agar overlay spot assays against the evolution strains in three biological replicates, except Per33 (round 0) and Paer60 (round 30) which were determined in duplicates. The calculated phage concentrations were then correlated with the corresponding dilution factors at the onset of rounds 1, 5, 10, 15, 20, 25 and 30, allowing for the comparison of the heat production, lag time and the phage concentration. Please refer to line 724ff. for additional details. The analysed data for all evolution strains underlying Figure E2 is provided in Table S1 included in the file “NCOMMS-23-23549A_DBPR_Supplementary_Information”.

We have revised the manuscript accordingly.

Line 133f. (r.): A total of 43 ampules (one growth control and four phage mixture dilutions per strain and three sterility controls) were used during each round of evolution.

NCOMMS-23-23549A_DBPR_Extended_Data

Line 30ff. (r): **a**, Heat reduction (%) plot of PAO1 exposed to the phage mixture at different concentrations (PFU/ml) – determined ex post – compared to the growth control sample in different rounds of the evolution after 8 h of monitoring.

14. Line 152, these 31 plaques were recovered and amplified on which *Pseudomonas* host(s)?

Response to comment: Evolved phages after round 15 and 30 of the evolution assay were isolated on soft agar overlays from each individual bacterial strain in the evolution assay (PAO1, Paer03, Paer09, Paer33, Paer57, Paer60, Paer84 and Paer85). Please refer to line 605ff. of the methods section for additional details. The evolved phages were propagated on their isolation strains.

We have clarified this in line 608ff..

Line 608ff. (m.): Based on qualitative plaque assessment and host strain we identified 31 phages that were purified and produced on their isolation strains as described above (Extended Data Figure E3).

15. Line 154, why pick only 31 plaques? More importantly, why focus on only 10 of them? What happened to the 21 others? Seems like a missed opportunity to analyze more, by picking many plaques on different hosts for example.

Response to comment: The phage mixtures of rounds 15 and 30 were spotted on all eight *P. aeruginosa* evolution strains, as those were the strains that the phages had been trained on and based on plaque morphology all differing phage plaques (n=31) were picked, amplified, and sequenced to determine their descent. Parameters of the qualitative plaque assessment were the plaques' size, contour, and turbidity/clarity. Figure E3 included in the file "NCOMMS-23-23549A_DBPR_Extended_Data" represents the isolation process of the evolved phages.

The subset of five phages for each round was chosen based on their descent to represent phages from all four ancestral phages (n=1 for FIM, n=1 for FK and n=3 for MK/JS). Due to the recombination of MK and JS, we decided to use three of these evolved phages. The decision to focus on only ten in total, five for each round, for further analysis was made considering the efficient allocation of personal and research resources. The remaining 21 evolved phages were stored at 4 °C in SM-buffer.

Extended Data Figure E3:

Fig. E3: Overview of the isolation of evolved bacteriophages after rounds 15 and 30.

a, Tenfold serial dilutions of the bacteriophage (phage) mixtures after round 15 and 30 were individually spotted (5 µl) on soft agar overlays for each individual bacterial strain in the evolution assay (PAO1, Paer03, Paer09, Paer33, Paer57, Paer60, Paer84 and Paer85). **b**, Based on the host strain and qualitative plaque assessment (plaques' size, contour, and turbidity/clarity) 17 evolved phages were picked from round 15 spot assays and 14 from round 30. The phages were purified by four consecutive single-plaque-passages. **c**, Each isolated phage was produced from a single plaque using either a liquid or solid propagation method. **d**, Phage genomes were extracted, sequenced with Illumina, and assembled before the corresponding ancestral phage was identified using BLASTn.

16. Line 157: "Representatives are unique phages descended from all four ancestral phages?"
 "It seems that no phages were recovered that are derived from JS."

Response to comment: Evolved phages (MK.R3-15, MK.R3-30, MK.R84-15, MK.R84-30, MK.R57-15, MK.R57-30) appear to be a recombination of the ancestral phages MK and JS. Please refer to the BLASTn comparison of the evolved phages with phages MK and JS found in Supplementary Data S2 included in the file “NCOMMS-23-23549A_DBPR_Supplementary_Information”.

17. Line 158: Seem surprising that those data are not shown at least in supplementary data.

Response to comment: The preliminary data was not shown as it was reproduced in biological replicates and presented in Figure 2, Figure 3 and the corresponding result sections.

The term “preliminary test (data not shown)” was intended to explain the selection of these representatives at this stage of the publication, but to avoid confusion we have removed this reference.

We have revised the phrase accordingly.

Line 160f. (r.): The representatives are unique phages descended from the ancestral phages, showing distinct efficiency and host range.

18. Line 158-161: I do not understand this sentence, seems out of place. Do you mean that the strain Paer36 was selected here because it was resistant to the four ancestral phages? Were the ten selected plaques able to infect these 3 strains?

Response to comment: We are happy to clarify this statement. This sentence briefly mentions why we decided to focus on these three strains for further analysis. Paer36 was selected as a representative strain from the collection that had not been included in the evolution assay, was susceptible to the four ancestral phages and had a high resistance profile (4MRGN). Please refer to Figure 2a for further details on the varying phage susceptibilities of the three strains.

We propose the following revision:

Line 161ff. (r.): Similarly, from the 80 bacterial strains in our collection, three strains with distinct phage susceptibility and resistance profile were selected, two of which were included in the evolution assay (Paer09 and Paer57) and one that was not included (Paer36), as target strains for further analysis of the antibiofilm and antimicrobial activity.

19. Line 171, should the percentage of increase be compared to the ancestral phages too?

Response to comment: We concur and have introduced this percentage for the reader’s convenience.

Line 173f. (r.): Compared to its genomic ancestors (MK and JK), phage MK.R57-30 showed the largest host range expansion by 76.5% and 42.9%, respectively.

20. Line 178, should indicate that the tree was made using the core genes (how many genes?). The tree, while useful still makes this figure difficult to analyze. It would have been much easier for the readers to group the sensitive strains according to their phage sensitivity profiles. As presented, it puts an emphasis on the tree, while the manuscript is more about phage sensitivity. The four clusters should be kept though.

Response to comment: The intention behind the presentation of the phylogenetic tree is to illustrate the phylogenetic diversity among the biobank and highlight that there is not one cluster that is fully susceptible or resistant to the phages. Among the 50 bacterial strains excluding the completely resistant strains (n=30) there are 34 unique phage susceptibility profiles that do not correlate with any of the other characteristics shown. Grouping those profiles would not facilitate the analysis. In Figures 2b and 2c the phage susceptibilities are compared and analysed.

We have revised the phrase and manuscript accordingly.

Line 188ff. (r.): a, Phylogenetic tree based on the core genes (3,667 of 19,556 genes) of the collection of *P. aeruginosa* strains (biobank of 80 strains) encompassing four genomic clusters (Cluster 1, black branch, 71.3%; Cluster 2, blue branch, 25.0%; Cluster 3, violet branch, 1.3%; Cluster 4, green branch, 2.5%).

Line 747ff. (m.): After determining the core genome (3,667 of 19,556 genes; 99% \leq strains \geq 100%; min. pct. identity for BLASTp: 95) using Roary v3.13.0⁹, RAxML v8.2.4¹⁰ was used to infer a maximum likelihood phylogenetic tree of the complete *P. aeruginosa* strain collection, which was then visualised with iTOL v6.5¹¹.

21. Line 182, MRGN should be defined here.

Response to comment: Thank you. Please also refer to line 584ff. of the methods section for three references explaining the MRGN classification in detail.

We have introduced this definition as requested.

Line 191f. (r.): Classification of each strain according to its resistance profile (4MRGN, red lettering; 3MRGN, blue lettering; MRGN, multi-resistance Gram-negative).

22. Line 192, where are these evolution strains in the Figure?

Response to comment: Figure 2d illustrates the cumulated infectivity gains, defined as the ability of an evolved phage to infect a bacterial strain which its closest unevolved progenitor could not. The grouping highlights if the infectivity gain occurred on an

evolution strain (ES) or one of the remaining strains (AS, excluding the ES and CRS) (CRS, completely resistant strains). Of the eight evolution strains, the 14 infectivity gains occurred on three evolution strains (Paer85, 8 gains; PAO1, 4 gains; Paer09, 2 gains) which are displayed as the stacked blue boxes.

23. Figure 2ab, you should indicate the number of strains that are infected by each phage.

Response to comment: Agreed. We have introduced these numbers in Figure 2b.

Line 195 (r.): **b**, Percentual (white lettering) and absolute (black lettering) infectivity of each phage among the biobank.

Figure 2:

Fig. 2: Host range analysis of unevolved and evolved phages.

24. *Figure 2d, can't understand this Figure* \neg .

Response to comment: Apologies for this. Figure 2d illustrates the cumulated infectivity gains, defined as the ability of an evolved phage to infect a bacterial strain which its closest unevolved progenitor could not. Taking Paer85, as an example, phage FJK could not infect this strain, while FJK.R9-15 and FJK.R9-30 could infect it (Figure 2a). This constitutes an infectivity gain for both evolved phages. Among the strains included in the evolution (n=8), three strains (stacked blue boxes) were susceptible to evolved phages and resistant to the phages' closest unevolved progenitor. In total, there were 14 infectivity gains (Paer85, 8 gains; PAO1, 4 gains; Paer09, 2 gains). These 14 infectivity gains are compared to 53 infectivity gains on the remaining strains (AS, excluding the ES and CRS) (CRS, completely resistant strains). For a direct comparability of the two strain populations (ES and AS), the cumulated values were divided by the number of strains in each group, resulting in an averaged 1.75 gains per strain (ES) and 1.26 gains per strain among the remaining strains (AS). This leads to a 38.7% greater increase in infectivity of the evolved phages among the eight bacterial strains included in the evolution assay than in the strains not included.

To summarize, comparing the average infectivity gains of the evolved phages on the evolution strains to the ones not included in the evolution shows that the phages had a higher infectivity increase on the strains included in the evolution.

To clarify this point, please find below the listed changes:

Line 198ff. (r): **c**, Representation of the infectivity gains and losses of the evolved phages. An infectivity gain (yellow bar) is defined as the capability of an evolved phage to infect a bacterial strain that was initially not susceptible to the phage's genomically closest unevolved progenitor phage. A loss of infectivity (red bar) occurred when a bacterial strain is susceptible to the unevolved ancestral phage but resistant to the corresponding evolved phage.

Line 202ff. (r): **d**, Representation of the cumulated infectivity gains of the evolved phages among the evolution strains (ES) or all strains (AS, excluding the ES and CRS). Among the ES (n=8), three strains (stacked blue boxes, numbers at the side of the boxes indicate the infectivity gains on the according strain) resulted in 14 infectivity gains (Paer85, 8 gains; PAO1, 4 gains; Paer09, 2 gains). The 53 infectivity gains among AS (n=42) correspond to gains on 22 strains (stacked grey boxes, numbers at the side of the boxes indicate the infectivity gains on the according strain). For a direct comparability of the two strain populations (ES and AS), the cumulated values were divided by the number of strains in each group, resulting in averaged gains per strain (yellow circles).

25. *Line 203, you are referring to which Figure here? 2? If so, yes but still limited.*

Response to comment: The introductory sentences (line 211ff.) refer to and summarize Figures 3b, f, j and Figures 3d, h, i which illustrate the significantly lower bacterial minimum heat flow and thus higher bacterial suppression, as well as the significantly increased lag time of the evolved phages compared to their respective unevolved ancestral phages.

26. Line 205, supplementary table S1, how many times were these assays performed? No standards errors were provided.

Response to comment: Isothermal microcalorimetry was used to compare the antimicrobial activity of ancestral and evolved phages isolated at round 15 and 30 against the biofilm of the strains Paer09, Paer36 and Paer57. The experiments were performed in two biological replicates with two technical replicates each.

We have revised Supplementary Table S1, S2, S3 and S4 accordingly.

27. Line 205, it should be noted that only three strains were tested here (Table S1).

Response to comment: The revised manuscript states and explains the use of these three strains for the following antimicrobial biofilm and antibiofilm efficacy tests in the introductory paragraph of this results section.

We have revised the manuscript accordingly.

Line 161ff. (r.): Similarly, from the 80 bacterial strains in our collection, three strains with distinct phage susceptibility and resistance profile were selected, two of which were included in the evolution assay (Paer09 and Paer57) and one that was not included (Paer36), as target strains for further analysis of the antibiofilm and antimicrobial activity.

28. Line 209, indicate the *Pseudomonas* strain for phage MK here.

Response to comment: This would indeed be clearer. We have revised the phrase accordingly.

Line 220ff. (r.): Ancestral phage MK showed a minimum heat flow of 77.2 μ W on Paer57, whereas the evolved phage MK.R57-30 reached a 96.5% ($p=0.0053$) greater reduction at 2.7 μ W (Figure 3f).

29. Line 221, should it be 74.8% instead of 74.9%? (see Supplementary Table S2)

Response to comment: Apologies for this mistake.

Line 232ff. (r.): After 6 h co-incubation of phage FJK.R9-30 and Paer09, the cell count was 74.8% ($p=0.0010$) lower compared to results from ancestral phage FJK (Figure 3a).

30. Lines 217-229, while the various percentages appear interesting, the improvement is still less than 2-log cell count reduction (see Supplementary Table S2 for the CFU counts). This is rather modest. On the positive side, the lag time before the emergence of resistant cells is notable (see Supplementary Table S1 for lag time). It would be of interest to know if these resistant cells can grow at the same level as their ancestral strains.

Response to comment: This is an interesting point. To test this hypothesis, we selected a representative subset of seven phage resistant bacterial mutants, one for each mutation observed (WapR, PslA, RmlA, RmlB, RmlC, RmlD and glycoside hydrolase). The growth curves of these strains and the untreated Paer09 were compared under optical density (OD) measurement. Please refer to line 877ff. of the methods section for additional details.

To summarize, only the growth curves of the RmlC (7.6 h) and RmlD (9.4 h) mutants showed a 116% and 166% prolonged lag time, respectively, compared with the naive Paer09 bacterium (3.5 h).

Furthermore, we investigated the seven representative isolates regarding their virulence in *Galleria mellonella*, biofilm formation and motility. Included in the file “NCOMMS-23-23549A_DBPR_Extended_Data” Figure E4 illustrates the results of the strain analysis of phage resistant bacterial mutants.

We have revised the manuscript accordingly.

Line 364ff. (r.): Six representative bacterial isolates, each mutated in only one of the six identified genes (RmlA, nonsense; RmlB, missense; RmlC, frameshift; RmlD, nonsense; WapR, frameshift; PslA, nonsense), were selected for further characterisation in terms of their growth, virulence in *Galleria mellonella* (*G. mellonella*), biofilm formation and motility (Extended Data Figure E4). Compared to the naive Paer09 bacterium (lag time of 3.5 h), the OD measured growth curves of the RmlC (7.6 h) and RmlD (9.4 h) mutants showed a 116% and 166% prolonged lag time, respectively, while the other mutants showed no remarkable difference. All mutant strains exhibited lower virulence *in vivo*, resulting in a higher larval survival rate after 100 h, when all naive Paer09-infected larvae were dead. While crystal violet staining demonstrated reduced biofilm biomasses for the RmlA, RmlB and RmlC mutants, only the RmlC mutant had a lower biofilm cell count on porous glass beads. The RmlA ($p < 0.01$), RmlC ($p < 0.001$) and RmlD ($p < 0.0001$) mutants showed a reduction in swarming motility. Contrarily, the WapR mutation (frameshift) resulted in a 10.5% ($p < 0.05$) greater swarming motility and both the RmlB and WapR mutants had an increased swimming motility of 32.8% ($p < 0.01$) and 68.0% ($p < 0.0001$), respectively. The other mutants displayed a reduced swimming motility.

Line 449ff. (r.): A representative subset of isolates, each mutated in only one of the three identified genes (RmlC, frameshift; WapR, frameshift; glycoside hydrolase, frameshift), were selected for further characterisation, and the glycoside hydrolase mutant showed no

altered growth curve or biofilm cell count change, but a reduced biofilm biomass and reduced virulence in *G. mellonella* (Extended Data Figure E4).

Extended Data Figure E4:

Fig. E4: Characterisation of phage treated Paer09 mutants.

a, Optical density (OD₆₀₀) measurements of a 24 h incubation of the planktonic naive Paer09 strain (GC, growth control) alongside seven phage treated Paer09 mutants with one altered gene (WapR, frameshift; PslA, nonsense; RmlA, nonsense; RmlB, missense; RmlC, frameshift; RmlD, nonsense; glycoside hydrolase, frameshift) per mutant. Each line represents the average of three biological replicates with three underlying technical replicates corrected by the blank control (TSB). **b**, Percentual survival of *Galleria mellonella* larvae after injection with 10 μ l of bacteria (10³ CFU/ml) (n=10 per strain; Paer09 and mutants stated above) or PBS (control; black line with no deaths; n=10) monitored for 100 h. **c**, Efficacy of plating (EOP) of five phages (FJK, FJK.R9-15, FJK.R9-30, MK, and MK.R3-15) for seven representative Paer09 mutants (mutants stated above) complemented with their respective wild-type gene. The EOP is defined as the concentration ratio of a phage on the complemented mutant (numerator; plasmid with wild-type gene) and the non-complemented mutant (denominator; empty plasmid). An EOP above 10 was considered as “increased” efficiency (green), while an EOP of 0.1 to 10 was ranked as “unchanged” efficiency (grey). A “reduced” efficiency was defined as an EOP between 0.001 and 0.1 (light blue). When no individual plaques were visible or the EOP was equal to or under 0.001 the efficiency was considered “greatly reduced” (dark blue). **d**, Optical density (OD₅₇₀) measurements of crystal violet/ethanol solution obtained from stained bacterial microtiter plate biofilms (Paer09 and mutants stated above). **e**, Bacterial load (CFU/bead) of 24 h pre-established bacterial biofilms (Paer09 and mutants stated above), determined by plating. **f**, Swarming motility of each strain (Paer09 and mutants stated above) on TSA plates (0.5% w/v

agar). **g.** Swimming motility of each strain (Paer09 and mutants stated above) on TSA plates (0.3% w/v agar).

Unless specified otherwise, all experiments were performed in three biological replicates with either three (a and e), two (d) or no (c) technical replicates, except the motility which was determined in four biological replicates (f and g). The error bars represent the standard error of the mean. The statistical analysis was conducted by unpaired two-tailed Student's *t* test and *p*-values were indicated by an asterisk (*, $p < 0.05$; **, $p < 0.01$; ***, $p < 0.001$; ****, $p < 0.0001$).

31. Line 259, indicate the number of bp deleted in the early gene region as well as the number of genes? Any predicted functions in the deleted genes/proteins?

Response to comment: Predicted as “hypothetical proteins” by the RASTtk pipeline of the Bacterial and Viral Bioinformatic Resource Center (BV-BRC) there are no known functions in the deleted genes and proteins.

We have revised the phrase accordingly.

Line 272ff. (r.): By directed evolution, phages FJK.R9-15 and FJK.R9-30 lost a part of the early gene region (FJK.R9-15, 2431 bp, 11 genes; FJK.R9-30, 1348 bp, 5 genes) and present single-nucleotide polymorphisms (SNP) in their genome leading to missense mutations in two genes encoding an amidoligase (gp22) and an L-glutamine-D-fructose-6-phosphate-aminotransferase (gp23), as well as mutations in four genes encoding structural proteins (Figure 4a).

32. Line 261 & 262, what are the gp number of these two enzymes?

Response to comment: The gene product number of the L-glutamine-D-fructose-6-phosphate-aminotransferase is 23 and the amidoligase has the 22.

We have introduced these gp numbers for the clarity of the reader.

Line 272ff. (r.): By directed evolution, phages FJK.R9-15 and FJK.R9-30 lost a part of the early gene region (FJK.R9-15, 2431 bp, 11 genes; FJK.R9-30, 1348 bp, 5 genes) and present single-nucleotide polymorphisms (SNP) in their genome leading to missense mutations in two genes encoding an amidoligase (gp22) and an L-glutamine-D-fructose-6-phosphate-aminotransferase (gp23), as well as mutations in four genes encoding structural proteins (Figure 4a).

33. Line 263, “remarkable” as compared to what?

Response to comment: We apologize for the ‘subjective’ terminology. We have revised the phrase to avoid this.

Line 276f. (r.): Although not in the same position, all SNPs for both evolved phages are in the same genes.

34. Line 272, you indicated above that four structural genes were mutated but you discussed only two here (gp52 and gp62). What about the other two (gp51, gp58?)?

Response to comment: The other two proteins were annotated as a structural protein (gp51) and a needle head protein (gp58). As their discussion would not provide any added value we refer to the corresponding Figure 4a for further details.

35. Line 282, what would those mutations do? Change specificity? Improve efficiency?

Response to comment: We are happy to introduce a hypothesis here.

Line 293ff. (r.): It is therefore likely that gp43 functions as an endosialidase and can specifically degrade bacterial polysialic acid on the phage's path to the bacterial cell membrane⁴⁵, thereby increasing their effectiveness, as more susceptible strains can be infected.

36. Line 285, how was it determined that the evolved phages derived from the ancestral phages MK and JS appear to be a recombination of the two?

Response to comment: The genomes of the ancestral phages were compared to each other and individually to each evolved phage (MK.R3-15, MK.R3-30, MK.R57-15, MK.R57-30, MK.R84-15 and MK.R84-30) using Snippy¹², a tool to identify single nucleotide polymorphisms (SNP), small deletions and insertions (indels). This three-way analysis revealed alternating mutation patterns suggesting the recombination of the evolved phages. Please refer to the below table for an exemplary simplified illustration.

MK vs. MK.R3-15	MK vs. JS	JS vs. MK.R3-15	Interpretation
No hits	NT_Pos: 64/246 AA_Pos: 22/81 (snp, 64C>T; Leu22Phe)	NT_Pos: 64/246 AA_Pos: 22/81 (snp, 64C>T; Leu22Phe)	MK gene in MK.R3-15
NT_Pos: 90/195 AA_Pos: 30/64 (complex, 90_91 del TG ins CA, Val31Ile)	NT_Pos: 90/195 AA_Pos: 30/64 (complex, 90_91 del TG ins CA, Val31Ile)	No hits	JS gene in MK.R3-15
NT_Pos: 32/393 AA_Pos: 11/130 (snp, 32T>C, Leu11Ser)	NT_Pos: 32/393 AA_Pos: 11/130 (snp, 32T>C, Leu11Ser)	No hits	JS gene in MK.R3-15
No hits	NT_Pos: 383/468 AA_Pos: 128/155	NT_Pos: 383/468 AA_Pos: 128/155	MK gene in MK.R3-15

	(snp, 383T>C, Val128Ala)	(snp, 383T>C, Val128Ala)	
--	-----------------------------	-----------------------------	--

("NT_Pos" is the nucleotide position of the variant within the feature / number of nucleotides and "AA_Pos" is the residue position / number of amino acids)

37. Line 307, should that predicted conformational change (line 271-272) be shown here?

Response to comment: The evolved phages FJK.R9-15/30 present a mutation in gp62 where cysteine (amino acid 98) was replaced by phenylalanine. As AlphaFold (structures were visualized with PyMOL) could not predict a change in structure we apologize for this hypothesis. No domains could be identified within the additional α -helix in gp62.

We have revised the phrase accordingly.

Line 283ff. (r.): Compared to TTPA, gp62 contains an additional α -helix close to the b-sheets, in which the mutation occurred, that could have an impact on the enzymatic activity, thereby potentially explaining the increased antibiofilm activity of the evolved phage.

38. Line 314-315, why have you selected those phage-host pairs for this assay?

Response to comment: To investigate the bacterial capacity to escape phage predation and identify mutational changes associated to phage resistance, we decided to work with phages FJK, FJK.R9-15 and FJK.R9-30 in combination with bacterium Paer09. In addition to the arguments for using this combination to test the phages' antibiofilm and antimicrobial biofilm activity (line 161ff.) we could complement these results with their antimicrobial activity against planktonic, non-biofilm, bacteria. Moreover, the observed resistance trade-off associated to Paer09 reinforced this decision.

We have further revised the manuscript accordingly.

Line 836ff. (m.): The co-incubation of the Paer09 strain with either phage FJK, phage FJK.R9-15, phage FJK.R9-30 or the combined phages FJK.R9-30 and MK.R3-15 (cocktail) was performed to induce phage resistance and selected upon in continuation of previous experiments and the observed phage-resistance trade-off.

39. Line 319, what were those trade-off events observed in preliminary tests (data not shown)? Was it the varying degree of phage resistance?

Response to comment: The resistance trade-off events observed in preliminary tests were the resistance trade-off between FJK-phages and MK-phages, with varying degrees of phage resistance illustrated by a negative correlation of susceptibility to phage FJK versus phage MK and MK.R3-15. Please refer to line 401ff. of the results section for

additional details. The preliminary data was reproduced in biological replicates and presented in Figure 5a and Figure 6a.

40. Line 335, I found this very surprising that only one mutation was observed per bacterial strain, especially considering the typical size of a *P. aeruginosa* genome (>6 Mb). Did you completely sequence the 48 genomes? What was the coverage? Did you get only one contig per strain? What about plasmids?

Response to comment: The ancestral Paer09 strain did not contain any plasmids. All phage-treated Paer09 isolates (n=48, individual phages; n=18, phage cocktail) were sequenced with Illumina. This identified one of seven genes, likely conferring resistance to phage predation, for each isolate. Five of those mutations (WapR, RmlA, RmlB, RmlC and RmlD) are involved in the O-antigen attachment, indicating this to be the potential receptor for FJK-phages. Thus, a loss of O-antigen through a single mutation would suffice to cause resistance to FJK-phages. The coverage of the Illumina reads was between 42.4 and 74.8 for the 66 isolates. The 66 genomes were not assembled for this experiment as Snippy¹² can directly compare the trimmed Illumina reads to the ancestral Paer09 GenBank file. We consider this approach more informative as it is critical to avoid loss of information.

41. Line 337, as indicated above (major point) to really confirmed that the mutations were indeed involved in the phage resistance phenotype, additional genetic experiments should have been performed (ie introduce the wild-type gene into the resistant clone and see if the phage sensitivity profile is restored). Also, were these mutations confirmed by PCR and Sanger sequencing directly on the respective mutants?

Response to comment: To further validate our results, we selected a representative subset of seven phage resistant bacterial mutants, one for each mutation observed (WapR, PslA, RmlA, RmlB, RmlC, RmlD and glycoside hydrolase). For these strains the mutations were confirmed by PCR and Sanger sequencing and a complementation assay performed. Please refer to line 910ff. of the methods section for additional details.

To summarize, all mutations could be validated and the plasmid complementation with the wild-type gene led to a reversal of the resistance trade-off, resulting in greater susceptibility (increased EOP) to FJK, FJK.R9-15 and FJK.R9-30 and loss of susceptibility (decreased EOP) to MK and MK.R3-15. This was most prominent in the RmlA, RmlC and WapR mutants (Extended Data Figure E4c).

42. Lines 362 & 379, how do you explain the absence of no apparent mutation and the resistance phenotype? It raises some question regarding the sequencing and/or the analyses.

Response to comment: After three days of co-incubation of the planktonic bacterium Paer09 with either ancestral phage FJK or one of the evolved phages (FJK.R9-15/30), 48 bacteria were isolated. From these, total bacterial genomic DNA was extracted and sequenced with Illumina. Please refer to line 845ff. and line 737ff. of the methods section

for additional details on the isolation and whole genome sequencing. Among these isolates, five (10.4%) showed no mutation and an identical phage susceptibility profile to the nonexposed Paer09 control (Figure 5a). In conclusion, five isolates managed to avoid phage contact, by for example remaining in an ecological, spatial refuge such as bacterial growth on the microplate well wall and did not need to adapt to phage predation, resulting in no mutations and no change of their phage susceptibility profile.

We have further revised the manuscript accordingly.

Line 410ff. (r.): Among the 48 bacterial isolates, five isolates (10.4%; no apparent mutations) showed the same phage susceptibility profile as the control, except one who had an increased efficiency of plating for phage FJK.R9-30 (Figure 5a). These five isolates and the control showed resistance to phage MK, while no isolate was resistant to the evolved phage MK.R3-15.

43. Lines 392-397 & 465, I think that a better comparison would be with a cocktail of the two ancestral phages (or at least FJK which is capable of infecting with the naive Paer09 strain). Was this tested? That a cocktail of phages performed better than a single phage is not that surprising. But are the evolved phages performing better than the ancestral phage?

Response to comment: For the antimicrobial activity, 24 h pre-established Paer09 biofilms or planktonic Paer09 under isothermal microcalorimetry or optical density monitoring, respectively, were either co-incubated with individual phages or the phage cocktail. The experimental setup of the study was designed so that each successive experiment builds on the previous results. In the initial direct comparison, the evolved phage (FJK.R9-30) showed a significantly higher heat flow suppression and lag time (Figure 3b, c, d and Figure 5d) than the ancestral phage (FJK). Building on this, we tested the evolved phages individually and as a 1:1 cocktail against Paer09. The cocktail demonstrates a significantly greater heat flow suppression and lag time (Figure 6d, e, f, i).

To summarize, the evolved phages outperformed the unevolved, ancestral phages and were in turn outperformed by the phage cocktail combining the two evolved phages. Although we do not have experimental confirmation, we strongly feel that exchanging the evolved phages for the unevolved ancestral phages would not improve the efficacy of the cocktail, the intended outcome of the experiment.

44. Line 411, how many isolates were retrieved? 18 (line 415)?

Response to comment: After the co-incubation of the phage cocktail with Paer09, 18 isolates were retrieved.

We have revised the phrase accordingly.

Line 445ff. (r.): Isolates retrieved after the co-incubation with the phage cocktail (n=18) revealed mutations in three proteins (RmlC, WapR and glycoside hydrolase) (with only one altered gene per mutant) (Figure 6a).

45. *Lines 500-526, in my opinion, this section was a bit wordy and perhaps could be reduced in size and should focus on the next steps to improve the proposed platform rather than the clinical aspects. Would it worked with other phages, as one of the phages (JS) doesn't seem to have evolved that much? It is also unclear if it would be applicable to other bacteria (Gram-positive for example).*

Response to comment: We concur, this section would benefit from a rationalisation. Regarding the broader applicability of our new evolution platform, we have revised the manuscript by adding focus on the next improvement steps and streamlining the clinical perspective.

Line 529ff. (d.): However, as the road from proof of concept to bedside is long, several hurdles remain. These include further validations of the evolution platform with different phage-bacteria systems, to reiterate the combined improvement of the infectivity parameters. Not limited to Gram-negative bacteria, the evolution assay could also find applicability to improve phage efficacy against Gram-positive bacteria and pathogens relevant in agriculture, aquaculture, and food safety to reduce the amount of antibiotics currently employed^{64,65}. At the same time, the platform itself could be further streamlined by optimizing the number and length of each evolution cycle, as it is a labour-intensive approach impeding generalized applicability. Potential for optimization includes the combination of more than two highly similar phages to increase the number of combinatory possibilities or have the phages undergo untargeted mutagenesis prior to selecting for antibiofilm efficacy. Improving the phage infectivity parameters in sequence rather than in parallel could result in varying adaptive trajectories and aid in the identifying of mutational sites of evolutionary adaptation.

Line 567ff. (d.): Taken together, our evolution platform could provide insights into evolutionary bacteria-phage interactions, defence strategies and interdependencies as well as strengthen phage therapy as a treatment option to improve the outcome of multidrug-resistant bacterial infections with trained resistance-adapted phage cocktails. For personalized medical approaches, phages could be specifically trained against patient strains before administration. All the while, the evolved phages would still be considered as natural, non-genetically engineered entities, limiting the risks if released into the environment and allowing for easier approval⁶⁶.

46. *Line 549, is it the first time that these four phages are described?*

Response to comment: Yes, the four ancestral phages (MK, JS, FJK, FIM) were isolated for this study and described here for the first time.

47. Line 562, still unclear how these 31 plaques were isolated, and the host used. You mentioned plaque morphology as a selection criterium, did the selected phages had a different plaque size compared to the ancestral phages?

Response to comment: Thank you. We have attempted to clarify this further for the reader's convenience. Figure E3 included in the file "NCOMMS-23-23549A_DBPR_Extended_Data" represents the isolation process of the evolved phages. The phage mixtures of rounds 15 and 30 were spotted on all eight *P. aeruginosa* evolution strains (PAO1, Paer03, Paer09, Paer33, Paer57, Paer60, Paer84 and Paer85) and based on the host strain and plaque morphology all differing phage plaques (n=31) were picked, amplified, and sequenced to determine their descent. Parameters of the qualitative plaque assessment were the plaques' size, contour, and turbidity/clarity.

Figure E3 has been referenced to clarify this:

Line 155f. (r.): Based on plaque morphology and host strain, we isolated 31 individual evolved phages from the phage mixtures (17 from round 15 and 14 from round 30) (Extended Data Figure E3).

Line 608ff. (m.): Based on qualitative plaque assessment and host strain we identified 31 phages that were purified and produced on their isolation strains as described above (Extended Data Figure E3).

Extended Data Figure E3:

Fig. E3: Overview of the isolation of evolved bacteriophages after rounds 15 and 30.

a, Tenfold serial dilutions of the bacteriophage (phage) mixtures after round 15 and 30 were individually spotted (5 µl) on soft agar overlays for each individual bacterial strain in the evolution assay (PAO1, Paer03, Paer09, Paer33, Paer57, Paer60, Paer84 and Paer85). **b**, Based on the host strain and qualitative plaque assessment (plaques' size, contour, and turbidity/clarity) 17 evolved phages were picked from round 15 spot assays and 14 from round 30. The phages were purified by four consecutive single-plaque-passages. **c**, Each isolated phage was produced from a single plaque using either a liquid or solid propagation method. **d**, Phage genomes were extracted, sequenced with Illumina, and assembled before the corresponding ancestral phage was identified using BLASTn.

48. Line 571, I didn't see any TEM figure in the manuscript. Morphotypes are mentioned in Data S2. Capsid and tail sizes should at least be mentioned in Data S2.

Response to comment: The morphology of the ancestral phages was visualized by transmission electron microscopy (TEM) and presented in Figure E1 included in the file "NCOMMS-23-23549A_DBPR_Extended_Data".

We have revised the manuscript and Supplementary Data S2 accordingly.

Line 630ff. (m.): For each phage, using ImageJ v1.54g¹³ four particles were used to calculate the tail length, tail width and average capsid size in three axes (Supplementary Data S2).

49. *Line 591, phages were diluted in which buffer?*

Response to comment: Thank you for highlighting this omission. SM-buffer was used for storage and dilution of the phages.

We have revised the manuscript accordingly.

Line 602f. (m.): Ultimately, phage lysates were concentrated and purified by PEG 8000 precipitation⁷³ before storage in SM-buffer at 4 °C for further use.

Line 605ff. (m.): Evolved phages after round 15 and 30 of the evolution assay were isolated by spotting 5 µl tenfold serial dilutions in SM-buffer of the corresponding phage mixture on soft agar overlays for each individual bacterial strain in the evolution assay (PAO1, Paer03, Paer09, Paer33, Paer57, Paer60, Paer84 and Paer85).

Line 610ff. (m.): Of those, 10 (MK.R3-15, MK.R3-30, MK.R57-15, MK.R57-30, MK.R84-15, MK.R84-30, FIM.R60-15, FIM.R60-30, FJK.R9-15 and FJK.R9-30), representing phages descended from the different ancestral phages determined by BLASTn v2.13.0⁷⁹, were concentrated and purified by PEG 8000 precipitation⁷⁸ before storage in SM-buffer at 4 °C for further use.

Line 641ff. (m.): Phage solutions prepared as tenfold SM-buffer dilutions (10^{-1} – 10^{-8}) in 96-well microplates (PN 353072; Corning Inc., Corning, USA) were spotted (5 µl) on the overlays and incubated overnight at 37 °C.

50. *Line 601, how many beads per well?*

Response to comment: There was one porous glass bead per well.

We have revised the phrase accordingly.

Line 651ff. (m.): Bacterial biofilms were formed on autoclaved 4 mm sintered porous glass beads (ROBU® Glasfilter-Geräte GmbH, Hattert, Germany) by incubation in a sterile 24-well plate (Corning Inc., Corning, USA). Each bead was individually incubated in a well containing 1 ml TSB inoculated with 1:100 dilution from a one-time use glycerol stock of *P. aeruginosa* and kept at 37 °C and 150 rpm orbital shaking for 24 h under humidity conditions.

51. *Line 607, the SEM data were not discussed in the manuscript.*

Response to comment: Scanning electron microscopy (SEM) images of the pre-established 24-h-biofilm of each *P. aeruginosa* strain included in the evolution assay were used to classify the biofilm density and thus the diversity of biofilm formation traits. The images were indirectly referenced in line 88ff. as part of Figure E1 included in the file “NCOMMS-23-23549A_DBPR_Extended_Data”.

52. *Line 626, which host was used to amplify the ancestral phages?*

Response to comment: The ancestral phages were propagated on their isolation strains.

We have clarified this in the manuscript.

Line 600ff. (m.): Next, each isolated phage was produced from a single plaque on their isolation strain using either a liquid⁷⁶ or solid⁷⁷ propagation method.

53. *Line 632, how was the phage genera determined?*

Response to comment: The genera of our ancestral phages were determined based on genome synteny, when the intergenomic distance between the evolved phage and its closest reference phage retrieved using BLASTn was above 70% as determined by VIRIDIC¹⁴.

We have revised the manuscript accordingly.

Line 765ff. (m.): The genera of the ancestral phages were determined by an intergenomic distance between them and their most similar reference phages above 70%, as determined by VIRIDIC¹⁴.

54. *Line 679, how many strains were sequenced? Did you assemble each of them into one contig? I am asking because you could not identify mutation in some of the phage-resistant derivative. Is there any plasmids in these strains?*

Response to comment: Whole genome sequencing was performed for all strains in the biobank (n=80) and each phage-treated Paer09 isolates (n=48, individual phages; n=18, phage cocktail). In reference to the above comments, the five isolates showing no apparent mutation had an identical phage susceptibility profile to the not exposed Paer09 control, excluding sequencing errors. The 66 genomes were not assembled for this experiment as Snippy¹² can directly compare the trimmed Illumina reads to the ancestral Paer09 GenBank file. We consider this approach more informative as it is critical to avoid loss of information.

55. *Line 705, how were the ends of the phage genomes determined?*

Response to comment: To determine the genome ends of the phages, the genome was aligned to its most similar reference phage genome using MEGA 11. Please refer to line 764ff. of the methods section for additional details.

56. *Line 711, GenBank*

Response to comment: Thank you for noticing this typo.

Line 770ff. (m.): GenBank files of the ancestral phages were finalized using Artemis v18.1.0¹⁰⁴ and deposited under the accession codes listed in Supplementary Data S2.

57. *Supplementary Data S2, not sure I understand what the percentage of coverage is and what the percentage of sequence means here. Should they be 100%?*

Response to comment: The query coverage is the percent of the query sequence length that is included in alignments against the comparative sequence. The percent identity is the percent of nucleotides that are identical between the aligned query and comparative sequence. We have revised the Supplementary Data S2 included in the file “NCOMMS-23-23549A_DBPR_Supplementary_Information” accordingly.

58. *The reference section will need some work. Hard to believe that the authors really looked at that section...*

Response to comment: Thank you very much for your detailed feedback on the bibliography. We have revised the manuscript according to your comments 59 to 67. Please find all revised references highlighted in the manuscript without an additional copy in this response.

59. *Refs 3, 6, 7, 14, 18, 19, 20, 23, 25, 29, 30, 33, 39, 40, 43, 44, 45, 46, 51, 52, 54, 55, 59, 62, 75, 86, 89, 94, 102: Remove capital letters in the titles.*

Response to comment: We have revised the references accordingly.

60. *Refs 5, 6, 7, 11, 13, 14, 15, 17, 18, 19, 21, 22, 24, 25, 26, 35, 36, 37, 40, 42, 48, 49, 50, 51, 52, 53, 55, 60, 62, 63, 70, 105: Bacterial name should be in italic.*

Response to comment: We have revised the references accordingly.

61. *Refs 5, 14, 15, 17, 19, 22, 26, 31, 35, 41, 44, 45, 48, 53, 54, 58, 59, 62, 65, 69, 70, 74, 83, 87, 91, 103, 104: Journal name is abbreviated while not in the others.*

Response to comment: We have revised the references accordingly.

62. *Refs 32, 76: Title is missing.*

Response to comment: We have revised the references accordingly.

63. Ref 78: *This has been published, see PMID: 28594827*

Response to comment: We have revised the reference accordingly.

64. Ref 83: *W259*

Response to comment: We have revised the reference accordingly.

65. Refs 23, 29, 33, 39, 52, 86: *page numbers or enumbers are missing.*

Response to comment: We have revised the references accordingly.

66. Ref 97: *This has been published, see PMID: 35637307*

Response to comment: We have revised the reference accordingly.

67. Ref 100: *Title and Year are missing.*

Response to comment: We have revised the reference accordingly.

References

- 1 Akusobi, C., Chan, B. K., Williams, E. S. C. P., Wertz, J. E. & Turner, P. E. Parallel evolution of host-attachment proteins in phage PP01 populations adapting to *Escherichia coli* O157:H7. *Pharmaceuticals* **11**, 60 (2018). <https://doi.org/10.3390/ph11020060>
- 2 Sackman, A. M. *et al.* Mutation-driven parallel evolution during viral adaptation. *Molecular Biology and Evolution* **34**, 3243-3253 (2017). <https://doi.org/10.1093/molbev/msx257>
- 3 Perry, E. B., Barrick, J. E. & Bohannon, B. J. The molecular and genetic basis of repeatable coevolution between *Escherichia coli* and bacteriophage T3 in a laboratory microcosm. *PLoS One* **10**, e0130639 (2015). <https://doi.org/10.1371/journal.pone.0130639>
- 4 Bull, J. J. *et al.* Exceptional convergent evolution in a virus. *Genetics* **147**, 1497-1507 (1997). <https://doi.org/10.1093/genetics/147.4.1497>
- 5 Zimmerli, W., Frei, R., Widmer, A. F. & Rajacic, Z. Microbiological tests to predict treatment outcome in experimental device-related infections due to *Staphylococcus aureus*. *Journal of Antimicrobial Chemotherapy* **33**, 959-967 (1994). <https://doi.org/10.1093/jac/33.5.959>
- 6 Konrat, K. *et al.* The Bead Assay for Biofilms: A Quick, Easy and Robust Method for Testing Disinfectants. *PLOS ONE* **11**, e0157663 (2016). <https://doi.org/10.1371/journal.pone.0157663>

- 7 Brockhurst, M. A., Buckling, A. & Rainey, P. B. Spatial heterogeneity and the stability of host-parasite coexistence. *Journal of Evolutionary Biology* **19**, 374-379 (2006). <https://doi.org/10.1111/j.1420-9101.2005.01026.x>
- 8 Burrowes, B. H., Molineux, I. J. & Fralick, J. A. Directed *in vitro* evolution of therapeutic bacteriophages: the Appelmans protocol. *Viruses* **11**, 241 (2019). <https://doi.org/10.3390/v11030241>
- 9 Page, A. J. *et al.* Roary: rapid large-scale prokaryote pan genome analysis. *Bioinformatics* **31**, 3691-3693 (2015). <https://doi.org/10.1093/bioinformatics/btv421>
- 10 Stamatakis, A. RAxML version 8: a tool for phylogenetic analysis and post-analysis of large phylogenies. *Bioinformatics* **30**, 1312-1313 (2014). <https://doi.org/10.1093/bioinformatics/btu033>
- 11 Letunic, I. & Bork, P. Interactive Tree Of Life (iTOL) v4: recent updates and new developments. *Nucleic Acids Research* **47**, W256-W259 (2019). <https://doi.org/10.1093/nar/gkz239>
- 12 Seemann, T. (GitHub, <https://github.com/tseemann/snippy>, 2015).
- 13 Schneider, C. A., Rasband, W. S. & Eliceiri, K. W. NIH Image to ImageJ: 25 years of image analysis. *Nature Methods* **9**, 671-675 (2012). <https://doi.org/10.1038/nmeth.2089>
- 14 Moraru, C., Varsani, A. & Kropinski, A. M. VIRIDIC-A novel tool to calculate the intergenomic similarities of prokaryote-infecting viruses. *Viruses* **12**, 1268 (2020). <https://doi.org/10.3390/v12111268>

Reviewer #3

This study set up an antibiofilm evolution assay to train the phages and obtained evolved phages showing an expanded host spectrum, and improved antimicrobial efficacy. This is an interesting and comprehensive study.

However, it is not a conceptual advance in this field as the evolution method is not novel, the overall results are not surprising to me as these mutations (FIG4) are quite common in phage-host coevolution experiments but the molecular mechanisms are not investigated, so it is descriptive. Most importantly, the final phage cocktail seems not to have a major impact on the antibiofilm efficacy.

[Major comments]

1)The rationale for using an isothermal microcalorimeter to monitor the evolution process is not clearly described, and this is not a commonly used approach in monitoring phage evolution. What's the correlation of heat flow with the anti-biofilm efficacy? There are various methods to monitor biofilm, but why only use an isothermal microcalorimeter or QPCR to monitor it? since these methods are not clearly described and the controls are not well established, I think the results might not be solid.

Response to comment: Thank you for this comment. We in fact regard the isothermal microcalorimeter approach as one of the key innovations in this platform, driven by in-house experience, so we consider it important to be clearly represented. With research toward the targeted improvement of phages' ability to combat biofilms being scarce, our study provides an *in vitro* evolution assay that improves multiple phage parameters in parallel including their antibiofilm efficacy. Our rationale was to expose phages in serial passage to a robust *in vitro* biofilm formed on a porous glass beads, which provide the bacteria a large three-dimensional surface for biofilm to form (Extended Data Figure E1)^{1,2}.

To be able to monitor and evaluate our pre-established biofilm beads during the evolution in real-time, we decided to use isothermal microcalorimetry³⁻⁷.

Isothermal microcalorimetry enables real-time monitoring of the heat flow (μW) and cumulative amount of heat (J) produced from biological processes. Its advantages for our study lie within the high sensitivity (0.2 μW), high accuracy ($< 5\%$), continuous real-time data, simplicity, and passivity⁸. This method has been widely used in life sciences⁹⁻¹², pharmaceutical development¹³⁻¹⁵, clinical analysis^{16,17} and also in phage research¹⁸⁻²⁰. The heat flow, being proportional to the rate of the observed process, allows for a highly sensitive assessment of bacterial cell activity in the pre-established biofilm beads. Bacterial growth is indicated by an increased heat flow and growing heat production while suppression and eradication of set bacteria results in the delayed or absence of heat production. The antimicrobial biofilm efficacy represents this surrogate bacterial cell activity monitored by isothermal microcalorimetry.

To investigate our phages' antibiofilm efficacy against the pre-established biofilm beads, we could not work with colouring assays (e.g.: crystal violet staining) and decided to use real-time quantitative polymerase chain reaction (RT-qPCR), as an up-to-date technique. Other than plating biofilm bacteria after sonication, qPCR results are not distorted by remaining phages lysing bacteria once plated. One limitation is that live and dead bacteria not dip-washed away are both counted, resulting in more conservative results of the assay.

qPCR is a well-established method, amplifying the target DNA to quantify the initial input number of DNA copies, each one representing a biofilm bacterium. This method has been widely used for the determination of bacterial cell counts in biofilms²¹⁻²⁷.

We have addressed these points specifically within the manuscript:

Line 543ff. (d.): Isothermal microcalorimetry allowed us to continuously monitor the phage-bacterial biofilm interaction in real-time with a high sensitivity and accuracy^{3,19} while RT-qPCR helped us to precisely quantify the antibiofilm degradation capabilities of our phages, providing a starting point for further usage of this technique. As RT-qPCR does not distinguish between live and dead bacteria, the results provided represent a more conservative antibiofilm efficiency, considering the possibility that DNA from dead bacteria not dip-washed away is also quantified.

Line 671ff. (m.): The *in vitro* phage evolution assay to improve multiple phage parameters in parallel, consisted of a serial passaging approach with thirty consecutive rounds, inspired by the directed evolution approach of the Appelmans protocol⁸¹. Adaptations to accommodate bacterial biofilms included the use of microcalorimetric real-time monitoring, revised active sample criteria and performance-dependant phage-mixture-dilutions.

2)The setup of this evolution experiment is not clearly described and is very similar to The Appelmans Protocol. But I am not convinced that using beads could select for phage with enhanced anti-biofilm ability. Is it better than the Appelmans Protocol??

Response to comment: Thank you for this comment. In short, we developed an evolutionary serial passage assay (Figure 1), that utilizes a directed evolution approach, to simultaneously improve several phage infectivity parameters. This assay was implemented using four lytic *P. aeruginosa* phages, which were trained on eight *P. aeruginosa* strains (Figure 2a; Extended Data Figure E1; Supplementary Data S1). During each round, pre-established (24 h) biofilms of ancestral bacteria were incubated with a mixture of the phages under isothermal microcalorimetric heat production control. After each passage, all samples showing a heat reduction greater than 75% (compared to growth control) were pooled and the undiluted phage samples (always included) together into the new phage mixture. In total, 30 rounds of evolution were performed. Please refer to Figure 1 and lines 671ff. of the methods for a detailed description of the experimental setup.

The revised methods section highlights why we adopted the method, refers to the previously established Appelmans protocol and points out adaptations made to accommodate biofilms. All phages and bacteria used are named and selection criteria for inclusion and exclusion are indicated (line 675ff., line 683ff.). The methods further describe the mediums used (line 696ff.), the controls employed (line 699ff.), the characteristics of each ampule monitored (line 696ff.), the experimental process (line 675ff.) and the monitoring in a 48-channel isothermal microcalorimeter (line 691ff). Further, active samples were defined and sample processing after each round described (line 703ff.). Criteria for the determination of the phage-mixture-dilution of each round were indicated (line 714ff.). Limitations of the evolution assay were highlighted in the discussion (line 510ff.).

The Appelmans protocol²⁸ consisted of a directed evolution approach that investigated and could expand the phages' host range on planktonic bacteria. In our study, we implemented an *in vitro* directed evolution approach against biofilm bacteria for the combined improvement of the phages' host spectrum, antimicrobial and antibiofilm capabilities. To exert strong selection pressure on phages to undergo improved antibiofilm activity, our rationale was to expose phages in serial passage to a robust *in vitro* biofilm formed on a porous glass beads, which provide the bacteria a large three-dimensional surface for biofilm to form (Extended Data Figure E1)^{1,2}. The phages that best reproduce within this environment are selected. Across all tested strain-phage combinations, we could show that the phages evolved for thirty rounds demonstrated a greater antibiofilm activity with lower biofilm cell counts compared to their unevolved ancestors. Please refer to Figures 3a, e, i and line 229ff. of the results for further details. Although the Appelmans protocol consisted of a serial passaging approach, it selected for phages' host range expansion on planktonic bacteria, which is why any direct comparison would be speculative, as this was neither the goal of our study nor do we have sufficient information for a definitive answer.

To emphasize these points within the manuscript, we have introduced the following changes:

Line 510ff. (d.): Nevertheless, the selective pressure to adapt to pre-established bacterial biofilms within our evolution assay, might have been weakened by the release of planktonic bacteria into the surrounding medium, only avoidable within dynamic biofilm models.

Line 671ff. (m.): The *in vitro* phage evolution assay to improve multiple phage parameters in parallel, consisted of a serial passaging approach with thirty consecutive rounds, inspired by the directed evolution approach of the Appelmans protocol⁸¹. Adaptations to accommodate bacterial biofilms included the use of microcalorimetric real-time monitoring, revised active sample criteria and performance-dependant phage-mixture-dilutions.

Line 696ff. (m.): Contained in airtight 4 ml disposable glass vials (Waters GmbH, Eschborn, Germany), each sample comprised 450 μ l TSB, one in PBS dip-washed 24-h biofilm glass bead and 50 μ l of the corresponding phage mixture dilution in SM-buffer.

3) *The mutation of the phage structural proteins is quite common in other phage-host co-evolution experiments, including the tail fiber gene, and polysaccharide depolymerase gene. However, the function and the mechanisms of these mutations are not validated, so it is descriptive.*

Response to comment: In our study, we integrated the combined improvement of the phages' host spectrum, antimicrobial and antibiofilm capabilities by implementing an *in vitro* directed evolution approach against biofilm bacteria. With a focus on our evolution assay, we described the phenotypical (expanded host range, improved antimicrobial activity and enhanced antibiofilm efficacy) and genotypical outcomes for our evolved phages, providing further correlative evidence.

Beyond the aim of our study, additional research is warranted to identify additional phage proteins and experimentally study phenotypic-genotypic correlations in altered proteins by morphology analysis, one-step growth curve and determination of the minimum inhibitory multiplicity of infection (miMOI) and single-cell burst size.

4) *Most importantly, from FIG6C, seems like this evolved phage could not significantly inhibit biofilm, and it seems not likely to enhance the antibiofilm activity in phage therapy. Because in your evolution experiment, phages might infect the released PA strains and not only survive by infecting the bacteria in the biofilm. So, the results seem not surprising to the phage field.*

Response to comment: The phages' antibiofilm activity was investigated by co-incubation of pre-established biofilms with the individual phages or the phage cocktail and quantification of the bacterial biofilm cell count reduction using real-time quantitative polymerase chain reaction (RT-qPCR). The use of real-time quantitative polymerase chain reaction (RT-qPCR) represents a conservative quantification of antibiofilm efficacy, as live and dead bacteria not dip-washed away are not differentiated. The results of the antibiofilm activity are presented in Figure 3, Figure 6, and the corresponding result sections and were achieved against a robust *in vitro* biofilm formed on a porous glass bead, providing a large three-dimensional surface for biofilm to form (Extended Data Figure E1)^{1,2}.

Figure 6c illustrates the comparison in antibiofilm efficacy of the phage cocktail (FJK.R9-30 + MK.R3-15) to the individual evolved phages (FJK.R9-30). In summary, while the phage cocktail does not have a major impact on the short-term (within the initial 24 h) antibiofilm efficacy compared to the individual phage FJK.R9-30, it exhibits a higher suppressive activity at prolonged incubation times (MOI of 0.0001; Figure 6h and f), as bacteria cannot develop full resistance to the phage cocktail.

For the individual evolved phages, Figures 3a, e and i demonstrate that among all tested strain-phage combinations, the phages evolved for thirty rounds had a greater antibiofilm activity with lower biofilm cell counts compared to their unevolved ancestors. Moreover,

the phages could significantly reduce the biofilm in comparison to the growth controls (Supplementary Table S3), which highlights the intended outcome of the evolution assay. Although, based on our study, we would postulate that due to the observed increased biofilm reduction capacity, our evolved phages should improve phage treatment regimens against biofilm-associated infections, this was neither the goal of our study nor do we have sufficient information for a definite answer.

Besides the use of dynamic models (e.g.: flow chambers, microfluidics), the release of bacteria into the surrounding medium cannot be avoided with static mature biofilm models.

We have revised the manuscript accordingly.

Line 510ff. (d.): Nevertheless, the selective pressure to adapt to pre-established bacterial biofilms within our evolution assay, might have been weakened by the release of planktonic bacteria into the surrounding medium, only avoidable within dynamic biofilm models.

5)The antimicrobial and antibiofilm efficacy of the phage and evolved phages should be tested using an animal model to complement the in vitro experiments.

Response to comment: Based on our discussions with the editor, validation in an animal model is not expected.

References

- 1 Zimmerli, W., Frei, R., Widmer, A. F. & Rajacic, Z. Microbiological tests to predict treatment outcome in experimental device-related infections due to *Staphylococcus aureus*. *Journal of Antimicrobial Chemotherapy* **33**, 959-967 (1994). <https://doi.org:10.1093/jac/33.5.959>
- 2 Konrat, K. *et al.* The Bead Assay for Biofilms: A Quick, Easy and Robust Method for Testing Disinfectants. *PLOS ONE* **11**, e0157663 (2016). <https://doi.org:10.1371/journal.pone.0157663>
- 3 Tkhilaishvili, T. *et al.* Real-time assessment of bacteriophage T3-derived antimicrobial activity against planktonic and biofilm-embedded *Escherichia coli* by isothermal microcalorimetry. *Research in Microbiology* **169**, 515-521 (2018). <https://doi.org:10.1016/j.resmic.2018.05.010>
- 4 Butini, M. E. *et al.* Real-time antimicrobial susceptibility assay of planktonic and biofilm bacteria by isothermal microcalorimetry. *Advances in Microbiology, Infectious Diseases and Public Health: Volume 13*, 61-77 (2019). https://doi.org:10.1007/5584_2018_291
- 5 Wang, L., Tkhilaishvili, T., Bernal Andres, B., Trampuz, A. & Gonzalez Moreno, M. Bacteriophage-antibiotic combinations against ciprofloxacin/ceftriaxone-resistant *Escherichia coli* *in vitro* and in an experimental *Galleria mellonella* model.

- International Journal of Antimicrobial Agents* **56**, 106200 (2020).
<https://doi.org/10.1016/j.ijantimicag.2020.106200>
- 6 Tkhilaishvili, T., Wang, L., Perka, C., Trampuz, A. & Gonzalez Moreno, M. Using bacteriophages as a trojan horse to the killing of dual-species biofilm formed by *Pseudomonas aeruginosa* and methicillin resistant *Staphylococcus aureus*. *Frontiers in Microbiology* **11**, 695 (2020). <https://doi.org/10.3389/fmicb.2020.00695>
- 7 Tkhilaishvili, T., Wang, L., Tavanti, A., Trampuz, A. & Di Luca, M. Antibacterial efficacy of two commercially available bacteriophage formulations, staphylococcal bacteriophage and PYO bacteriophage, against methicillin-resistant *Staphylococcus aureus*: prevention and eradication of biofilm formation and control of a systemic infection of *Galleria mellonella* larvae. *Frontiers in Microbiology* **11**, 110 (2020). <https://doi.org/10.3389/fmicb.2020.00110>
- 8 Braissant, O., Wirz, D., Göpfert, B. & Daniels, A. U. Use of isothermal microcalorimetry to monitor microbial activities. *FEMS Microbiology Letters* **303**, 1-8 (2010). <https://doi.org/10.1111/j.1574-6968.2009.01819.x>
- 9 Abderrahmane, A., Yi, L., Wen-Ying, G., Ping, S. & Song-Sheng, Q. Microcalorimetric studies on the promoter function in *E. coli* TG1 from *P. maltophilia* AT18 chromosome DNA. *Journal of Thermal Analysis and Calorimetry* **68**, 909-916 (2002). <https://doi.org/10.1023/A:1016138505617>
- 10 Bokhari, M. H. *et al.* Isothermal microcalorimetry measures UCP1-mediated thermogenesis in mature brite adipocytes. *Communications Biology* **4**, 1108 (2021). <https://doi.org/10.1038/s42003-021-02639-4>
- 11 Braissant, O. *et al.* Isothermal microcalorimetry accurately detects bacteria, tumorous microtissues, and parasitic worms in a label-free well-plate assay. *Biotechnology Journal* **10**, 460-468 (2015). <https://doi.org/10.1002/biot.201400494>
- 12 Oliva, A. *et al.* Activities of fosfomycin and rifampin on planktonic and adherent *Enterococcus faecalis* strains in an experimental foreign-body infection model. *Antimicrobial Agents and Chemotherapy* **58**, 1284-1293 (2014). <https://doi.org/10.1128/aac.02583-12>
- 13 O'Neill, M. A. & Gaisford, S. Application and use of isothermal calorimetry in pharmaceutical development. *International Journal of Pharmaceutics* **417**, 83-93 (2011). <https://doi.org/10.1016/j.ijpharm.2011.01.038>
- 14 Gaisford, S. & Buckton, G. Potential applications of microcalorimetry for the study of physical processes in pharmaceuticals. *Thermochimica Acta* **380**, 185-198 (2001). [https://doi.org/10.1016/S0040-6031\(01\)00669-4](https://doi.org/10.1016/S0040-6031(01)00669-4)
- 15 Di Luca, M. *et al.* in *Advances in Microbiology, Infectious Diseases and Public Health: Volume 9* (ed Gianfranco Donelli) 1-27 (Springer International Publishing, 2018).
- 16 Cichos, K. H. *et al.* Isothermal microcalorimetry improves the time to diagnosis of fracture-related infection compared with conventional tissue cultures. *Clinical Orthopaedics and Related Research®* **480**, 1463-1473 (2022). <https://doi.org/10.1097/corr.0000000000002186>
- 17 Braissant, O., Wirz, D., Göpfert, B. & Daniels, A. U. Biomedical use of isothermal microcalorimeters. *Sensors* **10**, 9369-9383 (2010). <https://doi.org/10.3390/s101009369>
- 18 Molendijk, M. M. *et al.* Microcalorimetry: A novel application to measure *in vitro* phage susceptibility of *Staphylococcus aureus* in human serum. *Viruses* **15**, 14 (2023). <https://doi.org/10.3390/v15010014>
- 19 Sigg, A. P. *et al.* A method to determine the efficacy of a commercial phage preparation against uropathogens in urine and artificial urine determined by isothermal microcalorimetry. *Microorganisms* **10**, 845 (2022). <https://doi.org/10.3390/microorganisms10050845>

- 20 Guosheng, L. *et al.* Study on interaction between T4 phage and *Escherichia coli* B by microcalorimetric method. *Journal of Virological Methods* **112**, 137-143 (2003). [https://doi.org:10.1016/s0166-0934\(03\)00214-3](https://doi.org:10.1016/s0166-0934(03)00214-3)
- 21 Suzuki, N., Yoshida, A. & Nakano, Y. Quantitative analysis of multi-species oral biofilms by TaqMan real-time PCR. *Clinical Medicine & Research* **3**, 176-185 (2005). <https://doi.org:10.3121/cmr.3.3.176>
- 22 Yasunaga, A. *et al.* Monitoring the prevalence of viable and dead cariogenic bacteria in oral specimens and *in vitro* biofilms by qPCR combined with propidium monoazide. *BMC Microbiol* **13**, 157 (2013). <https://doi.org:10.1186/1471-2180-13-157>
- 23 Gomez, E. *et al.* Prosthetic joint infection diagnosis using broad-range PCR of biofilms dislodged from knee and hip arthroplasty surfaces using sonication. *Journal of Clinical Microbiology* **50**, 3501-3508 (2012). <https://doi.org:10.1128/jcm.00834-12>
- 24 Magajna, B. & Schraft, H. Evaluation of propidium monoazide and quantitative PCR to quantify viable *Campylobacter jejuni* biofilm and planktonic cells in log phase and in a viable but nonculturable state. *Journal of Food Protection* **78**, 1303-1311 (2015). <https://doi.org:10.4315/0362-028x.Jfp-14-583>
- 25 Ammann, T. W., Bostanci, N., Belibasakis, G. N. & Thurnheer, T. Validation of a quantitative real-time PCR assay and comparison with fluorescence microscopy and selective agar plate counting for species-specific quantification of an *in vitro* subgingival biofilm model. *Journal of Periodontal Research* **48**, 517-526 (2013). <https://doi.org:10.1111/jre.12034>
- 26 Dalwai, F., Spratt, D. A. & Pratten, J. Use of quantitative PCR and culture methods to characterize ecological flux in bacterial biofilms. *Journal of Clinical Microbiology* **45**, 3072-3076 (2007). <https://doi.org:10.1128/jcm.01131-07>
- 27 Guilbaud, M. *et al.* Quantitative detection of *Listeria monocytogenes* in biofilms by real-time PCR. *Applied and Environmental Microbiology* **71**, 2190-2194 (2005). <https://doi.org:10.1128/aem.71.4.2190-2194.2005>
- 28 Burrowes, B. H., Molineux, I. J. & Fralick, J. A. Directed *in vitro* evolution of therapeutic bacteriophages: the Appelmans protocol. *Viruses* **11**, 241 (2019). <https://doi.org:10.3390/v11030241>

Reviewer #4

*This manuscript explores the possibility of directing phage evolution towards more effective infective processes against planktonic and biofilm forming strains of *Pseudomonas aeruginosa*. It uses experimental evolution of a mixture of characterized phages against 8 different strains of *P. aeruginosa*. The authors find that there is an overall gain in infectivity, that initial mutations among evolved phages are mainly associated to structural genes, and that bacterial phage resistance evolution can also concomitantly lead to trade-offs that maximize infectivity in the long-term and minimize resistance evolution. The findings are interesting and relevant, and well analyzed and presented. I only have a major criticism associated to the choice of phage mixture for experimental evolution. I explain more in detail below:*

[Major comments]

- The authors used a mix of four phages for the directed evolution and not have control treatments with the individual phages against the strains used. This is in principle fine, and it may also have been a choice of feasibility which is very important and valid. My main concern is that there is no tracking implemented during the rounds of evolution of the proportion of the phages at the end of each round, and hence each round/strain/replicate might have had different evolutionary outcomes and selection processes due to these differences. Not having information about the proportions of the phages throughout makes it unclear which evolutionary path is being selected and how this influences the final outcome. This might be the reason why there is not much information about the evolved lines of phage JS or FIM.*

Response to comment: Thank you for raising this point. Our study represents a proof of concept for this novel *in vitro* directed phage evolution platform allowing for the improvement of multiple phage parameters in parallel by 30 serial passages. For each round, the phage mixture combining all active samples from the previous round was co-incubated with the eight non-evolving pre-established biofilm strains. The decision to work with a mixture of four phages rather than individual phages, was based on two factors. Firstly, Burrowes *et al.* showed that their control experiments with individual phages in their serial passage evolution assay demonstrated little host-range expansion, the target of their assay¹. Secondly, the efficient allocation of personal and research resources was considered. Feasibility was a decisive factor in the decision to not continuously track the phages, as it would have involved the isolation, sequencing, and further testing of phages after each of the 30 rounds, while not allowing for a fully quantitative analysis of phage proportions due to a process inherent qualitative selection (line 605ff.).

Instead, we focused on two time points (rounds 15 and 30), to isolate, sequence and characterize evolved phages. Based on plaque morphology and host strain, we isolated 31 individual evolved phages from the phage mixtures (17 from round 15 and 14 from round 30). This provided us insights into the evolved infective capabilities of the phages (line 145ff.) and underlying genomics (line 264ff.) for all ancestral and corresponding evolved phages. Exemplary, across all tested strain-phage combinations, phages evolved

for thirty rounds demonstrated a greater antibiofilm activity compared to their unevolved ancestors and, with one exception, than their counterparts evolved for fifteen rounds. Evolved phages (MK.R3-15, MK.R3-30, MK.R84-15, MK.R84-30, MK.R57-15, MK.R57-30) appear to be a recombination of the ancestral phages MK and JS. In reference to Supplementary Data S2 included in the file “NCOMMS-23-23549A_DBPR_Supplementary_Information” the evolved MK phages show a high degree of sequence identity (BLASTn) to both ancestral phages JS and MK, which show 96.24% sequence identity between them. With one exception, the homology of the evolved phages is highest to the ancestral phage MK. Although phage MK.R3-15 has a higher sequence identity with phage JS (98.02%) than with MK (97.53%), it was also named after MK for an easier readability of the manuscript.

We have further adapted the manuscript to highlight these points.

Line 297ff. (r.): Evolved phages derived from the phages MK and JS (MK.R3-15, MK.R3-30, MK.R84-15, MK.R84-30, MK.R57-15, MK.R57-30) appear to be a recombination of these two ancestral phages.

Line 522ff. (d.): Thus, determining the phage proportions in each round and isolating phages for characterisation, not just in round 15 and 30, would have provided more insights into the phages' adaptive pathway within the evolution assay.

- *It is also unclear if the authors have information about potential interference between the phages against the different strains or if there are synergistic effects between the phages.*

Response to comment: Both synergistic and antagonistic interferences among the ancestral phages (FIM, FJK, MK and JS) were not experimentally evaluated, as the focus of our study was to improve the phages' antibiofilm and antimicrobial efficacy. We propose to indicate this as a limitation of the study in the discussion.

We have revised the manuscript accordingly.

Line 525ff. (d.): Investigating synergistic and antagonistic interferences within the evolutionary phage mixture or the two-phage cocktail, would have also provided a deeper understanding of phage-phage interactions and the evolutionary outcomes^{59,60}.

- *Overall, not knowing how the proportion of phages changes over time and how the phages interfere or interact with each other make it hard to assess the full potential of the method/system and the phages individually. Perhaps evolved lines of phage JS could have been more effective than any of the others derived from the mixtures. Without those additional treatments/controls it is impossible to determine. I understand that it would have been a massive amount of work. Perhaps the authors can acknowledge those limitations more thoroughly in the discussion and/or in the beginning of the methods.*

Response to comment: Thank you for this comment. We have addressed this directly in the revised the manuscript, as requested.

Line 522ff. (d.): Thus, determining the phage proportions in each round and isolating phages for characterisation, not just in round 15 and 30, would have provided more insights into the phages' adaptive pathway within the evolution assay.

Line 525ff. (d.): Investigating synergistic and antagonistic interferences within the evolutionary phage mixture or the two-phage cocktail, would have also provided a deeper understanding of phage-phage interactions and the evolutionary outcomes^{59,60}.

Minor comments

- I was not familiar with heat suppression as a method to determine growth. I would appreciate a description of the principle or the justification for its use in the methods, or through the results. I am aware the authors mention a paper where the whole setup is described, but that is not enough.

Response to comment: We gladly accommodate this request.

Line 671ff. (m.): The *in vitro* phage evolution assay to improve multiple phage parameters in parallel, consisted of a serial passaging approach with thirty consecutive rounds, inspired by the directed evolution approach of the Appelmans protocol⁸¹. Adaptations to accommodate bacterial biofilms included the use of microcalorimetric real-time monitoring, revised active sample criteria and performance-dependant phage-mixture-dilutions.

Line 691ff. (m.): A 48-channel isothermal microcalorimeter (TAM III; TA Instruments, New Castle, USA) was used to monitor, in real time and with high sensitivity (0.2 μ W), the heat flow produced by each sample in each round. The heat flow, proportional to the observed exothermic biological processes, allows for an assessment of microbial metabolism, such as that bacterial growth is indicated by an increased heat flow, while the suppression and eradication of these bacteria results in delayed or absent heat production²⁻⁴.

- Related to above, it is unclear why a 75% heat reduction is the threshold selected by the authors. More clarity about their methods, their measures and what those represent would be helpful.

Response to comment: We concur with this request. Within the *in vitro* evolution assay, active samples were defined based on a reduction in heat (J) of $\geq 75\%$ compared to the corresponding growth control sample. This threshold was newly established to select isothermal microcalorimetry samples for subsequent rounds and relates to a reduction in microbial metabolism indicated by heat production of $\geq 75\%$ due to phage predation. The

threshold was set at $\geq 75\%$ to detect valid phage activity and at the same time avoid excluding too many samples, which in turn would reduce the phage diversity of our assay.

Please note below the specific changes introduced to clarify the methods.

Line 703ff. (m.): Active samples were defined based on a reduction in heat (J) of $\geq 75\%$ compared to the corresponding growth control sample. This threshold strikes a balance between detecting phage activity through bacterial heat production reduction and the exclusion of samples which would reduce phage diversity. During the first fifteen rounds of the evolution assay, the comparative heat reduction analysis was conducted considering the cumulative heat after 24 h, while from round 16 onwards, the cumulative heat from the initial 8 h of the assay were considered (Figure 1c). Active samples and samples containing the undiluted phage mixture across all bacterial strains were pooled into a single mixture after each round. This pooled mixture was centrifuged at 7,000 rpm for 20 min and the supernatant filtered (0.22 μm) before introduction into the next round.

- Is plaque morphology sufficient to differentiate between and among the ancestral and evolved phages? It would be helpful to have either a supplementary figure or a main figure showing the differences in plaques between the phages. Also, is there any knowledge of the likelihood of plaque morphology to change during experimental evolution?

Response to comment: These are interesting points. The phage mixtures of rounds 15 and 30 were spotted on all eight *P. aeruginosa* evolution strains (PAO1, Paer03, Paer09, Paer33, Paer57, Paer60, Paer84 and Paer85) and based on the host strain and plaque morphology all differing phage plaques (n=31) were picked, amplified, and sequenced to determine their descent. The differentiation between ancestral and evolved phages was based on the whole genome sequencing of all phages. Figure E3 included in the file “NCOMMS-23-23549A_DBPR_Extended_Data” represents this isolation process of the evolved phages.

Gallet *et al.* could show that individual phage traits (adsorption rate, lysis timing, phage morphology) influenced the plaque size, plaque productivity, and average phage concentration per plaque⁵. This in turn, would support the idea that phage traits altered by evolution experiments could lead to different phage plaque morphologies, but we do not want to postulate this, as this was neither the goal of our study nor do we have sufficient information for a definitive answer.

We have revised the manuscript to address the questions raised.

Line 155f. (r.): Based on plaque morphology and host strain, we isolated 31 individual evolved phages from the phage mixtures (17 from round 15 and 14 from round 30) (Extended Data Figure E3).

Line 608ff. (m.): Based on qualitative plaque assessment and host strain we identified 31 phages that were purified and produced on their isolation strains as described above

(Extended Data Figure E3). Of those, 10 (MK.R3-15, MK.R3-30, MK.R57-15, MK.R57-30, MK.R84-15, MK.R84-30, FIM.R60-15, FIM.R60-30, FJK.R9-15 and FJK.R9-30), representing phages descended from the different ancestral phages determined by BLASTn v2.13.0⁷⁹, were concentrated and purified by PEG 8000 precipitation⁷⁸ before storage in SM-buffer at 4 °C for further use.

Extended Data Figure E3:

Fig. E3: Overview of the isolation of evolved bacteriophages after rounds 15 and 30.

a, Tenfold serial dilutions of the bacteriophage (phage) mixtures after round 15 and 30 were individually spotted (5 μ l) on soft agar overlays for each individual bacterial strain in the evolution assay (PAO1, Paer03, Paer09, Paer33, Paer57, Paer60, Paer84 and Paer85). **b**, Based on the host strain and qualitative plaque assessment (plaques' size, contour, and turbidity/clarity) 17 evolved

phages were picked from round 15 spot assays and 14 from round 30. The phages were purified by four consecutive single-plaque-passages. **c**, Each isolated phage was produced from a single plaque using either a liquid or solid propagation method. **d**, Phage genomes were extracted, sequenced with Illumina, and assembled before the corresponding ancestral phage was identified using BLASTn.

- In Figure 2 the colors of the clusters in the tree are not really easy to visualize, perhaps thicker lines would be better

Response to comment: We concur and have revised Figure 2a accordingly.

Figure 2:

Fig. 2: Host range analysis of unevolved and evolved phages.

- The data shown in figure 2 is very interesting to me. I would appreciate some more information or discussion. Were the losses and gains in infectivity genetically related, i.e., do the gains in infectivity occur in clusters genetically different from those it had or lost infectivity before? Also, are the changes in infectivity or number of strains each phage is able to infect, significant?

Response to comment: Thank you. The host range of each phage was determined by soft agar overlay spot assays against the entire collection of *P. aeruginosa* strains. Infectivity gains are defined as the capability of an evolved phage to infect a bacterial strain that was initially not susceptible to the phage's genomically closest unevolved progenitor phage. A loss of infectivity occurred when a bacterial strain is susceptible to the unevolved ancestral phage but resistant to the corresponding evolved phage. The experiment was either conducted as two biological replicates with two technical replicates each (ancestral phages) or as three biological replicates (evolved phages). Due to identical outcomes for each replicate, there is no data variation, and a significance would not be informative.

Among the strains not susceptible to any phage (n=30; CRS, completely resistant strains) antibiotic resistant strains (3 or 4 MRGN) make up 43.3% (n=13) while only comprising 30.0% of the entire *P. aeruginosa* collection (n=80). In total, 67 infectivity gains were found on 25 strains, while 28 losses occurred on 14 strains. Between these two groups only three strains exhibit both gains and losses simultaneously. Both infectivity gains and losses occurred mostly on strains from cluster 1 while the overrepresentation is lower for the gains group (73.1% vs. 71.3% among all strains) than the strains with infectivity losses (89.3% vs. 71.3% among all strains).

We have adapted the manuscript to introduce additional information.

Line 170ff. (r.): In total, 67 infectivity gains were found on 25 strains, while 28 losses occurred on 14 strains. Between these two groups only three strains (Paer58, Paer85 and Paer90) exhibited both gains and losses simultaneously.

- Line 324-325, this sentence needs revision.

Response to comment: Apologies, we have revised the phrase accordingly.

Line 335ff. (r.): Altogether, 43 of the 48 isolates (89.6%) were found to be resistant to phage FJK (n=16, isolates from FJK incubation), while for evolved phages FJK.R9-15 (n=17) and FJK.R9-30 (n=15), 25 (52.1%) and 21 (43.8%) resistant isolates were found, respectively.

- It is unclear why the authors do optical density in figure 5 and 6 instead of the previous calorimetric method. Perhaps a line or two clarifying why this distinction matters at this point would be helpful.

Response to comment: This decision was based on available laboratory resources and does not follow a specific scientific reasoning, as optical density measurement and isothermal microcalorimetry can be used complementarily.

References

- 1 Burrowes, B. H., Molineux, I. J. & Fralick, J. A. Directed *in vitro* evolution of therapeutic bacteriophages: the Appelmans protocol. *Viruses* **11**, 241 (2019). <https://doi.org/10.3390/v11030241>
- 2 Braissant, O. *et al.* Isothermal microcalorimetry accurately detects bacteria, tumorous microtissues, and parasitic worms in a label-free well-plate assay. *Biotechnology Journal* **10**, 460-468 (2015). <https://doi.org/10.1002/biot.201400494>
- 3 Braissant, O., Wirz, D., Göpfert, B. & Daniels, A. U. Use of isothermal microcalorimetry to monitor microbial activities. *FEMS Microbiology Letters* **303**, 1-8 (2010). <https://doi.org/10.1111/j.1574-6968.2009.01819.x>
- 4 Butini, M. E. *et al.* Real-time antimicrobial susceptibility assay of planktonic and biofilm bacteria by isothermal microcalorimetry. *Advances in Microbiology, Infectious Diseases and Public Health: Volume 13*, 61-77 (2019). https://doi.org/10.1007/5584_2018_291
- 5 Gallet, R., Kannoly, S. & Wang, I. N. Effects of bacteriophage traits on plaque formation. *BMC Microbiol* **11**, 181 (2011). <https://doi.org/10.1186/1471-2180-11-181>

REVIEWER COMMENTS

Reviewer #1 (Remarks to the Author):

I would like to congratulate the authors for the revision that very much improved the manuscript. Additional experiments (complementation, virulence assay) are convincing. All my concerns and comment have been satisfactorily addressed. Comments from other reviewers have been in my opinion have also been addressed.

The author might discuss perhaps why resistant bacteria only exhibit one mutation in each isolate. Are those mutations mutually exclusive?

Reviewer #2 (Remarks to the Author):

I reviewed this manuscript approx 9 months ago and I certainly appreciate the efforts made by the authors to answer the several questions raised by the reviewers (rebuttal file of 58 pages). Specifically, I valued the inclusion of new data on *S. aureus*. While I agree it may not warrant inclusion in the manuscript, it underscores the platform's relevance to other pathogens, particularly in broadening the host range. On the other hand, antibiofilm activity appears to vary depending on the bacterial strain in *S. aureus*.

I also appreciate the new data on the characterization of the mutated strains in terms of their growth, virulence in the Galleria model, biofilm formation and motility. The phage data on the complemented mutant is also noteworthy. Although the EOPs may need to be recalculated.

Surprisingly, the authors still didn't compare the evolved phage cocktail to an ancestral phage cocktail. If the overall hypothesis was that evolved phages would be better in phage cocktail than ancestral phages, why not compare it at the end?

I still have a few minor comments

1. Figure 2 : The color legend should be moved up, closer to the appropriate panel.
2. Figure 5: I would add the EOP color code to panel A. Panels D-E-F-G should be color coded to identify each phage and to clearly indicate that black lines represent individual replicates.
3. Figure 6: Panel A, while I saw that the EOP color code was explained in the legend of the Figure but perhaps it should be added instead in the Figure itself? Just a suggestion.
4. Figure E4c: The EOP data were calculated using the titer of the non-complemented mutant as the denominator. However, should it be the wild-type strain instead? I was aiming to observe the EOP of the mutated strain on the wild-type and compare it with the EOP of the complemented strain on the wild-type (with the empty plasmid). Additionally, colored maps for this type of data are not optimal; a table displaying the EOPs or even a comparison of plaques would be more effective, with the latter being often used in phage defense system papers nowadays.

5. Figure E4: Panel d, specify that the optical density was measured at 570 nm on the Y axis.
6. L105-108: While I understand the point here, it does not read well. Very unlikely that you had only 1 phage (10^0) and even so, it would not be a phage "mixture".
7. L153 As RT-qPCR could also refers to Reverse Transcriptase quantitative PCR, I would thus suggest changing to qPCR.
8. L181-183: This sentence needs to be clarified "Focusing instead on the entire collection of *P. aeruginosa* strains (n=80), 30 strains (37.5%) were not susceptible to any phage, and overall, the evolved phages were able to increase the number of susceptible strains to 45, an increase of 9.8% ». If 30 / 80 strains were not susceptible, it would mean that 50 are susceptible. If so, the rest of the sentence is incorrect.
9. Line 283-285: Could this be shown either in Figure 4 or in supp mat ? no structural comparison between gp62 and TTPA (or other homologs which could possess the additional helix containing C98) is shown.
10. Line 283: if the enzymatic activity is indeed improved, this platform may be also useful for those interested in improving the antibiofilm activities of phage enzymes.
11. Line 375-378: I recommend mentioning Figure E4 in the text here.
12. Line 560: Include a reference to figure(s).
13. Line 570: Given the time-consuming and labor-intensive nature of the method proposed here, it's highly improbable that this approach would be utilized for phage therapy, particularly since swift treatment of infected patients is presumed necessary. However, it may find utility in isolating newly adapted or evolved phages for inclusion in biobanks. Furthermore, the stability of these evolved phages, especially those with mutations in structural proteins, remains to be determined.
14. Line 631: Minor detail but usually more than four phages are used to calculate dimensions.
15. Line 780: indicate PDB 4-character code instead of article DOI
16. Line 805: Portugal -> Portugal
17. Line 913, One would assume that the clones were confirmed by sequencing?
18. Line 922: Replace non-complimented by non-complemented.
19. Line 924-926: This is an incorrect assumption and should not be done. Another reason to use the wild-type strain (containing the empty plasmid) to determine the EOP.

Reviewer #3 (Remarks to the Author):

The manuscript has been extensively revised and most of the concerns are addressed. This study provided quite a lot of data to describe the evolved phages. And limitations of this study are also discussed.

I only have one minor suggestion:

LINE12, "80% of the human microbial infections are biofilm-associated" . I don't think it is as high as 80%!!!

Point-by-point response

to the reviewers' comments on the manuscript entitled "Targeting MDR *Pseudomonas aeruginosa* biofilm with an evolutionary trained bacteriophage cocktail exploiting phage resistance trade-offs" (NCOMMS-23-23549B).

We would like to thank the reviewers for their constructive scientific comments and final textual suggestions to further strengthen our study. In the revised manuscript, taking all reviewers' comments into consideration, we have made significant clarifying revisions.

For each line reference the corresponding manuscript section is abbreviated as (a.) (abstract), (i.) (introduction), (r.) (results), (d.) (discussion), (m.) (methods) and (r.) (references).

Reviewer #1

I would like to congratulate the authors for the revision that very much improved the manuscript. Additional experiments (complementation, virulence assay) are convincing. All my concerns and comment have been satisfactorily addressed. Comments from other reviewers have been in my opinion have also been addressed.

Response to comment: Thank you very much for your words of praise and appreciation of our revised manuscript.

The authrory might discussed perhaps why resistant bacteria only exhibit one mutations in each isolate. Are those mutations mutually exclusive?

Response to comment: Illumina sequencing of all phage-treated Paer09 isolates (n=48, exposed to individual phages; n=18, exposed to phage cocktail) identified one of seven genes, likely conferring resistance to phage predation, for each isolate. Five of those mutations (WapR, RmlA, RmlB, RmlC and RmlD) are involved in the O-antigen attachment, indicating this to be the potential receptor for FJK-phages. As a consequence, the loss of O-antigen through a single mutation would suffice to cause resistance to FJK-phages. The other two mutations (pslA and glycoside hydrolase) occurred in only three isolates (4.5%).

Although we did not observe any isolate presenting two or more of the above mutations, we would postulate that the mutations are not per se mutually exclusive, but rather selected for based on the competitive advantages of fewer mutational changes within a bacterium. Wichman *et al.* point out that early mutational changes conferring greater boosts in fitness may not always show in all replicates and would therefore result in different adaptation pathways¹. This could further explain why we only identified one mutation per isolate.

We have revised the manuscript accordingly.

Line 523ff. (d.): In addition, Wichman *et al.* point out that early mutational changes conferring greater boosts in fitness may not always show in all replicates, which could explain the fact that each mutant had only a single mutation, which might set them on different adaptation pathways¹.

References

- 1 Wichman, H. A., Badgett, M. R., Scott, L. A., Boulianne, C. M. & Bull, J. J. Different trajectories of parallel evolution during viral adaptation. *Science* **285**, 422-424 (1999). <https://doi.org/10.1126/science.285.5426.422>

Reviewer #2

*I reviewed this manuscript approx 9 months ago and I certainly appreciate the efforts made by the authors to answer the several questions raised by the reviewers (rebuttal file of 58 pages). Specifically, I valued the inclusion of new data on *S. aureus*. While I agree it may not warrant inclusion in the manuscript, it underscores the platform's relevance to other pathogens, particularly in broadening the host range. On the other hand, antibiofilm activity appears to vary depending on the bacterial strain in *S. aureus*.*

*I also appreciate the new data on the characterization of the mutated strains in terms of their growth, virulence in the *Galleria* model, biofilm formation and motility. The phage data on the complemented mutant is also noteworthy. Although the EOPs may need to be recalculated.*

Surprisingly, the authors still didn't compare the evolved phage cocktail to an ancestral phage cocktail. If the overall hypothesis was that evolved phages would be better in phage cocktail than ancestral phages, why not compare it at the end?

Response to comment: Again, we are very thankful for the appreciation of our revised manuscript and the further experimentation that helped clarify the key points raised.

As indicated previously, based on the experimental design of our study the evolved phages have outperformed the unevolved, ancestral phages with regard to antimicrobial activity, as tested within 24-h-biofilms and planktonic Paer09 under isothermal microcalorimetry and optical density monitoring, respectively.

In view of our previous rationale, we have now revised the manuscript to address and clarify textual concerns at the editor's suggestion.

I still have a few minor comments

1. Figure 2 : The color legend should be moved up, closer the appropriate panel.

Response to comment: We concur and have revised figure 2 accordingly.

Figure 2:

2. Figure 5: I would add the EOP color code to panel A. Panels D-E-F-G should be color coded to identify each phage and to clearly indicate that black lines represent individual replicates.

Response to comment: This is a good idea. In addition to the legend, we have included an illustration to explain the EOP colour code in the figure itself. In panels D, E and F, the averaged curves of phage FJK (d), phage FJK.R9-15 (e) and phage FJK.R9-30 (f) are colour coded and thicker than the black lines representing the individual replicates, which we also indicated in the figure itself and in the legend.

We have revised the legend accordingly.

Line 395ff. (r): **d, e, f**, Optical density (OD₆₀₀) measurements of a three-day co-incubation of planktonic naive Paer09 strain with phage FJK (d), phage FJK.R9-15 (e) and phage FJK.R9-30 (f) at an MOI of 0.001. The thin black lines represent each individual replicate (n=8), and the thick coloured line represents the average. For phages FJK.R9-15/30 the completely suppressed replicate was excluded from the average calculation. **g**, Illustration of the averaged co-incubation curves for phages FJK (grey line; panel d), FJK.R9-15 (violet line; panel e) and FJK.R9-30 (yellow line; panel f) over

three days in comparison to a growth control (dotted black line) and negative control (black line).

Figure 5:

3. Figure 6: Panel A, while I saw that the EOP color code was explained in the legend of the Figure but perhaps it should be added instead in the Figure itself? Just a suggestion.

Response to comment: For the reader's convenience we have included an illustration to explain the EOP colour code in the figure itself.

Figure 6:

4. Figure E4c: The EOP data were calculated using the titer of the non-complemented mutant as the denominator. However, should it be the wild-type strain instead? I was aiming to observe the EOP of the mutated strain on the wild-type and compare it with the EOP of the complemented strain on the wild-type (with the empty plasmid). Additionally, colored maps for this type of data are not optimal; a table displaying the EOPs or even a comparison of plaques would be more effective, with the latter being often used in phage defense system papers nowadays.

Response to comment: Co-incubation of planktonic bacterium Paer09 with either ancestral phage FJK or one of the evolved phages (FJK.R9-15/30) resulted in the development of resistance to the treatment phages. Parallel to this resistance, we observed increased susceptibility to MK-phages, resulting in a resistance trade-off between the FK- and MK-phages. Overall, we could identify seven genes, likely conferring resistance to phage predation. To test the role of these genes, we complemented seven representative bacterial mutants with their wild-type gene. The relative efficacy of plating (EOP) was defined as the concentration ratio of a phage on the complemented mutant (numerator) and the non-complemented mutant (denominator).

While we understand the comment given, we prefer to illustrate the genes importance via a comparison with the transformed non-complemented mutant (empty plasmid). This

enables us to focus on the reversal of the trade-off for the gene's involvement in the resistance mechanism, as this trade-off is a central aspect to our study and especially to its storyline. Thereby, in our opinion, it supports the readers' understanding and reduces unneeded complexity.

In line with the illustration of the trade-off (Figure 5a and 6a), we opted to display its reversal in an identical manner, for the readers' convenience.

We have revised the manuscript and, as suggested, provided the EOPs of the plasmid complementation assay in a table in Supplementary Data S5 included in the file "NCOMMS-23-23549B_DBPR_Supplementary_Information".

Line 928 ff. (m.): The relative efficacy of plating (EOP) was defined as the concentration ratio of a phage on the complemented mutant (numerator) and the non-complemented mutant (denominator) (Supplementary Data S5).

5. *Figure E4: Panel d, specify that the optical density was measured at 570 nm on the Y axis.*

Response to comment: Throughout the manuscript, the wavelength of each optical density measurement is indicated in the legend and has not been additionally referenced in the graph itself.

6. *L105-108: While I understand the point here, it does not read well. Very unlikely that you had only 1 phage (10e0) and even so, it would not be a phage "mixture".*

Response to comment: We concur and have revised the phrase accordingly.

Line 105ff. (r.): Ultimately, we observed how approximately 10 PFU/ml (plaque forming units/ml) of the phage mixture resulted in heat reductions of 84.2% and 91.5% for the strains PAO1 (Extended Data Figure E2a; round 30) and Paer57 (Extended Data Figure E2d; round 15), respectively.

7. *L153 As RT-qPCR could also refers to Reverse Transcriptase quantitative PCR, I would thus suggest changing to qPCR.*

Response to comment: Thank you for raising this point. We have revised the manuscript accordingly.

Line 16ff. (a.): The obtained evolved phages show an expanded host spectrum, improved antimicrobial efficacy and enhanced antibiofilm performance, as assessed by isothermal microcalorimetry and qPCR, respectively.

Line 149ff. (r.): For direct comparison of the phages' host range, antimicrobial biofilm activity and antibiofilm efficacy we employed soft agar overlay spot assays, co-incubated pre-established biofilms with phages under isothermal microcalorimetry monitoring and

determined the bacterial biofilm count reduction by real-time quantitative polymerase chain reaction (qPCR).

Line 423ff. (r.): Using optical density monitoring, isothermal microcalorimetry and qPCR, we then compared the cocktails' planktonic and biofilm antimicrobial activity, as well as its antibiofilm efficacy, with the individual phages (Supplementary Table S4).

Line 550ff. (d.): Isothermal microcalorimetry allowed us to continuously monitor the phage-bacterial biofilm interaction in real-time with a high sensitivity and accuracy^{63,64} while qPCR helped us to precisely quantify the antibiofilm degradation capabilities of our phages, providing a starting point for further usage of this technique. As qPCR does not distinguish between live and dead bacteria, the results provided represent a more conservative antibiofilm efficiency, considering the possibility that DNA from dead bacteria not dip-washed away is also quantified.

Line 793ff. (m.): A real-time quantitative polymerase chain reaction (qPCR) was used to quantify the number of viable cells following exposure of 24-h-biofilms of three representative *P. aeruginosa* strains: Paer09, Paer57 (both included in the evolution assay) and Paer36 (not included in the evolution assay) to ancestral and evolved phages isolated at round 15 and 30 by adapting a previously described method¹¹¹.

Line 950f. (m.): qPCR results were statistically analysed using an unpaired two-tailed Student's *t* test analysis integrated in GraphPad Prism 9.

8. L181-183: *This sentence needs to be clarified "Focusing instead on the entire collection of P. aeruginosa strains (n=80), 30 strains (37.5%) were not susceptible to any phage, and overall, the evolved phages were able to increase the number of susceptible strains to 45, an increase of 9.8% ». If 30 / 80 strains were not susceptible, it would mean that 50 are susceptible. If so, the rest of the sentence is incorrect.*

Response to comment: Apologies, you are correct. We have revised the phrase to clarify this misunderstanding. Our biobank comprises 80 *P. aeruginosa* strains. Among those are 30 strains (37.5%) which are neither susceptible to the unevolved nor evolved phages. The remaining 50 strains (62.5%) are either susceptible to one or both the unevolved and evolved phages. Among those 50 strains there are 41 strains (51.3%) susceptible to the unevolved phages and nine strains (11.3%) solely susceptible to the evolved phages. Complementarily there are 47 strains (58.8%) susceptible to the evolved phages (round 15, 45 strains; round 30, 45 strains; 43 strains are susceptible to both) and three strains (3.8%) solely susceptible to the unevolved phages.

We have revised the phrase accordingly.

Line 180ff. (r.): Focusing instead on the entire collection of *P. aeruginosa* strains (n=80), 30 strains (37.5%) were not susceptible to any phage (unevolved and evolved), and overall, the evolved phages (rounds 15 and 30) were able to increase the number of susceptible strains to 47, an increase of 14.6%.

9. Line 283-285: Could this be shown either in Figure 4 or in supp mat ? no structural comparison between gp62 and TTPA (or other homologs which could possess the additional helix containing C98) is shown.

Response to comment: Included in the file “NCOMMS-23-23549B_DBPR_Extended_Data” figure E4 illustrates the tertiary structure of FJK_gp62 and a superimposition of TTPA and FJK_gp62.

Line 283ff. (r.): Compared to TTPA, gp62 contains an additional α -helix close to the β -sheets, in which the mutation occurred, that could have an impact on the enzymatic activity, thereby potentially explaining the increased antibiofilm activity of the evolved phage (Extended Data Figure E4).

Extended Data Figure E4:

Fig. E4: Overview of the tertiary structure for FJK_gp62 and TTPA.

a, Tertiary structure prediction of EPS depolymerase FJK_gp62. In red, the cysteine residue on position 98 is highlighted, which is mutated to the aromatic amino acid phenylalanine in the evolved phage. **b**, Superimposition of EPS depolymerase FJK_gp62 (blue) and the tail tubular protein A (TTPA) of Klebsiella phage KP32 (green). Both structures contain a compact α -helical domain on the one side and β -strands and loops on the other side. These β -strands constitute two antiparallel β -sheets. FJK_gp62 contains an extra α -helix close to these β -sheets (arrow).

10. Line 283: if the enzymatic activity is indeed improved, this platform may be also useful for those interested in improving the antibiofilm activities of phage enzymes.

Response to comment: This is an interesting point and we have introduced the following phrase in our discussion to highlight it.

Line 538ff. (d.): Underpinning the phage improvement, the efficacy of phage-derived enzymes (depolymerases and lysins) could be enhanced, and the identification of such mutational sites could provide new targets for genetic engineering and enzyme-based therapies^{1,2}.

11. Line 375-378: *I recommend mentioning Figure E4 in the text here.*

Response to comment: We concur and have revised the phrase accordingly.

Line 376ff. (r.): Contrarily, the WapR mutation (frameshift) resulted in a 10.5% ($p < 0.05$) greater swarming motility and both the RmlB and WapR mutants had an increased swimming motility of 32.8% ($p < 0.01$) and 68.0% ($p < 0.0001$), respectively (Extended Data Figure E5f, g).

12. Line 560: *Include a reference to figure(s).*

Response to comment: Thank you. We have mentioned the corresponding figure for the reader's convenience.

Line 565ff. (d.): In extension to those experiments, our two-phase cocktail at a concentration of approx. 10^3 PFU/ml (MOI of 0.001) showed a continued suppression of planktonic bacteria (clinical isolate Paer09) up to seven days, while no isolated bacterial mutant had developed a dual-phage-resistance (Figure 6a, g, i).

13. Line 570: *Given the time-consuming and labor-intensive nature of the method proposed here, it's highly improbable that this approach would be utilized for phage therapy, particularly since swift treatment of infected patients is presumed necessary. However, it may find utility in isolating newly adapted or evolved phages for inclusion in biobanks. Furthermore, the stability of these evolved phages, especially those with mutations in structural proteins, remains to be determined.*

Response to comment: We respectfully disagree with this assessment, as in clinical practice under current guidelines, phages are not administered to patients as a first treatment option, but as a last resort under magistral treatment regimens. Please note our recent publication in Nature Microbiology for reference³. Certainly, intensive care units where patients need to be cured rapidly, implementation of a well-defined trained phage would be difficult due to time limitations. However, in the treatment of chronic infections, e.g. in patients with recurrent lung, wound or prosthetic joint infections, there is often more time for a tailored, highly personalized strategy by training phages on patient's strain once it becomes available⁴. As our study represents a proof of concept for this novel *in vitro* phage evolution platform allowing to improve multiple phage parameters in parallel it will need further streamlining by optimizing the number and length of each evolution cycle to improve its generalized applicability.

While the stability of evolved phages will need to be further validated, we see the value that trained phages would add to phage biobanks, especially when prepared as a trained resistance-adapted phage cocktail.

We have revised the manuscript accordingly.

Line 571ff. (d.): Considering clinical applications, biofilm-adapted phages could be included in biobanks and combinations with antibiotics could be assessed, as could the translatability from *in vitro* to *in vivo* models^{69,70}.

14. *Line 631: Minor detail but usually more than four phages are used to calculate dimensions.*

Response to comment: We agree, but due to the limited number of virus particles for phage FJK with sufficient resolution, we decided to measure four particles for each phage.

15. *Line 780: indicate PDB 4-character code instead of article DOI*

Response to comment: Thank you for pointing this out.

Line 786f. (m.): The related TTPA structure was downloaded from the Protein Data Bank (PDB code 5MU4).

16. *Line 805: Portugal -> Portugal*

Response to comment: We have corrected this typo.

Line 810ff. (m.): The NZYTech *Pseudomonas aeruginosa* Real-time PCR Kit targeting the toxin A synthesis regulating gene (RegA) was used according to the manufacturer's instructions (MD02381; NZYTech, Lisboa, Portugal).

17. *Line 913, One would assume that the clones were confirmed by sequencing?*

Response to comment: For the complementation assay, the seven representative bacterial mutants were transformed with plasmids containing their respective wild-type genes. Successful transformation was confirmed by colony PCR and Sanger sequencing to verify the presence of correct plasmids after selection for plasmid-carrying strains with 200 µg/ml carbenicillin in the growth mediums.

We have revised the manuscript accordingly.

Line 920ff. (m.): Controls were transformed with empty plasmids (non-complemented mutants) and induction for both mutants was carried out with a final concentration of 0.2% arabinose, while a final concentration of 200 µg/ml carbenicillin was added to select for the plasmid-carrying strains, confirmed by colony PCR and Sanger sequencing.

18. *Line 922: Replace non-complimented by non-complemented.*

Response to comment: Thank you for noticing this typo.

Line 928ff. (m.): The relative efficacy of plating (EOP) was defined as the concentration ratio of a phage on the complemented mutant (numerator) and the non-complemented mutant (denominator) (Supplementary Data S5).

19. Line 924-926: This is an incorrect assumption and should not be done. Another reason to use the wild-type strain (containing the empty plasmid) to determine the EOP.

Response to comment: To verify the mutations' involvement in the phage resistance we performed a complementation assay with the cloned wild-type genes transformed to a representative subset of seven phage resistant bacterial mutants, one for each mutation observed. Phage susceptibility was evaluated by spotting tenfold serial dilutions of the phages (FJK, FJK.R9-15, FJK.R9-30, MK, and MK.R3-15) on soft agar overlays. The relative efficacy of plating (EOP) was defined as the concentration ratio of a phage on the complemented mutant (numerator) and the non-complimented mutant (denominator). As with any division, the denominator cannot be 0 when calculating the EOP. Therefore, in cases where no phage plaques were visible, we opted for a conservative approach, assuming that the phage concentration corresponded to our detection limit for the spot assay.

Phage MK was not active on the wild-type strain (Figure 5a and 6a, Control), which would lead to the same challenge as described above when using the wild-type Paer09 in the denominator for the EOP calculation.

References

- 1 Glonti, T., Chanishvili, N. & Taylor, P. W. Bacteriophage-derived enzyme that depolymerizes the alginic acid capsule associated with cystic fibrosis isolates of *Pseudomonas aeruginosa*. *Journal of Applied Microbiology* **108**, 695-702 (2010). <https://doi.org:10.1111/j.1365-2672.2009.04469.x>
- 2 Olszak, T. *et al.* The O-specific polysaccharide lyase from the phage LKA1 tailspike reduces *Pseudomonas* virulence. *Scientific Reports* **7**, 16302 (2017). <https://doi.org:10.1038/s41598-017-16411-4>
- 3 Pirnay, J. P. *et al.* Personalized bacteriophage therapy outcomes for 100 consecutive cases: a multicentre, multinational, retrospective observational study. *Nature Microbiology* **9**, 1434-1453 (2024). <https://doi.org:10.1038/s41564-024-01705-x>
- 4 Rohde, C. *et al.* Expert Opinion on Three Phage Therapy Related Topics: Bacterial Phage Resistance, Phage Training and Prophages in Bacterial Production Strains. *Viruses* **10**, 178 (2018).

Reviewer #3

The manuscript has been extensively revised and most of the concerns are addressed. This study provided quite a lot of data to describe the evolved phages. And limitations of this study are also discussed.

I only have one minor suggestion:

LINE12, “80% of the human microbial infections are biofilm-associated” . I don’t think it is as high as 80%!!!

Response to comment: Thank you for your comment. We concur that this is a high percentage of cases. The exact number stems from two grant announcements by the National Institute of Health (NIH)^{1,2}. The Centers for Disease Control and Prevention (CDC) estimates that 65% of bacterial infections in humans involve biofilms^{3,4}.

We have revised the manuscript accordingly.

Line 12ff. (a.): However, while up to 80% of human microbial infections are biofilm-associated, research towards targeted improvement of phages’ ability to combat biofilms remains scarce.

Line 42 (i.): Biofilms cause between 65% to 80% of human microbial infections¹⁻⁴.

References

- 1 <https://grants.nih.gov/grants/guide/pa-files/pa-03-047.html>. (National Institute of Health, 2002).
- 2 <https://grants.nih.gov/grants/guide/pa-files/PA-06-537.html>. (National Institute of Health, 2006).
- 3 Potera, C. Forging a link between biofilms and disease. *Science* **283**, 1837, 1839 (1999). <https://doi.org:10.1126/science.283.5409.1837>
- 4 Costerton, J. W. Cystic fibrosis pathogenesis and the role of biofilms in persistent infection. *Trends Microbiol* **9**, 50-52 (2001). [https://doi.org:10.1016/s0966-842x\(00\)01918-1](https://doi.org:10.1016/s0966-842x(00)01918-1)